# Rare and common vertebrates span a wide spectrum of population trends

Gergana N. Daskalova [1✉], Isla H. Myers-Smith [1] & John L. Godlee [1]

The Earth's biota is changing over time in complex ways. A critical challenge is to test whether specific biomes, taxa or types of species benefit or suffer in a time of accelerating global change. We analysed nearly 10,000 abundance time series from over 2000 vertebrate species part of the Living Planet Database. We integrated abundance data with information on geographic range, habitat preference, taxonomic and phylogenetic relationships, and IUCN Red List Categories and threats. We find that 15% of populations declined, 18% increased, and 67% showed no net changes over time. Against a backdrop of no biogeographic and phylogenetic patterning in population change, we uncover a distinct taxonomic signal. Amphibians were the only taxa that experienced net declines in the analysed data, while birds, mammals and reptiles experienced net increases. Population trends were poorly captured by species' rarity and global-scale threats. Incorporation of the full spectrum of population change will improve conservation efforts to protect global biodiversity.

[1] School of GeoSciences, University of Edinburgh, Edinburgh EH9 3FF Scotland, UK. ✉email: gndaskalova@gmail.com

Ecosystem-level change is currently unfolding all around the globe and modifying the abundances of the different species forming Earth's biota. As global change continues to accelerate[1,2], there is a growing need to assess the factors explaining the variation in ecological changes observed across taxa and biomes[3]. However, existing empirical studies of the predictors of the abundance of individuals of different species over time (hereafter, population change) mostly focus on either specific taxa[4] or on population declines alone[2,5]. A critical research challenge is to disentangle the sources of heterogeneity in available data across the full spectrum of population change. Recent compilations of long-term population time series, extensive occurrence, phylogenetic, habitat preference and IUCN Red List Category data[6–8] provide a unique opportunity to test which species- and population-level attributes explain variation in population trends and fluctuations among vertebrate species monitored around the world. Such population change is the underlying process leading to community reassembly[9] and the resulting alterations to biodiversity are vitally important for ecosystem functions and services[10].

The distributions of global change drivers such as land-use change, habitat change, pollution, invasion by non-native species and climate change show distinct clustering across space[11–13]. Spatial clustering has also been documented for biodiversity trends derived from assemblage time series, with the marine realm emerging as a hotspot for rapid changes in community composition[14]. As assemblages are made up of populations, the biogeographic patterns at the assemblage level suggest similar clustering might occur at the population level as well[15]. In addition to geographic patterns in exposure to anthropogenic activities, species' vulnerability and traits can moderate population responses to natural and anthropogenic environmental change[16], both across evolutionary time[6–8] and in the modern day[17,18]. Building on known variability in species' vulnerability[16,19,20], we expected taxonomic and phylogenetic signals in population trends and fluctuations (e.g., greater declines, increases, or fluctuations in abundance for specific taxa and among specific clades). Understanding which biomes, taxa and types of species are experiencing the most acute changes in abundance over time could provide key insights for conservation prioritisation.

Conservation efforts often focus on protecting rare species—those with restricted geographic extents, small population sizes or high habitat specificity—as they are assumed to be more likely to decline and ultimately go extinct[21–23]. Species with a smaller geographic range might have more concentrated exposure to environmental change, with fewer opportunities to find refugia or disperse, thus increasing the likelihood of declines[1,9]. As per population dynamics theory[24,25] and Taylor's power law[26], species with small populations are more likely to undergo stochastic fluctuations that could lead to pronounced declines, local extinction and eventually global extinction[5]. Small populations are also more likely to decline owing to inbreeding, but there are also instances of naturally small and stable populations[27,28]. Allee effects, the relationship between individual fitness and population density, further increase the likelihood of declines due to lack of potential mates and low reproductive output once populations reach a critically low density[29,30]. Furthermore, environmental change might have disproportionately large effects on the populations of species with high habitat specificity, as for these species persistence and colonisation of new areas are limited by strict habitat preferences[1,31]. The fossil record indicates that on millennial time scales, rare species are more likely to decline and ultimately go extinct[32], but human actions have pushed Earth away from traditional geological trajectories[33], and the relationships between rarity and population change across the planet have yet to be tested across the Anthropocene.

On a global scale, species are exposed to a variety of threats, among which habitat change, resource exploitation and hunting dominate as key predictors of extinction risk[34]. Species' IUCN Red List Categories are often used in conservation prioritisation and more threatened species tend to be the focus of conservation initiatives[35]. At more local scales, there might be variation in how populations are changing over time in different locations, in isolation from their overall conservation status[4,36]. Testing population change across species' IUCN Red List Categories (Supplementary Fig. 16) allows us to link contemporary changes in abundance with long-term probability of extinction[37]. Determining how local-scale population trends vary across species' IUCN Red List Categories has practical applications for assessing species' recovery, which is useful for the proposed IUCN Green List of Species[38].

Here, we ask how the trends and fluctuations of vertebrate populations vary with biogeography, taxa, phylogenetic relationships and across species' rarity metrics and IUCN Red List Categories and threat types from the species' IUCN Red List profiles. We test the following predictions: (1) There will be biogeographic patterns in population trends and fluctuations across the planet's realms and biomes, in line with particular regions of the world experiencing high rates of environmental change (e.g., tropical forests[39]). (2) Populations of rare species will be more likely to decline and fluctuate than the populations of common species. (3) Populations of species with a higher IUCN Red List Category and higher number of threats will be more likely to decline and fluctuate than the populations of least concern species and those exposed to a lower number of threats. We quantify differences in population trends and fluctuations across latitudes and biomes within the freshwater, marine and terrestrial realms to test the presence of distinct hotspots of declines and increases. In addition, we use data from the VertLife and BirdLife Databases[6–8] to assess taxonomic and phylogenetic signals. We measure rarity using three separate metrics—geographic range derived from GBIF records, mean population size (mean number of individuals that were recorded during the monitoring for each population in the Living Planet Database) and habitat specificity derived from the species' IUCN Red List profiles. In a post hoc analysis, we compile threat types and number of threats derived from the species' IUCN Red List profiles to determine how threats influence local-scale population change. Using the largest currently available compilation of population records over time, we conduct a global synthesis of population trends and fluctuations to provide key empirical evidence for the management, conservation and prediction of ecological changes across the Anthropocene.

We show that vertebrate species from shark, bony fish, amphibian, bird, mammal and reptile taxa span a wide spectrum of population change across four decades. Among the heterogeneous population change, we highlight amphibians as a taxon in decline. The diverse range of trends and fluctuations over time was not influenced by species' rarity, particularly, their geographic range, mean population size or habitat specificity. Overall, we demonstrate that the abundances of monitored vertebrates around the world are being altered in a variety of ways, testifying to the complexity in species responses to global change across the biomes of the world.

## Results

**Overall approach.** We analysed 9286 vertebrate population time series from 2084 species part of the Living Planet Database (133,092 records) over the period between 1970 and 2014. These time series represent repeated monitoring surveys of the number of individuals in a given area (species' abundance over time),

hereafter called 'populations'. We focused on two aspects of population change—overall changes in abundance over time (population trend, $\mu$) and abundance variability over time (population fluctuations, $\sigma^2$). In the first stage of our analyses, we quantified trends and fluctuations for each population using state-space models that account for observation error and random fluctuations[40] (Supplementary Fig. 1). In the second stage, we modelled the population trend and fluctuation estimates from the first stage across latitude, realm, biome, taxa, rarity metrics, phylogenetic relatedness, species' IUCN Red List Categories and threat type using a Bayesian modelling framework (Supplementary Fig. 2). We included a species random intercept effect to account for the possible correlation between the trends of populations from the same species (see table Supplementary Table 1 for sample sizes). As sensitivity analyses, we additionally used variance weighting of the population trend estimates ($\mu$) by the observation/measurement error around them ($\tau^2$) and population trend estimates from linear model fits (slopes instead of $\mu$) as the input variables in the second-stage models, as well as several different fluctuations estimates. We also repeated our analyses on a single-country scale, using only populations within the United Kingdom, where monitoring efforts are particularly rigorous and extensive. All different analytical approaches yielded very similar results and are described in further detail in the methods and Supplementary Figs. 1–2, 16.

**Vertebrate population change.** We found a broad spectrum of trends across vertebrate populations within the Living Planet Database. Across the time series we analysed, 15% (1381 time series) of populations were declining, 18% (1656 time series) were increasing and 67% (6249 time series) showed no net changes in abundance over time, in contrast to a null distribution derived from randomised data (Supplementary Fig. 5b). Trends were considered statistically different from no net change when the confidence intervals around the population trend estimates did not overlap zero. Our results were similar when we weighted population trends by the observation error derived from the state-space models (Figs. 1–4 and Supplementary Tables 2, 3).

**Biogeographic patterns of population change.** We found that globally, population increases, declines and fluctuations over time occurred across all latitudes and biomes within the freshwater, marine and terrestrial realms, with no strong biogeographic patterning and no specific hotspots of population declines (Fig. 1, Supplementary Table 2). Across realms, monitored vertebrate populations experienced net population increases (freshwater slope = 0.005, CI = 0.002–0.01; marine slope = 0.004, CI = 0.002–0.01; terrestrial slope = 0.003, CI = 0.001–0.005, Fig. 1d, e). In the freshwater and terrestrial realms, there was a bimodal distribution of population trends, driven largely by terrestrial bird species showing small increases and decreases over time (Hartigans' dip test, $D = 0.04$, $p < 0.01$). Across biomes, populations in Mediterranean forests, montane grasslands, polar freshwaters, temperate wetlands, tropical forests, and tropical coral biomes were more likely to increase, whereas populations from the remaining studied biomes experienced no net changes (Fig. 1h, Supplementary Table 2). Population fluctuations were less pronounced in the terrestrial realm (slope = 0.02, CI = 0.018–0.021, Fig. 1f, g), but those populations were also monitored for the longest duration across systems (average duration−28 years for terrestrial, 18 years for marine and 21 years for freshwater populations, Supplementary Fig. 3, Supplementary Table 2).

**Taxonomic and phylogenetic patterns of population change.** We found taxonomic, but not phylogenetic patterns, in

population trends and fluctuations over time among ~10,000 populations from over 2000 vertebrate species, with amphibians emerging as the taxa experiencing pronounced declines (Fig. 2, Supplementary Table 2). Amphibians experienced net declines over time (slope = −0.01, CI = −0.02 to −0.005), whereas birds, mammals and reptiles experienced net increases (slope = 0.004, CI = 0.003 to 0.01; slope = 0.01, CI = 0.01–0.01; slope = 0.02, CI = 0.01–0.02), with birds having a bimodal trend distribution, indicating greater numbers of increasing and decreasing trends (Hartigans' dip test, $D = 0.04$, $p < 0.01$, Fig. 1a, see Supplementary Figs. 5, 6 and 12). Bony fish population trends were centred on zero (slope = −0.001, CI = −0.004–0.002, Fig. 1a, b) and sharks and rays showed net declines, but the credible intervals overlapped zero (slope = −0.01, CI = −0.02–0.01). Fluctuations were most common for amphibian populations (slope = 0.04, CI = 0.036–0.049, Fig. 2d), which were monitored for the shortest time period on average (11 years, Supplementary Fig. 3, Supplementary Table 2). We did not detect finer scale species-level phylogenetic clustering of population change (both trends and fluctuations) within amphibian, bird and reptile classes (Fig. 2, Supplementary Fig. 15, Supplementary Table 4). Similarly, species identity within amphibian, bird and reptile classes did not explain variation in population trends or fluctuations (Fig. 2, Supplementary Fig. 15, Supplementary Table 4). There were no distinct clusters of specific clades that were more likely to undergo increases, decreases or fluctuations in population abundance (Fig. 2).

**Population change across rarity and threats.** Species-level metrics, such as rarity and global IUCN Red List Category, did not explain the heterogeneity in trends of monitored populations in the Living Planet Database. Both rare and common species experienced declines, increases and fluctuations in population abundance over time (Figs. 3 and 4). Across these time series, species with smaller ranges, smaller population sizes or narrower habitat specificity (i.e., rare species) were not more prone to population declines than common species (Fig. 3, Supplementary Table 2). Populations that experienced more fluctuations had smaller mean population sizes on average (slope = −0.001, CI = −0.001 to −0.001, Fig. 3f). We found increasing, decreasing and stable populations across all IUCN Red List Categories (Fig. 4a). For example, a population of the least concern species red deer (*Cervus elaphus*) in Canada declined by 68% over seven years going from 606 to 194 individuals and a population of the critically endangered Hawksbill sea turtle (*Eretmochelys imbricate*) from Barbados increased by 269% over seven years going from 89 to 328 individuals. We found more fluctuations (least concern: slope = 0.022, CI = 0.021–0.023; critically endangered: slope = 0.035, CI = 0.028–0.041, Supplementary Fig. 18), but not more population declines, with increasing IUCN Red List Category (Fig. 4, Supplementary Table 2). Populations from species with a higher number of threats from the species' IUCN Red List profiles did not experience greater declines when compared with those categorised with a smaller number of threats (Fig. 4f). There were no distinct signatures of threats from the species' IUCN Red List profiles that were associated with predominantly declining local trends of monitored populations (Fig. 4e) and there were increasing, decreasing and stable trends across all threat types.

**Discussion**
Taken together, our analysis of ~10,000 vertebrate population time series using a state-space modelling approach demonstrated ubiquitous alterations in vertebrate abundance over time across all biomes on Earth. We revealed that population change includes both increasing and decreasing populations and spans a wide

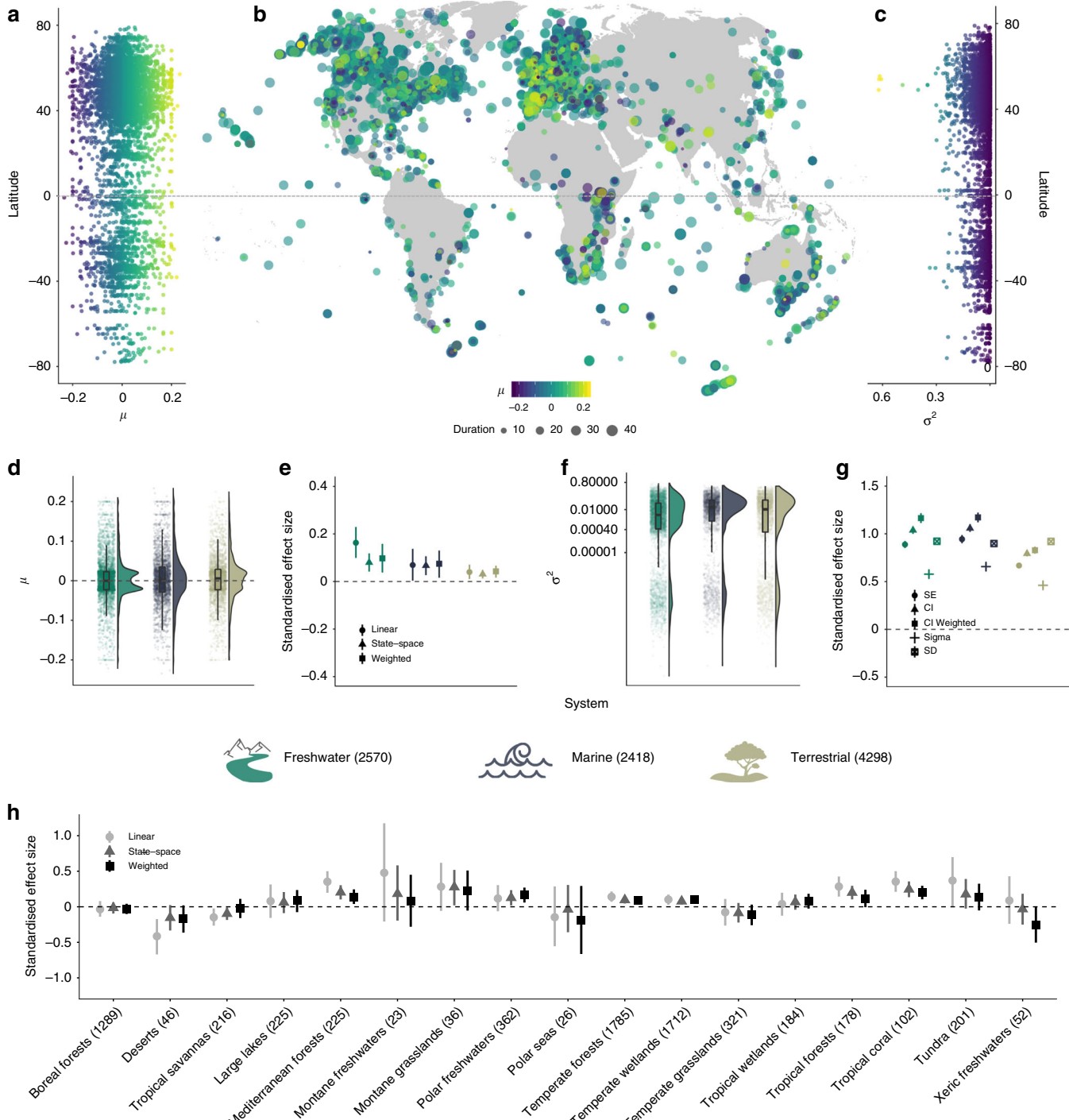

spectrum of magnitudes. While anthropogenic impacts have accelerated in recent decades, our results highlight that vertebrate species are influenced in varying ways by the number and types of threats to which they might be exposed. Against a backdrop of no biogeographic patterning of population trends and fluctuations (Fig. 1), we uncovered distinct taxonomic signals, with amphibians representing the only taxa that exhibited pronounced net declines, whereas birds, mammals and reptiles on average became more abundant over time (Fig. 2). Within amphibian, bird and reptile taxa, there was no distinct phylogenetic clustering of closely related species experiencing similar population trends or fluctuations (Fig. 2). We found that both rare and common species experienced the full spectrum of population change, from

declines to no net changes and increases. Species' geographic range, mean population size and habitat specificity did not explain variation in population trends, but species with smaller population sizes were more likely to fluctuate, potentially increasing their extinction risk (Fig. 3). There was no consistent pattern of greater population declines with increasing IUCN Red List Category (Fig. 4). On a global scale, the vertebrate species in the Living Planet Database are exposed to a variety of threats according to the species' IUCN Red List profiles, but on more local scales, none of the threats were characterised by predominantly declining populations (Fig. 4), testifying to the diverse ways in which populations are likely responding to global change across the Anthropocene.

**Fig. 1 Population declines, increases and fluctuations over time occur across all latitudes and biomes within the freshwater, marine and terrestrial realms.** Results include 9286 populations from 2084 species. The lack of biogeographic patterning in vertebrate population trends was also apparent on a UK scale (Supplementary Fig. 13 and Supplementary Table 3). The numbers in the legend for **d**–**g** and on the x axis in **c** show the sample sizes for realms and biomes, respectively. The $\mu$ values of population trend **a**, **b**, **d**, **e**, **h** and the $\sigma^2$ values of population fluctuation **c**, **f**–**g** are from state-space models of changes in abundance over time for each population. **d** and **f** show the distribution of population trends across realms including raw values (points) and boxplots (including the mean, first and third quartiles and boxplot whiskers that cover 1.5 times the interquartile range). **e**, **g** and **h** show the effect sizes (centre of error bars) and the 95% credible intervals of population trends **e**, **h** across realms and biomes, and fluctuations across realms **g**. For variation in fluctuations across biomes, see Supplementary Fig. 8. The three estimates in **e** and **h** refer to different analytical approaches: population trends calculated using linear models (circles), state-space models ($\mu$, triangles) and population trends ($\mu$) weighted by $\tau^2$, the observation error estimate from the state-space models (squares). The five estimates in **g** refer to different analytical approaches, where the response variables in the models were: (1) the standard error around the slope estimates of the linear models of abundance versus year (circles), (2) half of the 95% confidence interval around the $\mu$ value of population change (triangles), (3) half of the 95% confidence interval around $\mu$ weighted by $\tau^2$, (full squares), (4) the process noise ($\sigma^2$) from the state-space models and (5) the standard deviation of the raw data for each population time series (empty squares). The process noise is the total variance around the population trend minus the variance attributed to observation error. See Supplementary Table 2 for model outputs. Icon credits: tree by FayraLovers, wave by Setyo Ari Wibowo, mountain and stream by Nikita Kozin.

Contrary to our initial predictions, we did not find a distinct geographic patterning of population change around the world, nor a consistent trend of increasing declines in population abundance with increasing IUCN Red List Category (Figs. 1 and 4). Similar lack of biogeographic signal has been documented in regional studies of population change from the Netherlands[36] and in temperate North America and Europe[41]. Coarsely-represented biogeographic regions and global-scale species' IUCN Red List Categories and threat types might not capture the drivers acting in the locations of the specific populations we studied[34,42–44]. Furthermore, the same driver can have opposing effects on population abundance at different sites[45]. A lack of biome-specific directional trends in population change, despite a spatial clustering of human pressure around the world[12], can also arise owing to differences in species traits and vulnerability to environmental change within biomes[16,19,20]. Accounting for divergent responses of species to global change is key when translating global policy, such as the upcoming post-2020 planetary biodiversity strategy[46], into conservation actions implemented on scales much finer than biogeographic realms.

Our results highlight variation in population change among taxa, with amphibians emerging as the taxa experiencing the most pronounced declines in the Living Planet Database. The remaining taxa showed either stable or increasing net changes in abundance over time (Fig. 2). Such taxonomic patterns could be driven by different taxon-specific factors including reproductive strategy, trophic level, generation time and life history traits[47,48]. For amphibians, population declines have been linked to the spread of a fungal disease (chytrid fungus, *Batrachochytrium dendrobatidis*), facilitated by global warming[49], as well as habitat loss and Allee effects in small populations[50]. Within bird, amphibian and reptile taxa, phylogenetic relatedness and species-level taxonomic classification did not explain variation in population trends and fluctuations. A similar lack of phylogenetic dependencies has been detected for the population growth rates of migratory birds[51]. Although phylogenetic clustering might be lacking in contemporary trends, there is evidence that phylogenetic relatedness predicts extinction, a process occurring over much longer time scales[6,7]. Over shorter time periods, species' traits and ability to persist, reproduce and disperse in ever changing landscapes might be influencing local abundance[16], which has created a mix of winners and losers across all taxa[15]. We demonstrate ongoing alterations in the abundances of six vertebrate taxa, which over time, may lead to shifts in community composition and ultimately alter ecosystem function as some species become locally extinct whilst others become more abundant[9,10].

Surprisingly, our results indicate that despite decades of conservation focus on rare species[21–23], both rare and common species in the Living Planet Database experienced declines and increases in population abundance over the period of monitoring. The lack of rarity effects on population trends can be explained by theory and empirical evidence, demonstrating that small populations do not necessarily have a higher likelihood of experiencing declines and some species are able to persist in small, but stable populations[52]. The power of rarity metrics to predict population trends could also be mediated by whether species are naturally rare, or have become rare owing to external drivers in recent years[53,54]. Naturally rare species might be more likely to persist over time, whereas species that have more recently become rare might be more likely to decline in response to environmental disturbance. Furthermore, the timing and magnitude of past and current disturbance events influence population trends[45,55] and there could be temporal lags in both positive and negative abundance changes over time[45,56]. However, disentangling the processes leading to rarity over time remains challenging, and across the 2084 species we studied, there are likely cases of both natural and human-driven vertebrate population change. We found that species with small populations were, nevertheless, more likely to fluctuate (Fig. 3f), which may increase their probability of extinction, a process that could play out over longer time scales than found for most population monitoring time series to date[24,25,57]. Our results highlight that rarity metrics alone do not capture the heterogeneity in local population change over time, and common species should not be overlooked in conservation prioritisation decisions as they could be as likely to decrease in abundance over time as rare species.

Our finding that declines are not universal, or even predominant, for vertebrate populations monitored for longer than five years in the Living Planet Database contrasts with reports of an overall decline in the Living Planet Index[58], a weighted summary of population change across all abundance time series in the Living Planet Database. Consistent with our results, the Living Planet Reports[58–60] also document that the numbers of declining and increasing species are similar across this database, but the Living Planet Reports document a larger magnitude of population declines relative to increases. The calculation of the Living Planet Index involves differential weighting of population trends derived using logged abundance data, geometric means and generalised additive models, which could explain the discrepancies between our study findings and those of the Living Planet Reports[61]. The Living Planet Index is hierarchically averaged from populations to species, taxa and realm and is also weighted by the estimated and relative number of species within biomes, which influences the direction and magnitude of the

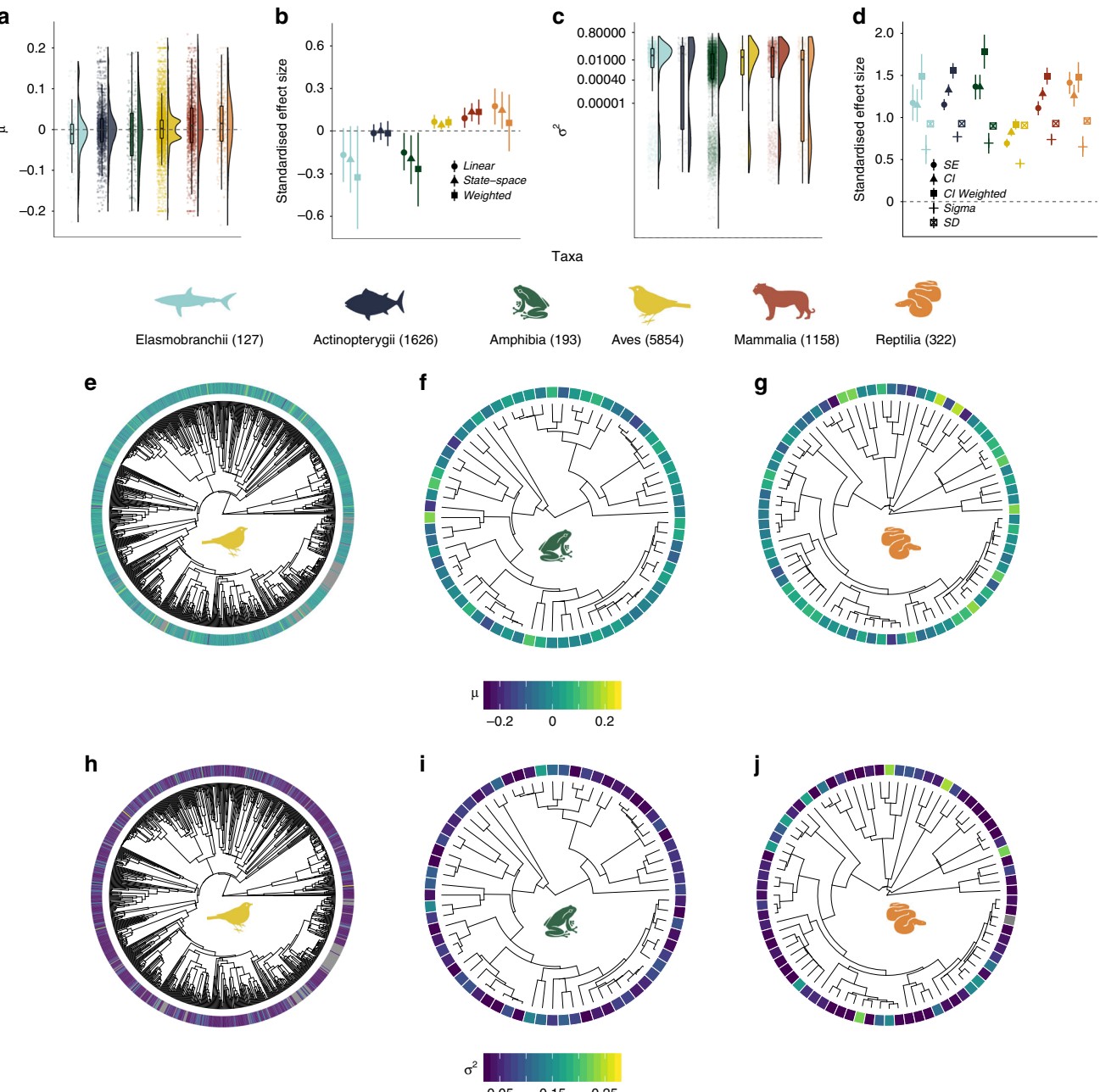

**Fig. 2 Population trends and fluctuations vary more among, rather than within, taxa, with amphibians being the only group showing pronounced declines over time.** There were no distinct phylogenetic patterns in population trends and fluctuations **e**–**j**. For details on phylogenetic models, see methods and Supplementary Fig. 15. Grey colour in the heatmap in **h** shows species for which no population trend data were available. The numbers in the legend for **a**–**d** show sample size for each taxon. The $\mu$ values of population trend **a**, **b**, **e**–**g** and the $\sigma^2$ values of population fluctuation **c**, **d**, **h**–**j** were derived from state-space model fits of changes in abundance over time for each population. **a** and **c** show the density distribution of population trends across taxa, the raw values (points) and boxplots (including the mean, first and third quartiles and boxplot whiskers that cover 1.5 times the interquartile range). **b** and **d** show the standardised effect sizes (centre of error bars) and the 95% credible intervals of population trends **b** and fluctuations **d** across the five studied taxa. Error bars in **b** and **d** show 95% credible intervals. See Fig. 1 caption for further details on effect sizes and Supplementary Tables 2 and 4 for model outputs. Icon credits: bird by Hernan D. Schlosman, snake and frog by parkjisun, fish by Julia Söderberg.

Living Planet Index[61,62]. In contrast, our analysis explores the heterogeneity in local trends and fluctuations of monitored species from the raw population abundance data, and thus, we did not use an index with weightings and we did not aggregate population trends to a species-level. Rather than summarising trends with an index, our goal was to explain variability in abundance over time across better monitored vertebrates around the world. We detected net population declines at local scales over

time only in the amphibian taxa, in contrast with the overall negative trend of the aggregate weightings of the Living Planet Index[58]. We caution that distilling the heterogeneity of local population change at sites around the world into a simple metric may hide diverging trends at local scales, where we found both increases and declines among species.

The magnitude of population trends could be influenced by how long populations are monitored[63], as well as whether

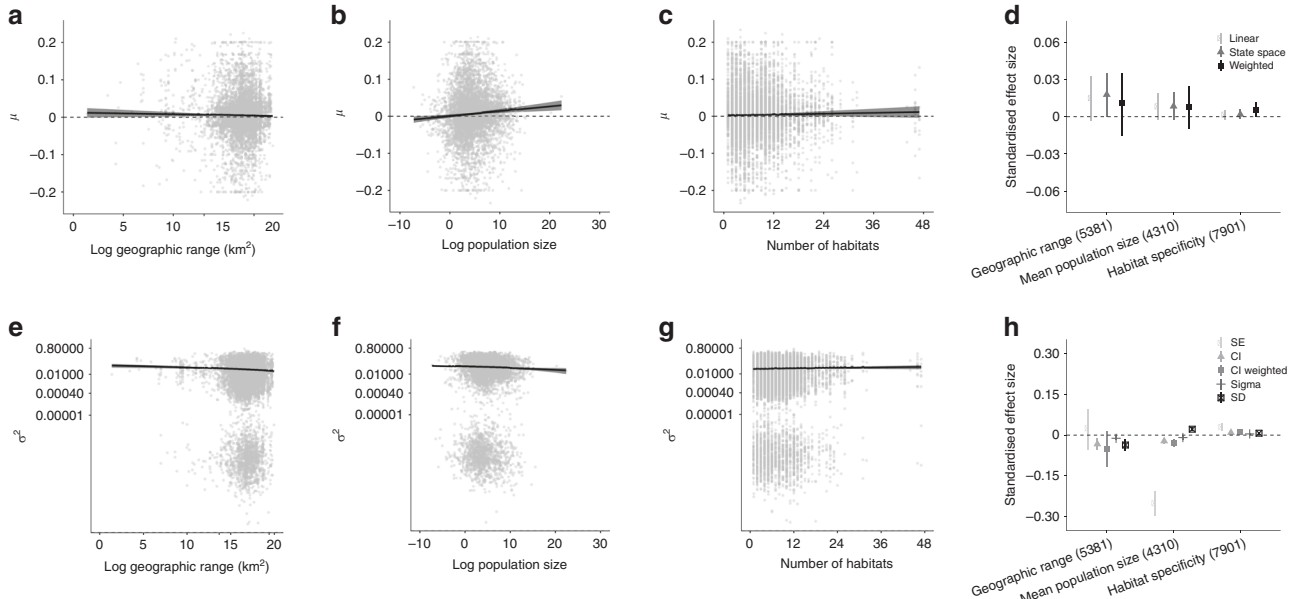

**Fig. 3 Rarity metrics do not explain heterogeneity in local population trends, and both rare and common species experienced declines and increases over time, whereas smaller populations fluctuated more over time.** Numbers on plots show sample size for each metric. Rarity metrics were calculated for all species for which information was available and cover all taxa represented in the Living Planet Database, with the exception of geographic range, which refers to the global range of only bird and mammal species in the global Living Planet Database **a–e**. The $\mu$ values of population trend **a–c** and the $\sigma^2$ values of population fluctuation **e–g** were derived from state-space model fits of changes in abundance over time for each population. **d**, **h** show the standardised effect sizes (centre of error bars) and the 95% credible intervals of three rarity metrics on population trends **d** and fluctuations **h**. Lines on **a–c** and **e–g** show model fits and 95% credible intervals. See Fig. 1 caption for further details on effect sizes, Supplementary Fig. 14 for UK-scale analyses and Supplementary Table 2 for model outputs.

monitoring began during a population peak or a population trough[64]. While overall, we did not find a strong effect of duration on the detected population trends in the Living Planet Database (Supplementary Figs. 6, 7, Supplementary Table 2), our findings demonstrate that for reptiles, time series with longer durations were more likely to capture declines (Supplementary Table 2). We also found a bimodal pattern of weak population increases and decreases in time series with longer durations particularly for terrestrial bird species with the monitoring unit being an index (Supplementary Fig. 12). Seven key challenges have been identified when drawing robust inference about population trends over time: establishment of the historical baseline, representativeness of site selection, robustness of time series trend estimation, mitigation of detection bias effects and ability to account for potential artefacts of density dependence, phenological shifts and scale-dependence in extrapolation from sample abundance to population-level inference[65]. New methods to rigorously account for different sources of uncertainty in time series monitoring will allow the analyses of available population data to better inform global estimates of net trends across taxa.

The strength of documented relationships between-population dynamics and global change could be influenced by how well-monitored populations capture the full range of variation in driver intensity. To attribute population trends and fluctuations to site-specific anthropogenic drivers, we need to go beyond previous studies that have focused exclusively on declines and extinctions[5,66]. We require attribution analyses that statistically test the links between observed changes in ecosystems and the experienced extrinsic pressure[3]. Through attribution studies that encompass the full spectrum of population change, including positive, negative and stable trends[45,67], we can better understand the variety of ways in which climate change, land-use change, and other drivers are altering global biodiversity. For a subset of the bird populations in the Living Planet Database, greater warming

of temperatures corresponded with a higher likelihood of population declines over time[67], which could be caused by worldwide and cross-biome phenological mismatches between breeding and resource availability[68]. Across terrestrial species represented in the Living Planet Database, peak forest loss was associated with accelerations in both population increases and decreases in the period following habitat alteration[45]. There is evidence from the marine realm that when species are simultaneously exposed to multiple drivers, the resulting biodiversity effects are antagonistic and could produce patterns of no net biodiversity changes[69]. The next critical step is to test how multiple global change drivers together[12] influence populations across both terrestrial and marine realms and determine how these relationships are mediated by species' traits and vulnerability to extrinsic threats[70].

In summary, our global analysis reveals the ubiquitous nature of population change over time across monitored vertebrate species. We show that in a time of accelerating global change, there were as many increases as there are decreases in population abundance over time. Among this heterogeneity, we uncovered pronounced declines in amphibian abundance as well as net abundance increases for birds, mammals and reptiles in the Living Planet Database. The taxonomic patterning of population change highlights amphibians as a conservation priority, especially as their declines can have further cascading effects across trophic levels within ecosystems. Rarity metrics, specifically geographic range, mean population size and habitat specificity, as well as IUCN Red List Categories, threat types and numbers and evolutionary history, did not explain the heterogeneity in population change across the data analysed in this study. Our findings caution the use of rarity metrics as a proxy for future global population trends, but suggest that such metrics, in particular mean population size, are nevertheless indicators of population fluctuations, which might ultimately be related to increased species extinction risk. On a global scale, both rare and common

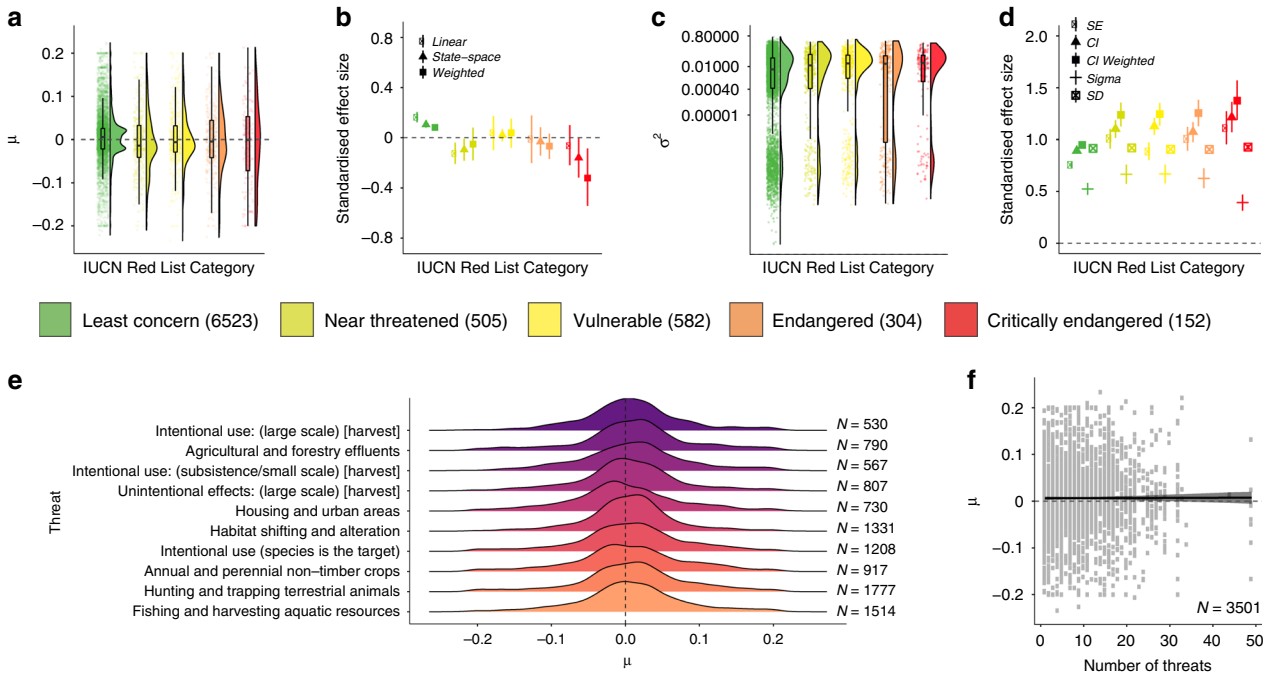

**Fig. 4 On local scales, there are increasing, decreasing and stable populations across the full spectrum of the globally-determined species' IUCN Red List Categories and anthropogenic threat type from the species' IUCN Red List profiles.** Numbers in the legend for **a–d** and in **e–f** show sample size for each metric. **a**, **c** show the density distributions of population trends across IUCN Red List Categories, the raw values (points) and boxplots with the mean, first and third quartiles and boxplot whiskers that indicate the distance that covers 1.5 times the interquartile range. **b** and **d** show the standardised effect sizes (centre of error bars) and the 95% credible intervals of population trends **b** and fluctuations **d** across IUCN Red List Categories. The $\mu$ values of population trend **a**, **e**, **f** and the $\sigma^2$ values of population fluctuation **c** were derived from state-space model fits of changes in abundance over time for each population. For the relationships between type and number of threats and population fluctuations, see Supplementary Fig. 18. **e** shows the distributions of population trends across different threats that the species face globally, with the central tendencies of all distributions overlapping with zero. Lines in **f** show model fit and 95% credible intervals, where 'number of threats' refers to the number of different threats that each species, whose populations are locally monitored, are exposed to on a global scale. See Fig. 1 caption for further details on effect sizes, Methods for details on deriving the number and types of threats and Table S2 for model outputs.

vertebrate species face numerous threats owing to resource exploitation and habitat change. As human activities continue, the next key step is to determine how intrinsic factors, such as rarity traits and threats, interact with extrinsic global change drivers and together influence the persistence of Earth's biota. Capturing the complexity of species' population dynamics will improve our estimates of shifts in community composition and of altered ecosystem functions and services around the world.

## Methods

**Workflow.** All data syntheses, visualisation and statistical analyses were conducted using R version 3.5.1[71]. For conceptual diagrams of our workflow, see Supplementary Figs. 1 and 2. Effect sizes plotted on graphs were standardised by dividing the effect size by the standard deviation of the corresponding input data.

**Population data.** To quantify vertebrate population change (trends and fluctuations), we extracted the abundance data for 9286 population time series from 2084 species from the publicly available Living Planet Database[72] (http://www.livingplanetindex.org/data_portal) that covered the period between 1970 and 2014 (Supplementary Table 1). These time series represent repeated monitoring surveys of the number of individuals in a given area, hereafter, called 'populations'. Monitoring duration differed among populations, with a mean duration of 23.9 years and a mean sampling frequency of 23.3 time points (Supplementary Fig. 3, see Supplementary Figs. 6 and 7 for effects of monitoring duration on detected trends). In the Living Planet database, 17.9% of populations were sampled annually or in rare cases multiple times per year. The time series we analysed include vertebrate species that span a large variation in age, generation times and other demographic-rate processes. For example, from other work that we have conducted, we have found that when generation time data were available (~50.0% or 484 out of 968 bird species, and 15.6% or 48 out of 306 mammal species), the mean bird generation time is 5.0 years (min = 3.4 years, max = 14.3 years) and the mean mammal generation time is 8.3 years (min = 0.3 years, max = 25 years)[45]. Thus, we

believe that most vertebrate time series within the LPD capture multiple generations.

In our analysis, we omitted populations which had less than five time points of monitoring data, as previous studies of similar population time series to the ones we analysed have found that shorter time series might not capture directional trends in abundance[63]. Populations were monitored using different metrics of abundance (e.g., population indices vs number of individuals). Before analysis, we scaled the abundance of each population to a common magnitude between zero and one to analyse within-population relationships to prevent conflating within-population relationships and between-population relationships[73]. Scaling the abundance data allowed us to explore trends among populations relative to the variation experienced across each time series.

**Phylogenetic data.** We obtained phylogenies for amphibian species from https://vertlife.org[4], for bird species from https://birdtree.org[8], and for reptile species from https://vertlife.org[6]. For each of the three classes (*Amphibia*, *Aves* and *Reptilia*), we downloaded 100 trees and randomly chose 10 for analysis (30 trees in total). Species-level phylogenies for the classes *Actinopterygii* and *Mammalia* have not yet been resolved with high confidence[74,75].

**Rarity metrics, IUCN Red List categories and threat types.** We defined rarity following a simplified version of the 'seven forms of rarity' model[76], and thus consider rarity to be the state in which species exist when they have a small geographic range, low population size, or narrow habitat specificity. We combined publicly available data from three sources: (1) population records for vertebrate species from the Living Planet Database to calculate mean population size, (2) occurrence data from the Global Biodiversity Information Facility[77] (https://www.gbif.org) and range data from BirdLife[78] (http://datazone.birdlife.org) to estimate geographic range size and (3) habitat specificity and Red List Category data for each species from the International Union for Conservation[79] (https://www.iucnredlist.org). The populations in the Living Planet Database[72] do not include species that have gone extinct on a global scale. We extracted the number and types of threats that each species is exposed to globally from their respective species' IUCN Red List profiles[79].

**Quantifying population trends and fluctuations over time**. In the first stage of our analysis, we used state-space models that model abundance (scaled to a common magnitude between zero and one) over time to calculate the amount of overall abundance change experienced by each population ($\mu$,[40,80]). State-space models account for process noise ($\sigma^2$) and observation error ($\tau^2$) and thus deliver robust estimates of population change when working with large data sets where records were collected using different approaches, such as the Living Planet Database[41,81,82]. Previous studies have found that not accounting for process noise and measurement error could lead to over-estimation of population declines[83], but in our analyses, we found that population trends derived from state-space models were similar to those derived from linear models. Positive $\mu$ values indicate population increase and negative $\mu$ values indicate population decline. State-space models partition the variance in abundance estimates into estimated process noise ($\sigma^2$) and observation or measurement error ($\tau^2$) and population trends ($\mu$):

$$X_t = X_{t-1} + \mu + \varepsilon_t, \tag{1}$$

where $X_t$ and $X_{t-1}$ are the scaled (observed) abundance estimates (between 0 and 1) in the present and past year, with process noise represented by $\varepsilon_t \sim gaussian(0, \sigma^2)$. We included measurement error following:

$$Y_t = X_t + F_t, \tag{2}$$

where $Yt$ is the estimate of the true (unobserved) population abundance with measurement error:

$$F_t \sim gaussian\left(0, T^2\right) \tag{3}$$

We substituted the estimate of population abundance ($Y_t$) into Eq. 1:

$$Y_t = X_{t-1} + \mu + \varepsilon_t + F_t. \tag{4}$$

Given

$$X_{t-1} = Y_{t-1} - F_{t-1} \tag{5}$$

then:

$$Y_t = Y_{t-1} + \mu + \varepsilon_t + F_t - F_{t-1} \tag{6}$$

For comparisons of different approaches to modelling population change, see 'Comparison of modelling approaches section'.

**Quantifying rarity metrics**. We tested how population change varied across rarity metrics—geographic range, mean population size and habitat specificity – on two different but complementary scales. In the main text, we presented the results of our global-scale analyses, whereas in the SI, we included the results when using only populations from the UK—a country with high monitoring intensity, Thus, we quantified rarity metrics for species monitoring globally and in the UK. The three rarity metrics used in this study were weakly correlated at both UK and global scales (Supplementary Fig. 11).

**Geographic range**. To estimate geographic range for bird species monitored globally, we extracted the area of occurrence in km$^2$ for all bird species in the Living Planet Database that had records in the BirdLife Data Zone[78]. For mammal species' geographic range, we used the PanTHERIA database[84] (http://esapubs.org/archive/ecol/E090/184/). To estimate geographic range for bony fish, birds, amphibians, mammals and reptiles monitored in the UK (see Supplementary Table 5 for species list), we calculated a km$^2$ occurrence area based on species occurrence data from GBIF[77]. Extracting and filtering GBIF data and calculating range was computationally intensive and occurrence data availability was lower for certain species. Thus, we did not estimate geographic range from GBIF data for all species part of the Living Planet Database. Instead, we focused on analysing range effects for birds and mammals globally, as they are a very well-studied taxon and for species monitored in the UK, a country with intensive and detailed biodiversity monitoring of vertebrate species. We did not use IUCN range maps, as they were not available for all of our study species, and previous studies using GBIF occurrences to estimate range have found a positive correlation between GBIF-derived and IUCN-derived geographic ranges[85].

For the geographic ranges of species monitored in the UK, we calculated range extent using a minimal convex hull approach based on GBIF occurrence data[77]. We filtered the GBIF data to remove invalid records and outliers using the *CoordinateCleaner* package[86]. We excluded records with no decimal places in the decimal latitude or longitude values, with equal latitude or longitude, within a one-degree radius of the GBIF headquarters in Copenhagen, within 0.0001 degrees of various biodiversity institutions and within 0.1 degrees of capital cities. For each species, we excluded the lower 0.02 and upper 0.98 quantile intervals of the latitude and longitude records to account for outlier points that are records from zoos or other non-wild populations. We drew a convex hull to most parsimoniously encompass all remaining occurrence records using the *chull* function, and we calculated the area of the resulting polygon using *areaPolygon* from the *geosphere* package[87].

**Mean size of monitored populations**. We calculated mean size of the monitored population, referred to as population size, across the monitoring duration using the

raw abundance data, and we excluded populations, which were not monitored using population counts (i.e., we excluded indexes).

**Habitat specificity**. To create an index of habitat specificity, we extracted the number of distinct habitats a species occupies based on the IUCN habitat category for each species' profile, accessed through the package *rredlist*[88]. We also quantified habitat specificity by surveying the number of breeding and non-breeding habitats for each species' online IUCN species profile (the 'habitat and ecology' section). The two approaches yielded similar results (Supplementary Fig. 10, Supplementary Table 2, key for the profiling method is presented in Supplementary Table 6). We obtained global IUCN Red List Categories and threat types for all study species through their IUCN Red List profiles[79].

**Testing the sources of variation in population trends and fluctuations**. In the second stage of our analyses, we modelled the trend and fluctuation estimates from the first stage analyses across latitude, realm, biome, taxa, rarity metrics, phylogenetic relatedness, species' IUCN Red List Categories and threat type using a Bayesian modelling framework through the package *MCMCglmm*[89]. We included a species random intercept effect in the Bayesian models to account for the possible correlation between the trends of populations from the same species (see Supplementary Table 1 for sample sizes). The models ran for 120,000 iterations with a thinning factor of ten, a burn-in period of 20,000 iterations and the default one chain. We assessed model convergence by visually examining trace plots. We used weakly informative priors for all coefficients (an inverse Wishart prior for the variances and a normal prior for the fixed effects):

$$Pr(\mu) \sim N\left(0, 10^8\right) \tag{7}$$

$$Pr(\sigma^2) \sim Inverse\ Wishart\left(V = 0, nu = 0\right) \tag{8}$$

**Population trends and fluctuations across latitude, biomes, realms and taxa**. To investigate the geographic and taxonomic patterns of population trends and fluctuations, we modelled population trends ($\mu$) and population fluctuations ($\sigma^2$), derived from the first stage of our analyses (state-space models), as a function of (1) latitude, (2) realm (freshwater, marine, terrestrial), (3) biome (as defined by the 'biome' category in the Living Planet Database, e.g., 'temperate broadleaf forest'[90] and (4) taxa (*Actinopterygii*, bony fish; *Elasmobranchii*, sharks and rays; *Amphibia*, amphibians; *Aves*, birds; *Mammalia*, mammals; *Reptilia*, reptiles). We used separate models for each variable, resulting in four models testing the sources of variation in trends and four additional models focusing on fluctuations. Each categorical model from this second stage of our analyses was fitted with a zero intercept to allow us to determine whether net population trends differed from zero for each of the categories under investigation. The model structures for all models with a categorical fixed effect were identical with the exception of the identity of the fixed effect, and below we describe the taxa model:

$$\mu_{i,j,k} = \beta_0 + \beta_{0j} + \beta_1 * taxa_{i,j,k}, \tag{9}$$

$$y_{i,j,k} \sim gaussian\left(\mu_{i,j,k}, \sigma^2\right), \tag{10}$$

where $taxa_{i,j,k}$ is the taxa of the *i*th time series from the *j*th species; $\beta_0$ and $\beta_1$ are the global intercept (in categorical models, $\beta_0 = 1$) and the slope estimate for the categorical taxa effect (fixed effect), $\beta_{0j}$ is the species-level departure from $\beta_0$ (species-level random effect); $y_{i,j,k}$ is the estimate for change in population abundance for the *i*th population time series from the *j*th species from the *k*th taxa.

**Population trends and fluctuations across amphibian, bird and reptile phylogenies**. To determine whether there is a phylogenetic signal in the patterning of population change within amphibian, bird and reptile taxa, we modelled population trends ($\mu$) and fluctuations ($\sigma^2$) across phylogenetic and species-level taxonomic relatedness. We conducted one model per taxa per population change variable—trends or fluctuations using Bayesian linear mixed effects models using the package *MCMCglmm*[89]. We included phylogeny and taxa as random effects. The models did not include fixed effects. We assessed the magnitude of the random effects (phylogeny and species) by inspecting their posterior distributions, with a distribution pushed up against zero indicating lack of effect, as these distributions are always bounded by zero and have only positive values. We used parameter-expanded priors, with a variance-covariance structure that allows the slopes of population trend (the $\mu$ values from the first stage analysis using state-space models) to covary for each random effect. The prior and model structure were as follows:

$$Pr(\mu) \sim N\left(0, 10^8\right), \tag{11}$$

$$Pr(\sigma^2) \sim Inverse\ Wishart\left(V = 1, nu = 1\right), \tag{12}$$

$$\mu_{i,k,m} = \beta_0 + \beta_{0k} + \beta_{0m}, \tag{13}$$

$$y_{i,k,m} \sim gaussian\left(\mu_{i,k,m}, \sigma^2\right), \tag{14}$$

where $\beta_0$ is the global intercept ($\beta_0 = 1$), $\beta_{0l}$ is the phylogeny-level departure from $\beta_0$ (phylogeny random effect); $y_{i,k,m}$ is the estimate for change in population abundance for the *ith* population time series for the *kth* species with the *mth* phylogenetic distance.

To account for phylogenetic uncertainty for each class, we ran ten models with identical structures, but based on different randomly selected phylogenetic trees. We report the mean estimate and its range for each class.

**Population trends and fluctuations across rarity metrics**. To test the influence of rarity metrics (geographic range, mean population size and habitat specificity) on variation in population trends and fluctuations, we modelled population trends ($\mu$) and fluctuations ($\sigma^2$) across all rarity metrics. We conducted one Bayesian linear model per rarity metric per scale (for both global and UK analyses) per population change variable—trends or fluctuations. The response variable was population trend ($\mu$ values from state-space models) or population fluctuation ($\sigma^2$ values from state-space models), and the fixed effects were geographic range (log transformed), mean population size (log transformed) and habitat specificity (number of distinct habitats occupied). The model structures were identical across the different rarity metrics and below we outline the equations for population trends and geographic range:

$$\mu_{i,k,n} = \beta_0 + \beta_{0k} + \beta_1 * geographic\ range_{i,k,n}, \tag{15}$$

$$y_{i,k,n} \sim gaussian\left(\mu_{i,k,n}, \sigma^2\right), \tag{16}$$

where *geographic range*$_{i,k,n}$ is the logged geographic range of the *kth* species in the *ith* time series; $\beta_0$ and $\beta_1$ are the global intercept and slope estimate for the geographic range effect (fixed effect), $\beta_{0j}$ is the species-level departure from $\beta_0$ (species-level random effect); $y_{i,k,n}$ is the estimate for change in population abundance for the *ith* population time series from the *jth* species with the *nth* geographic range.

**Population trends across species' IUCN Red List Categories**. To investigate the relationship between-population change and species' Red List Categories, we modelled population trends ($\mu$) and fluctuations ($\sigma^2$) as a function of IUCN Red List Categories (categorical variable). We conducted one Bayesian linear model per population change metric per scale (for both global and UK analyses). To test variation in population trends and fluctuations across the types and number of threats to which species are exposed, we conducted a post hoc analysis of trends and fluctuations across threat type (categorical effect) and number of threats that each species is exposed to across its range (in separate models). The model structures were identical to those presented above, except for the fixed effect which was a categorical IUCN Red List Category variable.

The analytical workflow of our analyses is summarised in conceptual diagrams (Supplementary Figs. 1 and 2) and all code is available on GitHub (https://github.com/gndaskalova/PopChangeRarity, DOI 10.5281/zenodo.3817207).

**Data limitations: taxonomic and geographic gaps**. Our analysis is based on 9286 monitored populations from 2084 species from the largest currently available public database of population time series, the Living Planet Database[72]. Nevertheless, the data are characterised by both taxonomic and geographic gaps that can influence our findings. For example, there are very few population records from the Amazon and Siberia (Fig. 1b)—two regions currently undergoing rapid environmental changes owing to land-use change and climate change, respectively. In addition, birds represent 63% of all population time series in the Living Planet Database, whilst taxa such as amphibians and sharks where we find declines are included with fewer records (Fig. 2 and Supplementary Fig. 4). On a larger scale, the Living Planet Database under-represents populations outside of Europe and North America and over-represents common and well-studied species[62]. We found that for the populations and species represented by current monitoring, rarity does not explain variation in population trends, but we note that the relationship between population change and rarity metrics could differ for highly endemic specialist species or species different to the ones included in the Living Planet Database[17]. As ongoing and future monitoring begins to fill in the taxonomic and geographic gaps in existing datasets, we will be able to reassess and test the generality of the patterns of population change across biomes, taxa, phylogenies, species traits and threats.

**Data limitations: monitoring extent and survey techniques**. The Living Planet Database combines population time series where survey methods were consistent within time series but varied among time series. Thus, among populations, abundance was measured using different units and over varying spatial extents. There are no estimates of error around the raw population abundance values available and detection probability likely varies among species. Thus, it is challenging to make informed decisions about baseline uncertainty in abundance estimates without prior information. We used state-space models to estimate trends and fluctuations to account for these limitations as this modelling framework is particularly appropriate for analyses of data collected using disparate methods[41,81,82]. Another approach to partially account for observer error that has

been applied to the analysis of population trends is the use of occupancy models[36]. Because the precise coordinates of the polygons where the individual populations were monitored are not available, we were not able to test for the potential confounding effect of monitoring extent, but our sensitivity analysis indicated that survey units do not explain variation in the detected trends (Supplementary Fig. 12).

**Data limitations: temporal gaps**. The population time series we studied cover the period between 1970 and 2014, with both duration of monitoring and the frequency of surveys varying across series. We omitted populations that had less than five time points of monitoring data, as previous studies of similar population time series data have found that shorter time series are less likely to capture directional trends in abundance[63]. In a separate analysis, we found significant lags in population change following disturbances (forest loss) and that population monitoring often begins decades to centuries after peak forest loss has occurred at a given site[45]. The findings of this related study suggest that the temporal span of the population monitoring does not always capture the period of intense environmental change and lags suggest that there might be abundance changes that have not yet manifested themselves. Thus, the detected trends and the baseline across which trends are compared might be influenced by when monitoring takes place and at what temporal frequency. Challenges of analysing time series data are present across not just the Living Planet Database that we analysed, but more broadly across population data in general, including invertebrate datasets[65]. Nevertheless, the Living Planet Database represents the most comprehensive compilation of vertebrate temporal population records to date, allowing for analyses of the patterns of vertebrate trends and fluctuations around the world.

**Data limitations: time series with low variation**. Eighty populations (<1% of the 9286 time series) had very little variance (see Supplementary Table 7 for full references for those studies). The majority of those studies are for bird species and come from the North American breeding bird survey with a measurement unit of an index[91]. We have also observed some time series that appear to show logistic relationships with little natural variance (e.g., time series 468, 10193, 17803, see Supplementary Table 8 for full references). Inspecting the raw data showed that some populations have abundances which follow an almost perfect linear or logarithmic increase over time, as could be the case for modelled, versus raw field data. We provide the references for these studies and cannot definitely attribute the low variance to a particular cause across all studies. Some of these studies are reported in units that are an index which may not capture variation in the same way as other raw units of population data. Some of these time series may represent modelled population data based on demographic information rather than only direct observations of populations (e.g., time series 1355[92]). We chose to not remove studies that may not be raw observation time series based on visual inspection of trends to avoid introducing bias against populations with naturally low variation into our analysis.

**Clustering in the values of population trends and fluctuations**. We found a clustering of population trend and fluctuation values in some parts of the population change spectrum. For example, we found two peaks—in small increases and in small decreases over time—which were most prevalent in terrestrial bird studies and species, monitored using an index (Fig. 2, Supplementary Fig. 12). Overall 11.4% of time series had trend values between 0.02 and 0.03 and 11.6% of time series had trend values between −0.03 and −0.02. There was also a similar, but smaller, clustering around trends of 0.25 and −0.25. All reported population trends are from models that converged successfully, and visual inspection indicated to us that the $\mu$ values are appropriate estimates for the individual time series (Supplementary Fig. 6e). We investigated the population time series where the value of the population trends over time were estimated to be the same value and found that they came from a variety of taxa, locations and survey methods (Supplementary Fig. 6e). We hypothesise that there might be a publication bias against publishing no net change studies, which could explain the trough in $\mu$ values of around zero in long-term studies. The clustering of values for some time series may sometimes be associated with the same time series that also have low variance (Supplementary Fig. 6e, see discussion above). With the information available in the Living Planet Database metadata, we cannot fully explain the clustering in population trends. We advocate for more detailed metadata in future versions of the Living Planet Database to allow researchers to filter the database appropriately for individual analyses.

**Challenges in estimating geographic range**. Estimating geographic range across taxa, and specifically for species that are not birds or mammals, remains challenging owing to data limitations. We used a static measure of geographic range, which does not account for changes in species distributions over time. Furthermore, species could naturally have a small range or the small range size could be due to historic habitat loss[93]. The UK populations included in the Living Planet Database are predominantly from species with wide geographic ranges (Supplementary Table 5), and our global-scale analysis of the relationship between-population change and geographic range is based on mammal and bird data. As

data availability improves, future research will allow us to test the effect of geographic range on the trends of other taxa, such as amphibians and sharks.

**Trends relative to null expectation.** We tested whether the number of increasing and decreasing populations trends differed from a null expectation using a data randomisation approach (Supplementary Fig. 5b). We used linear models to estimate trends in the data and randomised data with identical structure to the Living Planet Database. We found that there were over 10 times more population declines and increases in the real data relative to the randomised data (2.29% of trends were declining and 2.30% were increasing in the randomised data, versus 28.9% and 32.5% of time series, which had significant negative and positive slopes in the real data, respectively).

**Monitoring duration, sampling methods and site-selection bias.** To assess the influence of monitoring duration on population trends, we used a Bayesian linear model. We modelled population trend ($\mu$) as a function of monitoring duration (years) for each population, fitted with a zero intercept, as when duration is zero, no population change has occurred. Monitoring duration was weakly positively related to vertebrate population trends, with slightly greater population increases found for longer duration studies (Supplementary Fig. 6, Supplementary Table 2). There was a similar weakly positive effect of number of time points within time series (Supplementary Table 2). In addition, we tested if monitoring duration influenced the relationships between population trends across systems, and population trends across taxa. We found that duration did not influence those relationships, with the exception of reptiles, where declines were more frequent as monitoring duration increased (Supplementary Table 2). Variation in population trends was not explained by sampling method across the five most commonly used abundance metrics (population index, number of individuals, number of pairs, number of nests and population estimate, Supplementary Fig. 12). Following Fournier et al.[64], we tested the time series that we analysed for site-selection bias. Removing the first five survey points reduces the bias stemming from starting population surveys at points when individual density is high, whereas removing the last five years reduces the bias of starting surveys when species are very rare. The distribution of population trend values across time series was not sensitive to the omission of the first five (left-truncation) or the last five years (right-truncation) of population records (Supplementary Fig. 5a). In addition, we used a data randomisation approach to compare the distribution of trends from the real data to a null distribution and found different patterns (Supplementary Fig. 5b). Overall, our sensitivity analyses suggest that our findings are robust to the potential confounding effects of differences in monitoring duration, sampling method and site-selection.

**Comparison of modelling approaches.** We conducted the following supplementary analyses: in the second-stage Bayesian models estimating population trends across systems, biomes, taxa and rarity metrics, (1) we weighed $\mu$ values by the square of $\tau^2$, the observation error estimate derived from the state-space models[40], (2) we used slopes of linear model fits of abundance (scaled at the population level, centred on zero and with a standard deviation of one)[73] instead of the $\mu$ estimates from state-space models, (3) we modelled the standard error around the slope values of the linear models, the error around $\mu$ (half of the 95% confidence interval) and the standard deviation of the raw population data for each time series as additional metrics of population variability. To allow comparison, we scaled the different metrics of population variability to be centred on zero and with a standard deviation of one before they were used as response variables in models. All different analytical approaches yielded very similar results (see main text and Supplementary Figs. 5, 6 and 16, Supplementary Table 2).

**Reporting summary.** Further information on research design is available in the Nature Research Reporting Summary linked to this article.

## Data availability
Raw data are available from the following websites: for population time series[72]—http://www.livingplanetindex.org/data_portal, GBIF occurrences[77]—https://www.gbif.org, bird geographic ranges[78]—http://datazone.birdlife.org, mammal geographic ranges[84]—http://esapubs.org/archive/ecol/E090/184/, species' habitat preferences, threat types and IUCN Red List Categories[79]—https://www.iucnredlist.org, and phylogenies[6–8]—https://vertlife.org and https://birdtree.org.

## Code availability
Code for all data processing and analyses and summary data sets are publicly available on GitHub and archived on Zenodo (https://doi.org/10.5281/zenodo.3817207)[94].

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

## Acknowledgements

We thank the teams behind the open-source databases we used for our study (Living Planet Team, GBIF, BirdLife, PanTHERIA, VertLife and BirdTree. We are grateful to the WWF and the RZSL for compiling and maintaining the Living Planet Database, specifically Louise McRae and Robin Freeman, and to Dmitry Schigel and Jan Legind at GBIF for providing access to a direct download link for the occurrence data. We thank the Critical Thinking in Ecology and Environmental Sciences 2015-16 tutorial group at the University of Edinburgh, including Hannah Louise Stevens, Ashlie Nithsdale, Lisa Kopsieker and Claudia Ardrey who helped us conceive the idea for the analysis. We thank Dan Greenberg for advice on state-space models and Kat Keogan and Albert Phillimore for advice on phylogenetic models. We thank Anne Magurran, Maria Dornelas, Albert Phillimore and Mark Vellend for providing very helpful feedback on the manuscript and analyses. We thank our reviewers for providing very constructive feedback improving our study. Funding for this research was provided by the NERC doctoral training partnership grant (NE/L002558/1) at the University of Edinburgh and the Carnegie Trust for the Universities of Scotland.

## Author contributions

G.N.D. and I.M.S. conceived the idea and conducted the statistical analyses. J.L.G. contributed to the calculation of geographic range estimates. All authors contributed to the integration of the LPI, GBIF and IUCN databases, which G.N.D. led. G.N.D. created all figures with input from I.M.S. G.N.D. wrote the first draught of the manuscript and all authors contributed to revisions. I.M.S. supervised the research as a senior author.

## Competing interests

The authors declare no competing interests.
