## [Peer Review File · Nature Communications]

Reviewers' Comments:

Reviewer #1:

Remarks to the Author:

In general, I thought this was an interesting paper - even though it's essentially a non-result (rarity doesn't affect trends). I was specifically asked to comment on the methods and statistics, so will try to focus my critiques there. I think the use of state-space models is appropriate, though throughout I was a bit disappointed in the lack of details and specifics (equations, citations). My 2 biggest criticisms of this are: 1. Interpreting uncertainty around μ as a metric of variability is incorrect, and 2. not providing reviewers code / data during the review is your responsibility -- a link to a public Github repo is really necessary

Methods:

Line 381: Please comment on, or provide more details about this process. Additionally, please comment on whether or not this scaling affects inference (or biases inference) about the trend and variance parameters.

For example, standardizing the data to be \sim Normal (0, 1) before fitting these state space models affects estimates of the observation error, but not the trend or process variance. Scaling the data to be in the interval (0, 1) affects estimates of all three parameters: process and observation variances, as well as the trends.

Line 389-444: compiling all these data seem like a big task. Are they included in the supplementary info? It might be helpful for future researchers if you included them

Line 448: I'm confused by the term 'freely-varying intercept'? I think writing out the equations, and citing software you used - or providing code - is necessary here

Line 455-457: This decision to use the uncertainty metric around μ is wrong. Uncertainty around μ represents uncertainty in the long term trend, not variability. Uncertainty around μ is influenced by a number of things - including observation error, missing data, etc. The process variance represents the variability around each population trend and is a better metric for variability. Please change.

Line 466-467: I'm confused about the 2-stage modeling. In a Bayesian world, μ is a distribution, not a point estimate. How did you carry through the uncertainty in μ into this 2nd stage? As before, please include some equations and citations for the software you used -- or code.

Line 471: Don't refer to predictors as 'significant' if you're using Bayesian methods

Line 497: Again, equations here, and more specifics (like the levels of each of the random effects is essential)

Line 512: You can't say "We used parameter expanded priors..." without giving any specifics or citations and expect the reader to follow. Maybe including a table of parameters and priors would be helpful

Line 534-541: I think including some citations for each of these metrics would be helpful.

Line 547: I find this kind of statement pretty difficult to reconcile as a reviewer. You should have included the code and link to the github repository -- with all data used in the paper -- in your initial

submission

"To review the code for our statistical analyses prior to publication, please contact the authors. Population time-series data are publicly available "

Reviewer #2:

Remarks to the Author:

The reams of tests and data presented are impressive at first glance, and I appreciate that the authors wish to establish a novelty that has been previously unappreciated. However, they fail to do this for both conceptual, data, and methodological reasons.

I have several major concerns with the data applied. First, despite applying a state-space model to their time series, most of the Living Planet Database does not provide measurement error for their time-specific estimates of abundance. This is not the authors fault, but it is a major issue with population-dynamical time series in general. As such, a state-space (or any other, for that matter) approach will not remove the huge probability of making substantial Type II errors in any hypothesis test. This is made even more serious and pertinent because the authors attempt to tell us something useful for rare species in particular, when the measurement error for these is likely to much, much higher.

Thus, even the relatively interesting result that small-population species tend to have higher temporal variance (but previously established much more comprehensively by others, such as in Melbourne & Hastings 2008 *Nature* 454:100 and Fagan et al. *Ecol Lett* 9:51 — the latter is a classic that the authors did not even cite) is rendered highly suspect because of the likely inflation of measurement error for these species.

The authors also make the naïve mistake of failing to distinguish the small- and declining-population paradigms (Caughley 1994 *J Anim Ecol* 63:215) in their analyses. The processes driving declines ARE NOT, nearly ever, the same that drive small populations to extinction. Thus, searching for relationships between rarity and decline is something of a waste of time, merely because we know that the processes differ between the two categories (separated by the concept of the minimum viable population size).

Another major concern is that 'extinction risk' is being inferred constantly throughout this manuscript from 'threat risk'. These are not the same thing. First, the IUCN Red List uses multiple criteria to set relative threat risk (not extinction risk), yet the authors do not distinguish among Criteria when testing their relationships. There is really only one Criterion here that is relevant - Criterion A justifications (population decline), yet the authors use all Criteria to establish 'extinction risk'. This mixes different drivers and will necessarily dilute the capacity to identify relationships. It is therefore not surprising in the slightest that the authors did not find the relationships they purport to test hypothetically.

For an example of one of the few articles I know that has tested drivers of 'extinction risk' vs. 'threat risk', Sodhi et al. (2008 *Div Disturb* 14:1; again, that the authors did not cite). Somewhat interestingly, the latter in fact supports the authors' contention that extinction risk is difficult to predict, but the authors here are in no way testing 'extinction risk'!

Finally, the authors take a decidedly narrow view of extinction of life, when in fact there are plenty of examples from palaeontological work that establishes rather convincingly the very patterns these

authors say do not exist. In fact, one of the best examples of how rarity and geographic size affect actual extinction risk is palaeo work across 1/2 billion years. Taking the modern snapshot to define generalities is therefore suspect because the paleo record is clear here:
doi:10.1098/rspb.2012.1902.

Some more specific comments follow:

- L78-79: This is not entirely true. If we use Red List status as 'threat risk', then there are many missing references and taxa showing strong effects of geographic range — including:
doi:10.1016/S0065-2881(09)56004-X (fish), doi:10.1371/journal.pone.0001636 (amphibians),
doi:10.1111/j.1365-2745.2008.01408.x (legumes) — and these are just some examples.
- L93: Another important study that the authors are missing, but that I think supports the contention is for British birds, is where local climate is the dominant driver of expansion or contraction of range, with contributions of life-history pace and geographical context: doi:10.1098/rspb.2014.0744 (i.e., drivers of change are complicated, and locally dependent).
- L204-205: I contend that this result arises merely because of the lower expected measurement error associated with the better-monitored terrestrial species
- L270-272: Some of these results could also be partially explained by the observation that different regions are experiencing different phases of the extinction process. For example, regions such as Europe have experienced massive declines in species for many millennia, resulting in a depauperate fauna that has adapted to the human-dominated landscape. Elsewhere in only recently disturbed areas, the most sensitive species are only now starting to dwindle. Thus, you cannot compare regions simply by species lists — you have to take time since disturbance into account, or the conclusions are not valid (apples and oranges, as we say).
- L288-289: Completely unsupported speculation
- L380: This is an arbitrary value with no justification. Some sensitivity analysis is required at the very least.

Reviewer #3:

Remarks to the Author:

This paper explores patterns of population change in vertebrates, using data from the Living Planet Index Database (LPD). It tests the hypothesis that species rarity and conservation status predict population declines and fluctuations.

Exploring these questions is of obvious interest to applied nature conservation perspective, but they also address the relationship population dynamics and species status, which fundamentally underpins macroecology and dynamic biogeography. Therefore, I expect the results of this analysis, which are based on an impressively large dataset, to generate a high level of interest among a broad readership. Overall I was impressed by the scope of the study and am sympathetic to its aims. However, I have a number of serious reservations about the conclusions, about potential biases in the analysis and about a lack of detail in the methods.

Major comments

1. This is an important analysis utilising LPD data in a novel way. The abstract and title are concise

and clear, but the manuscript itself doesn't provide the same kind of synthesis – it's a list of results. For me the big conclusion is not so much that rarity metrics are unimportant, but that variation within categories is greater than among. In addition, the study reported no net population declines (except for amphibians) which stands in contrast to the LPI report which found a 58% decline in species abundance. Seeing as both analyses use the same database, this should definitely be discussed.

2. Sceptics might argue that the selected populations don't capture the full range of threats experienced by species. You can only measure population trends on extant populations whereas habitat loss completely removes populations. I would hazard that most species are listed under Criterion A not because populations have been monitored, but because habitat is perceived to have been lost. For this reason, the lack of correlation between IUCN rating and population trend may not be surprising.

3. Use of 'population size' is confusing and potentially wrong. I think here it means 'the number of animals that were measured by monitoring'. This is different from the ecological concept of a population, which is based on being a relatively close demographic unit. The relationship between these two numbers is likely to vary systematically with body size and mobility. For example, surveying a 1km² may capture an entire ecological population of a small, immobile species, but will only provide a subsample for larger species with greater dispersal distances.

4. The results are not interpretable without having read the methods. This is a structural issue that needs to be addressed. It is not clear what the slopes are that are reported in the results. Moreover, the Methods are lacking in key details. For example, the reader isn't told whether analyses were done at species or population level. This lack of detail makes it difficult to understand results, and hindering reproducibility. For example, what does 'scaling' abundance mean (Line 381-382)?

Minor comments

- Introduction: Try some other words in place of However, Yet, and Thus.

- Line 43: Fundamental (not fundament)

- Line 62: "However, although..." is just bad English

- line 65: the definition of "population change" is not clear at this point. Readers could understand this as synonymous with trend, or as something broader. Trend I think is clear: a directional change in population size, but change is more vague and ambiguous. Conceivably "trend" could be interpreted as special case of "change", along with "fluctuations" (which is introduced later): other types of change might include a change in the frequency of population cycles, or a directional trend in interannual variance in numbers¹ (independent from a trend in numbers). Similarly, they use both "characteristics" and "metrics" when referring to the trio of geographic range size, habitat specialism and population size.

- Lines 65-81: This is a critical paragraph that is potentially misleading to readers. The paragraph is framed in terms of population change, but all the examples are about traits correlated with extinction risk (i.e. a species-level metric, not a population metric). This is confusing. Also, whilst the Rabinovitz paper is a nice way to introduce the concepts, I'm not convinced that "rarity" is a useful theme around which to pin these analysis, for two reasons. First, species can be listed as at risk of extinction due to a rapid decline (criterion A) without meeting any of the seven definitions of rarity, so it's not clear that one would expect any kind of "generalizable relationship" (line 78). Second, my interpretation is that "rarity" is an outdated concept that was killed-off by Rabinovitz. If you're interested in the traits associated with change at the population level then there is already a literature¹⁻⁴ that is more relevant than the citations on correlates of extinction risk.

- Fig 1 legend does not stand alone

- Fig 2 & 3: Clarify whether these are credible intervals on the mean estimate, or across all species?

- Line 51: Why is extinction risk defined here when it is being discussed before (eg Line 41)

- Line 86: Mentions previous studies but doesn't provide reference

- Line 199: There is a literature on Taylor's power law that defines the expected relationship between population size and the magnitude of fluctuations. Please place this result (and the methods used to

test it) within that literature.

- Line 262: "Thus, although..." is grammatically okay but doesn't read well
- Line 376: indicate whether this is ALL the LPD data
- Line 441: Mentions a software package but it is not clear that R is being used until line 543
- Line 446: Why is the model not presented? This text is not enough to reproduce.

REFERENCES

1. Oliver, T. H., Roy, D. B., Brereton, T. & Thomas, J. A. Reduced variability in range-edge butterfly populations over three decades of climate warming. *Glob. Chang. Biol.* 18, 1531–1539 (2012).
2. Mace, G. M., Collen, B., Fuller, R. A. & Boakes, E. H. Population and geographic range dynamics: Implications for conservation planning. *Philos. Trans. R. Soc. B Biol. Sci.* 365, 3743–3751 (2010).
3. Cowlshaw, G., Pettifor, R. A. & Isaac, N. J. B. High variability in patterns of population decline: the importance of local processes in species extinctions. *Proc. Biol. Sci.* 276, 63–9 (2009).
4. Isaac, N. J. B. & Cowlshaw, G. How species respond to multiple extinction threats. *Proc. Biol. Sci.* 271, 1135–41 (2004).

Response to reviewers

All is not decline across global vertebrate populations (NCOMMS-18-28274-T)

Gergana N. Daskalova, Isla H. Myers-Smith and John L. Godlee

We thank the reviewers for providing constructive feedback. We have fully revised our manuscript and have addressed all of the reviewers' comments, as well as added new analyses to further strengthen our work. We appreciate the positive comments highlighting the contributions of our study.

In our revisions, we paid specific attention to 1) better communicating the most novel and important elements to our findings, 2) enhancing the breadth and depth of the discussion, 3) addressing how the types of threats to which species are exposed could influence detected population change (new analyses), and 4) clarifying the methods by adding more details, equations, as well as all code and data for the analyses to enhance reproducibility.

The major revisions and new analyses we have undertaken are summarised below and discussed in detail in the point-by-point responses.

Revision 1. Fully revised main text, including a streamlined introduction, where we focus explicitly on population change, and a discussion with more synthesis and interpretation of our findings.

Revision 2. New paragraph about our methods and statistical approach in the introduction to improve the readability and comprehension of our findings.

Revision 3. New paragraph discussing the reasons behind the differences between our study and similar analyses of population time-series and the Living Planet Index.

Revision 4. Additional content discussing the limitations of the Living Planet Database and how they might influence our findings.

Revision 5. Further details in the methods, including equations, code, references, conceptual diagrams in the supplementary information and more detailed figure captions that can stand alone. All code and data for our analyses are available on GitHub for review. To access the private repository, go to <https://github.com>, log in using username *PopChangeRarity* and password *atimeofchange9* and then navigate to <https://github.com/gndaskalova/PopChangeRarity>.

Revision 6. New analyses of population fluctuations across realms, biomes, latitude, taxa, phylogenetic relatedness, rarity metrics, IUCN conservation status and threats using the process error (sigma from the state-space models) as a metric of population fluctuation.

Revision 7. New analyses testing population trends and fluctuations across species' threats (derived from their IUCN profiles) showing that across all threat types, there is a continuum of increasing, decreasing and stable populations over time.

Revision 8. New analyses of shark and ray time-series, thus expanding the variety of taxa
across which we test variation in population change. While shark and rays had a negative
effect size for changes in abundance over time, the credible intervals around it overlapped
zero, suggesting that on average, shark and ray abundance has remained stable over time.

Revision 9. New sensitivity analyses testing for site-selection biases and duration effects
which demonstrated a lack of such biases.

We believe that we have been able to address all of the reviewer's feedback and that our
revisions have substantially improved our manuscript.

Reviewers' comments:

**Reviewer #1 (Remarks to the Author):**

In general, I thought this was an interesting paper - even though it's essentially a
non-result (rarity doesn't affect trends).

We thank the reviewer for this positive comment about our work. Our study tests
heterogeneity in population change across four themes: 1) biogeography, 2) taxonomy and
phylogenetic relatedness, 3) species' rarity and 4) species' threats and conservation status.
We find support for our null hypotheses that 1) increasing, stable and declining population
trends occur across all realms and biomes, 2) both rare and common species experience the
full spectrum of population trends, and 3) there are increasing and decreasing local
population trends across all levels of species IUCN conservation status, but also highlight
that we found 1) distinct taxonomic differences in population trends, with amphibians being
the only taxa that experienced net declines in the analyzed data, while birds, mammals and
reptiles on average became more abundant over time, and 2) more population fluctuations
for species with smaller mean population sizes.

I was specifically asked to comment on the methods and statistics, so will try to focus
my critiques there. I think the use of state-space models is appropriate, though
throughout I was a bit disappointed in the lack of details and specifics (equations,
citations).

We appreciate the reviewer taking the time to fully assess our analytical approach including
the state-space modelling framework. To provide more comprehensive detail, we have
revised our manuscript including in particular adding to our methods. The main text of the
manuscript now includes a new paragraph outlining the summary methods:

We analyzed vertebrate population time-series from the Living Planet Database (133,092
records) covering the period between 1970 and 2014. These time-series represent repeated
monitoring surveys of the number of individuals in a given area (species' abundance over

time), hereafter called “populations”. We focus on two aspects of population change – overall changes in abundance over time (population trend) and abundance variability over time (population fluctuations). In the first stage of our analyses, we quantified trends and fluctuations for each population using state-space models that account for observation error and random fluctuations³⁶ (Figure S1). In the second stage, we modelled the trend and fluctuation estimates from the first stage across latitude, realm, biome, taxa, rarity metrics, phylogenetic relatedness, species’ conservation status and threat type using a Bayesian modelling framework (Figure S2). We included a species random intercept effect to account for the possible correlation between the trends of populations from the same species (see table Table S1 for sample sizes). As sensitivity analyses, we additionally used variance weighting of the population trend estimates (μ) by the observation/measurement error around them (τ) and population trend estimates from linear model fits (slopes instead of μ) as the input variables in the second stage models, as well as several different fluctuations estimates. We also repeated our analyses on a single-country scale, using only populations within the United Kingdom, where monitoring efforts are particularly rigorous and extensive. All different analytical approaches yielded very similar results and are described in further detail in the methods.

We have provided equations for all models as well as citations that provide evidence to back up our statistical decisions (see lines 493-518). In addition, we have included two conceptual diagrams in the Supplementary Information outlining the two stages of our analyses, the input data, model and prior structure and model outputs (Figures S1 and S2). All code and data for our analyses are available on GitHub for review. To access the private repository, go to <https://github.com>, log in using username *PopChangeRarity* and password *atimeofchange9* and then navigate to <https://github.com/gndaskalova/PopChangeRarity>.

Revisions 2 and 5

Figure S1. Conceptual diagram of the first stage of our analyses where we calculated population trends and fluctuations. We analyzed vertebrate population time-series from the Living Planet Database (133,092 records) covering the period between 1970 and 2014. These time-series represent repeated monitoring surveys of the number of individuals in a

125 | given area (species' abundance over time), to which we refer as "populations". Diagram
| shows one sample population of Red squirrel (*Sciurus vulgaris*). We quantified two aspects
| of population change – overall change in abundance over time (population trends) and
| abundance variability over time (population fluctuations). We used state-space models that
| account for observation error and random fluctuations¹. The input abundance data for the
| state-space models were scaled to a common magnitude between zero and one to analyze
| within-population relationships to prevent conflating within-population relationships and
| between-population relationships². See methods for additional details.

2nd stage analyses: Test heterogeneity in population trends and fluctuations

Prior structure 1:
Hierarchical models in a Bayesian framework with weakly informative (flat) priors

$$Pr(\mu) \sim N(0, 10^8)$$

$$Pr(\sigma^2) \sim \text{Inverse Wishart}(V = 0, nu = 0)$$

Prior structure 2:
Hierarchical models in a Bayesian framework with weakly informative (parameter expanded) priors and a variance-covariance structure that allows the slopes of population trends and fluctuations to covary for each random effect.

$$Pr(\mu) \sim N(0, 10^8)$$

$$Pr(\sigma^2) \sim \text{Inverse Wishart}(V = 1, nu = 1)$$

Figure S2. Conceptual diagram of the second stage of our analyses where we quantified the geographic, taxonomic, rarity and threat patterns within vertebrate population trends and fluctuations. We modelled the trend and fluctuation estimates from the first stage (Figure S1) across latitude, realm, biome, taxa, rarity metrics, phylogenetic relatedness, species' conservation status and threat type using a Bayesian modelling

framework³. Each model included a species random intercept effect to account for the
possible correlation between the trends of populations from the same species. The prior
structure (weakly informative priors) was identical across all models except the phylogeny
models from the taxonomic patterns section, where the prior structure allowed for an
additional phylogeny random effect. See methods for additional details.

My 2 biggest criticisms of this are: 1. Interpreting uncertainty around μ as a metric
of variability is incorrect, and 2. not providing reviewers code / data during the review
is your responsibility -- a link to a public Github repo is really necessary

We thank Reviewer 1 for bringing attention to the different possible ways to estimate how
populations vary across time a theme of around one third of our manuscript. We agree with
Reviewer 1 that τ (the process error metric from the state-space models) can be used as a
metric of population variability. Our approach (using the confidence interval around μ)
addresses a slightly different question - as Reviewer 1 says, “the uncertainty in the long-term
trend”, which is related to how much populations fluctuate and is similar to testing the error
around the slope from a linear model (the other modelling approach we use). Testing how
τ varies based on species’ rarity and among taxa and biomes tests just the variance in the
process error (excluding any variance introduced due to the monitoring methods) and we
have redone all of our analyses using the process error. Our findings were consistent across
all the different metrics of population variability across time.

We have updated all figures to include results using the process error and have improved
each caption to explain what the different metrics of population change show, e.g.:

The five estimates in plot **g** refer to different analytical approaches, where the response
variables in the models were: 1) the standard error around the slope estimates of the linear
model fits of abundance versus year (circles), 2) half of the 95% confidence interval around
the μ value of population change (triangles), 3) half of the 95% confidence interval around μ
weighted by τ^2 , (full squares), 4) the process noise (σ^2) from the state-space models, and 5)
the standard deviation of the raw data for each population time-series (empty squares).
Effect sizes were standardized by dividing the effect size by the standard deviation of the
corresponding input data. The process noise is the total variance around the population
trend minus the variance attributed to observation error.

We very much agree with Reviewer 1 about the importance of transparency and
reproducibility and making code and data for studies publicly available and we have been
working towards best science open practice in all of our work. We have made all code and
data for our analyses available on GitHub for review. To access the private repository, go to
<https://github.com>, log in using username *PopChangeRarity* and password *atimeofchange9*
and then navigate to <https://github.com/gndaskalova/PopChangeRarity>. We will make all
code and data publicly available at time of publication with links and DOIs. We would like to
keep our code available only to the editorial board and reviewers until publication as these
analyses with open access datasets could be easily adapted by others working in this area,
potentially precluding the novelty of our study and its findings.

Revisions 5 and 6

Methods:

Line 381: Please comment on, or provide more details about this process. Additionally, please comment on whether or not this scaling affects inference (or biases inference) about the trend and variance parameters.

For example, standardizing the data to be \sim Normal (0, 1) before fitting these state space models affects estimates of the observation error, but not the trend or process variance. Scaling the data to be in the interval (0, 1) affects estimates of all three parameters: process and observation variances, as well as the trends.

We thank Reviewer 1 for the excellent feedback on the need to clarify the reasons behind our analytical decisions. We scaled the population data to be between 0 and 1 to analyse within-population relationships and to make sure that we were not conflating within-population relationships and between-population relationships (see van de Pol and Wright 2009 Animal Behaviour). We consider this approach to be appropriate for our analyses because the Living Planet Database includes thousands of populations measured using many different methods and units. We have detailed this justification in the methods:

Populations were monitored using different metrics of abundance (e.g., population indices vs. number of individuals). Before analysis, we scaled the abundance of each population to a common magnitude between zero and one to analyze within-population relationships to prevent conflating within-population relationships and between-population relationships⁶⁹. This allowed us to explore trends among populations relative to the variation experienced across each time series.

Another approach to transforming the data to prepare it for analysis is to log the raw abundance data, model $\log(\text{abundance})$ over time and then back-transform the population trends to show percentage change over time. We consider scaling the abundance data to be between zero and one more appropriate for the purposes of our analyses, but below we show that using logged abundance, instead of scaled abundance, produces a very similar distribution of population trend values (i.e., most populations are stable over time, and there are approximately equal amounts of increasing and decreasing populations).

Figure 1. The distribution of population change (percentage change in abundance over the duration of each population time-series) is centered on zero, with similar amounts of increasing and decreasing populations. Population change values were calculated using state-space models where the input data were logged abundance records. Using logged abundance produced very similar results to the approach we use (using scaled abundance between zero and one).

Reference:

van de Pol, M. & Wright, J. A simple method for distinguishing within- versus between-subject effects using mixed models. *Anim. Behav.* **77**, 753–758 (2009).

Revisions 2 and 5

Line 389-444: compiling all these data seem like a big task. Are they included in the supplementary info? It might be helpful for future researchers if you included them

We agree with Reviewer 1 and have included our data files in the GitHub repository accompanying our study. The data files are described in the README.md file. We will make the repository public at time of publication.

Revision 5

Line 448: I'm confused by the term 'freely-varying intercept'? I think writing out the equations, and citing software you used - or providing code - is necessary here

We appreciate Reviewer 1's suggestions on how to improve the clarity of our methods and have followed all of them – we have revised the text to say “random intercept” and include further details, we have added the equations, references to the software we have used, and have made all code available on GitHub.

*Revisions 2 and 5*

Line 455-457: This decision to use the uncertainty metric around μ is wrong.
Uncertainty around μ represents uncertainty in the long term trend, not variability.
Uncertainty around μ is influenced by a number of things - including observation
error, missing data, etc. The process variance represents the variability around each
population trend and is a better metric for variability. Please change.

See above. We thank the reviewer for bringing this to our attention and have redone all of
our analyses using the process error from the state-space models. We have clarified what
the different metrics of population variability show in the main text and in all figure captions,
and we have presented results using the different metrics to allow comparison. An example
caption now reads:

**Figure 1. Population declines, increases and fluctuations over time occur across all**
**latitudes and biomes within the freshwater, marine and terrestrial realms.** Results
include 9286 populations from 2084 species. The lack of biogeographic patterning in
vertebrate population trends was also apparent on a UK scale (Figure S6 and Table S2).
The numbers in the legend for plots **d-g** and on the x axis in plot **c** show the sample sizes for
realms and biomes, respectively. The μ values of population trend (plots **a-b, d-e, h**) and the
σ^2 values of population fluctuation (plots **c, f-g**) were derived from state-space model fits of
changes in abundance over the monitoring duration for each population. Plots **d** and **f** show
the density distribution of population trends across realms, the raw values (points) and
boxplots with the mean, first and third quartiles and boxplot whiskers that indicate the
distance that covers 1.5 times the interquartile range. Plots **e, g** and **h** show the effect sizes
and the 95% credible intervals of population trends (**e, h**) across realms and biomes, and
fluctuations across realms (**g**). For variation in fluctuations across biomes, see SI Figure S9.
The three estimates for each realm/system in plots **e** and **h** refer to different analytical
approaches: population trends calculated using linear models (circles), state-space models
(μ , triangles), and population trends (μ) weighted by τ^2 , the observation error estimate from
the state-space models (squares). The five estimates in plot **g** refer to different analytical
approaches, where the response variables in the models were: 1) the standard error around
the slope estimates of the linear model fits of abundance versus year (circles), 2) half of the
95% confidence interval around the μ value of population change (triangles), 3) half of the
95% confidence interval around μ weighted by τ^2 , (full squares), 4) the process noise (σ^2)
from the state-space models, and 5) the standard deviation of the raw data for each
population time-series (empty squares). Effect sizes were standardized by dividing the effect
size by the standard deviation of the corresponding input data. The process noise is the total
variance around the population trend minus the variance attributed to observation error.
Error bars in plots **e, g** and **h** show 95% credible intervals. See Table S2 for model outputs.

*Revisions 5 and 6*

Line 466-467: I'm confused about the 2-stage modeling. In a Bayesian world, μ is a
distribution, not a point estimate. How did you carry through the uncertainty in μ

into this 2nd stage? As before, please include some equations and citations for the
software you used -- or code.

We have included two conceptual diagrams in the Supplementary Information (Figures S1
and S2) to clarify the inputs and outputs of each of the two stages in our analyses, as well as
the model and prior structure (where error is incorporated) for each. Carrying uncertainty
through multistage analyses is an important analysis aspect. We have repeated all of our
analyses using a variance-weighting meta-analytical approach, where in the second stage of
our study, e.g. when testing population trends across biomes, we weighted the response
variable (population trend, μ , from the state-space models) by the observation error (τ
from the state-space models). The results from the weighted analyses are presented in all of
the main text figures to allow comparison between different methods. Our findings were
consistent across the different approaches. We have added equations, citations for software
and have made the code available on GitHub.

Due to the complexity of the state-space models and size of the datasets, we were not able
to implement a fully integrated modelling framework in this study, though that would be our
preference in other projects (e.g. Bjorkman *et al.* 2018) and is something that we advocate
for when possible. For the present study, the two-stage analysis was preferable because it
allowed us to use state-space models to quantify population change (first stage) and then
test population change across different factors like taxa, realm, etc., using a Bayesian
modelling framework (second stage). These two types of modelling frameworks complement
each other, but it is not possible to integrate them in a single model.

Reference

Bjorkman, Anne D., Isla H. Myers-Smith, Sarah C. Elmendorf, Signe Normand, Nadja R ger,
Pieter SA Beck, Anne Blach-Overgaard *et al.* (2018). Plant functional trait change across a
warming tundra biome.", *Nature* **562**.

*Revision 5*

Line 471: Don't refer to predictors as 'significant' if you're using Bayesian methods

We have removed all mentions of "significant" and agree that the term is not appropriate for
reporting Bayesian methods, though it can help to communicate to broader audiences.

Line 497: Again, equations here, and more specifics (like the levels of each of the
random effects is essential)

We have added more details (lines 493-518, lines 569-658, see comments above as well).
The random effect structure for each statistical test is outlined in Figures S1 and S2, as well
as in the methods where we have also included equations.

*Revision 5*

Line 512: You can't say "We used parameter expanded priors..." without giving any specifics or citations and expect the reader to follow. Maybe including a table or parameters and priors would be helpful

We thank the reviewer for this thoughtful suggestion and we have produced two conceptual diagrams (Figures S1 and S2) that outline all parameters and model structures in our statistical analyses. We hope that these conceptual diagrams will concisely yet comprehensively communicate our analytical framework to our readers.

Revision 5

Line 534-541: I think including some citations for each of these metrics would be helpful.

Following the reviewer's recommendation, we have included references for the metrics we have used and have added more details in the methods to further clarify them.

Humbert, J.-Y., Scott Mills, L., Horne, J. S. & Dennis, B. A better way to estimate population trends. *Oikos* **118**, 1940–1946 (2009).

van de Pol, M. & Wright, J. A simple method for distinguishing within- versus between-subject effects using mixed models. *Anim. Behav.* **77**, 753–758 (2009).

Revision 5

Line 547: I find this kind of statement pretty difficult to reconcile as a reviewer. You should have included the code and link to the github repository -- with all data used in the paper -- in your initial submission

"To review the code for our statistical analyses prior to publication, please contact the authors. Population time-series data are publicly available "

We wholeheartedly support open science and reproducibility and we will make sure our work matches those values at the time of publication, but we note that sometimes there are legitimate reasons why one might not make the code public at time of submission of a manuscript. In our case, we have not made all of our code publicly available before submission because this is a very timely analysis that uses publicly available data, with other research groups working in similar areas, making it possible for people to easily use our work and conduct the same analyses prior to the publication of our study. We will make the code and data compilation public at time of publication (and we note that all data are already publicly available from their respective sources).

We have made all code and data for our analyses available on GitHub for review. To access the private repository, go to <https://github.com>, log in using username *PopChangeRarity* and

386 password *atimeofchange9* and then navigate to
387 <https://github.com/gndaskalova/PopChangeRarity>. We note that the code will only run on a
388 high-memory computing system due to the computing intensity of the data integration and
389 statistical analyses particularly at the phylogeny modelling stage.

*Revision 5*

**Reviewer #2 (Remarks to the Author):**

The reams of tests and data presented are impressive at first glance, and I
appreciate that the authors wish to establish a novelty that has been previously
unappreciated.

We thank the reviewer for this positive assessment of our work and in particular for
highlighting the breadth and novelty of our analyses.

However, they fail to do this for both conceptual, data, and methodological reasons.

We thank Reviewer 2 for suggesting areas for improvements in our work and we have
undertaken major revisions and conducted new analyses to enhance the quality of our study
that we outline below:

Conceptual improvements

We have revised the text of our manuscript to focus on population change and have clarified
how population change links to extinction risk in the relevant sections (see lines 85-86, 104-
106, 362-364, 384-387). We have included two conceptual diagrams (Figures S1 and S2)
that summarize our workflows. We have made a clear distinction between species' IUCN
conservation status and the specific threats that each species faces, and we have added a
new analysis of population change across species' threats showing that across all threat
types, there is a continuum of negative, positive and stable population trends at local scales
(lines 252-257, Figure 4).

Discussion of data and methodological limitations

We agree with the data limitations outlined by Reviewer 2. We have added new content to
our manuscript specifically addressing the limitations of the Living Planet Database and their
potential influences on our findings, which reads:

**Data limitations**

*Taxonomic and geographic gaps*

Our analysis is based on 9286 monitored populations from 2084 species from the largest
currently available public database of population time-series, the Living Planet Database⁶⁷.
Nevertheless, the data are characterized by both taxonomic and geographic gaps that can
influence our findings. For example, there are very few population records from the Amazon
and Siberia (Figure 1b) – two regions currently undergoing rapid environmental changes due
to land-use change and climate change respectively. Additionally, birds represent 63% of all

population time-series in the Living Planet Database, whilst taxa such as amphibians and
sharks where we find declines are included with fewer records (Figures 2 and S5). On a
larger scale, the Living Planet Database under-represents populations outside of Europe and
North America and over-represents common and well-studied species⁵⁹. We found that for
the populations and species represented by current monitoring, rarity does not explain
variation in population trends, but we note that the relationship between population change
and rarity metrics could differ for highly endemic specialist species that are not included in
the Living Planet Database. As ongoing and future monitoring begins to fill in the taxonomic
and geographic gaps in existing datasets, we will be able to re-assess and test the generality
of the patterns of population change across biomes, taxa, phylogenies, species traits and
threats.

*Monitoring extent and survey techniques*

The Living Planet Database combines population time-series where survey methods were
consistent within time-series but varied among time-series. Thus, among populations,
abundance was measured using different units and over varying spatial extents. There are
no estimates of error around the raw population abundance values available and detection
probability likely varies among species. We used state-space models to estimate trends and
fluctuations to account for these limitations as this modelling framework is particularly
appropriate for analyses of data collected using disparate methods^{37,78,79}. Because the
precise coordinates of the polygons where the individual populations were monitored are not
available, we were not able to test for the potential confounding effect of monitoring extent,
but our sensitivity analysis indicated that survey units do not explain variation in the detected
trends (Figure S13).

*Temporal gaps*

The population time-series we studied cover the period between 1970 and 2014, with both
duration of monitoring and the frequency of surveys varying across time-series. We only
included populations with more than five survey points in time in our analyses, because this
amount of data has been found to be enough to detect directional trends in 80-90% of
cases⁶⁸. In a separate analysis, we found significant lags in population change following
disturbances (forest loss) and that population monitoring often begins decades to centuries
after peak forest loss has occurred at a given site⁴¹. The findings of this related study
suggest that the temporal span of the population monitoring does not always capture the
period of intense environmental change and lags suggest that there might be abundance
changes that have not yet manifested themselves. Thus, the detected trends and the
baseline across which trends are compared might be influenced by when monitoring takes
place and at what temporal frequency. Nevertheless, the Living Planet Database represents
the most comprehensive compilation of temporal population records to date, allowing for the
most comprehensive analyses possible into the patterns of vertebrate trends and
fluctuations around the world.

We have clarified that our result of no effects of rarity on population trends is exclusively for
the species we studied, and the rarity-population trend relationships might be different for
highly endemic species that are not represented in the Living Planet Database (lines 714-
716). Developing and collating a comprehensive global database with both species

abundance and measurement error information across time is a future research priority.
Going beyond state-space models and exploring how we can incorporate data quality into
analyses of global population change over time is something that our team is very interested
in working on, but such endeavour goes beyond the scope of this particular analysis.

In our revised manuscript, we include an analysis of the relationship between population
change and geographic range for both mammal and bird species, thus covering the species
for which range data are currently available (Figure 3). For the remaining two rarity metrics,
mean population size and habitat specificity, we included all possible species across bird,
mammal, bony fish, shark and ray, amphibian and reptile taxa.

The Living Planet Database population time-series data are widely used and reported on in
the scientific literature, and so far, addressing the lack of time-specific estimates of
measurement error, has not been possible (Leung *et al.* 2017, van Strien *et al.* 2016,
Spooner *et al.* 2018). The Living Planet Database is the largest currently available data
compilation for testing the biogeography and drivers of population change around the world,
and by using a state-space modelling framework, we address the uncertainty in the time-
series as best as currently possible.

References:

Leung, B., Greenberg, D. A. & Green, D. M. Trends in mean growth and stability in
temperate vertebrate populations. *Divers. Distrib.* **23**, 1372–1380 (2017).

van Strien, A. J. *et al.* Modest recovery of biodiversity in a western European country: The
Living Planet Index for the Netherlands. *Biol. Conserv.* **200**, 44–50 (2016).

Spooner, F. E. B., Pearson, R. G. & Freeman, R. Rapid warming is associated with
population decline among terrestrial birds and mammals globally. *Glob. Change Biol.* (2018).
doi:10.1111/gcb.14361

*Revisions 1, 2, 3, 4 and 5*

I have several major concerns with the data applied. First, despite applying a state-
space model to their time series, most of the Living Planet Database does not
provide measurement error for their time-specific estimates of abundance. This is not
the authors fault, but it is a major issue with population-dynamical time series in
general. As such, a state-space (or any other, for that matter) approach will not
remove the huge probability of making substantial Type II errors in any hypothesis
test. This is made even more serious and pertinent because the authors attempt to
tell us something useful for rare species in particular, when the measurement error
for these is likely to much, much higher.

The reviewer's point about the lack of time-specific measurement error around the
population estimates is valid, and we have added text to our manuscript that discusses this
limitation of the database (see lines 679-689, and the full text of the new content in the

response to the reviewer's previous point). Incorporating the measurement error for each
population (derived from the state-space models) in our modelling framework showed very
similar results to when the measurement error was not included in the models, thus
increasing our confidence in the finding of no relationships between rarity and population
trends.

With large datasets such as the Living Planet Database, we can find small effects that are
statistically significant, and equally we can find a lot of variation that could obscure
relationships. This means that both Type I and Type II errors could potentially be at play,
over which as the reviewer pointed out we have no control, but what we can do is use
appropriate modelling framework (state-space models) that partition variance around
population estimates into observation error and population fluctuations. By making the code
and data available, we are facilitating further engagement with the analyses, allowing other
potentially interested researchers to explore details of our study that are beyond the scope of
the present analysis. The transparency in our workflows also means that as new data
become available and for example, scientific advances improve the precision and accuracy
of the monitoring of rare species, our analyses can be re-run, thus testing if the finding of no
relationship between rarity and population change is also valid for species not currently
included in the Living Planet Database.

For further details, please see our responses above addressing the state-space modeling
approach and limitations of the data set.

Thus, even the relatively interesting result that small-population species tend to have
higher temporal variance (but previously established much more comprehensively by
others, such as in Melbourne & Hastings 2008 Nature 454:100 and Fagan et al. Ecol
Lett 9:51 — the latter is a classic that the authors did not even cite) is rendered highly
suspect because of the likely inflation of measurement error for these species.

We thank the reviewer for recognizing our finding as interesting and for providing useful
suggestions for additional literature to better place our findings in the broader context of
population change and extinction risk. A real signal of population change always has the
potential to be obscured by measurement errors when populations are small but testing this
relationship across many populations increases the confidence in the generality of the
findings. Melbourne & Hastings 2008 was based on one laboratory population of *Tribolium*
*castaneum* (red flour beetle), and Fagan & Holmes 2005 is based on ten wild vertebrate
populations. We quantified population trends and fluctuations for 9284 population time-series
from 2084 species and we used state-space models specifically to address the data
limitations raised by the reviewer (state-space models account for process noise (σ^2) and
observation error (τ^2) and thus deliver robust estimates of population change when working
with large datasets where records were collected using different approaches, see methods
for further details). We have revised our manuscript to also include results using the process
error, sigma, as an estimate of population fluctuations. The process noise is the remaining

variance around the trend once the observation error has been accounted for. We have
included the Fagan *et al.* Ecol Letters reference as per the reviewer's suggestion (see line
387). We have also tested the sensitivity of our findings to the measurement error, derived
from the state-space models, for each population. We repeated all of our analyses using a
variance-weighting meta-analytical approach, where in the second stage of our study, we
weighted the response variable (population trend, μ , from the state-space models) by the
observation error (τ from the state-space models). Our findings were consistent across the
different analytical approaches.

*Revisions 1 and 4*

The authors also make the naïve mistake of failing to distinguish the small- and
declining-population paradigms (Caughley 1994 J Anim Ecol 63:215) in their
analyses. The processes driving declines ARE NOT, nearly ever, the same that drive
small populations to extinction. Thus, searching for relationships between rarity and
decline is something of a waste of time, merely because we know that the processes
differ between the two categories (separated by the concept of the minimum viable
population size).

We thank the reviewer for raising this important point, we have clarified the distinction
between small and declining populations in our manuscript and have added text to highlight
that population declines and species' extinction occur on very different time scales (lines 83-
86, 104-106, 362-364, 384-387), with the revised text being:

As per population dynamics theory^{22,23} and Taylor's power law²⁴, species with small
populations are more likely to undergo stochastic fluctuations that could lead to pronounced
declines, local extinction and eventually global species extinction⁵. Small populations are
also more likely to decline due to inbreeding^{25,26}. Allee effects, the relationship between
individual fitness and population density, further increase the likelihood of declines due to
lack of potential mates and low reproductive output once populations reach a critically low
density^{27,28}.

...

Testing population change across species' IUCN conservation status allows us to link
contemporary changes in abundance with long-term probability of extinction³⁴.

...

While phylogenetic clustering might be lacking in contemporary trends, there is evidence that
phylogenetic relatedness predicts extinction, a process occurring over much longer time
scales^{6,7}.

...

We found that species with small populations were, nevertheless, more likely to fluctuate
(Figure 3f), which may increase their probability of extinction, a process that could play out
over longer time-scales than found for most population monitoring time-series^{22,23,54}.

*Revision 1*

Another major concern is that 'extinction risk' is being inferred constantly throughout
this manuscript from 'threat risk'. These are not the same thing. First, the IUCN Red
List uses multiple criteria to set relative threat risk (not extinction risk), yet the authors
do not distinguish among Criteria when testing their relationships. There is really only
one Criterion here that is relevant - Criterion A justifications (population decline), yet
the authors use all Criteria to establish 'extinction risk'. This mixes different drivers
and will necessarily dilute the capacity to identify relationships. It is therefore not
surprising in the slightest that the authors did not find the relationships they purport to
test hypothetically.

We agree that threat and extinction are not the same conceptually; however, our goal was
not to redefine terms already in use in the literature. In our study, we use the classification
provided by the IUCN, which they title “extinction risk” categories. By incorporating the IUCN
extinction risk categories, we sought to test how local scale population trends vary across
the globally determined Red List species’ status. We use the IUCN Red List categories as
extinction risk metrics to be consistent with IUCN’s terminology -
see <https://www.iucnredlist.org> and specifically the definition “The IUCN Red List Categories
and Criteria are intended to be an easily and widely understood system for classifying
species at high risk of global extinction.” The Red List classifications have been frequently
used as metrics of extinction risk, for example the recent publication published in the journal
Nature Communications by Di Marco *et al.* [https://www.nature.com/articles/s41467-018-
07049-5](https://www.nature.com/articles/s41467-018-07049-5).

The reviewer is correct that the IUCN Red List categories are determined based on different
criteria, or a combination of criteria. Most of our study species were classified based on
multiple risk criteria, thus providing an extraction of only Criterion A was not possible.
However, we are using these categories in the same ways as other studies and they do
provide a test of whether species categorized by the IUCN as globally under threat are
experiencing more or less population change, including declines, at local scales. We found
increasing, decreasing and stable populations across all Red List categories of conservation
status (Figure 4a). In a follow up analysis, we explore land-use and forest cover change
specifically (Daskalova *et al.*, preprint available on bioRxiv). Additionally, we are in the
process of exploring the cumulative and interactive effects of climate change, land-use
change and other types of global change on population trends over time, but that is the
focus of another manuscript.

References:

Daskalova, G.N., Myers-Smith, I.H., Bjorkman, A.D., Blowes, S.A., Supp, S.R., Magurran,
 655 A., Dornelas M. (2018) Forest loss as a catalyst of population and biodiversity change.
 bioRxiv <https://doi.org/10.1101/473645>

 *Revisions 1 and 7*

 **Figure 4. On local scales, there are increasing, decreasing and stable populations**
 **across the full spectrum of the globally-determined species' conservation status and**
 **anthropogenic threats (IUCN Red List Index).** Numbers in the legend for plots a-d and in
 plots e-f show sample size for each metric. Plots a and c show the density distribution of
 population trends across Red List status, the raw values (points) and boxplots with the
 mean, first and third quartiles and boxplot whiskers that indicate the distance that covers 1.5
 667 times the interquartile range. Plots b and d show the effect sizes and the 95% credible
 intervals of population trends (b) and fluctuations (d) across Red List status categories. The
 μ values of population trend (plots a, e-f) and the σ^2 values of population fluctuation (plots c)
 were derived from state-space model fits of changes in abundance over the monitoring
 duration for each population. For the relationships between type and number of threats and
 population fluctuations, see Figure S19. Plots b and d show the effect sizes and the 95%
 credible intervals for population trends (b) and fluctuations (d) across conservation status
 across the IUCN Red List categories. The three estimates for each metric in plot b refer to
 different analytical approaches: population trends calculated using linear models (circles),
 state-space models (μ , triangles), and population trends (μ) weighted by τ^2 , the observation
 error estimate from the state-space models (squares). The five estimates in plot d refer to
 different analytical approaches, where the response variables in the models were: 1) the
 standard error around the slope estimates of the linear model fits of abundance versus year
 (circles), 2) half of the 95% confidence interval around the μ value of population change
 (triangles), 3) half of the 95% confidence interval around μ weighted by τ^2 , (full squares), 4)
 the process noise (σ^2) from the state-space models, and 5) the standard deviation of the raw

data for each population time-series (empty squares). The process noise is the total variance
around the population trend minus the variance attributed to observation error. Effect sizes
(plots **b** and **d**) were standardized by dividing the effect size by the standard deviation of the
corresponding input data. Error bars in plots **b** and **d** show 95% credible intervals. Plot **e**
shows the distributions of population trends across different threats that the species face
globally, with the central tendencies of all distributions overlapping with zero. Lines in plot **f**
show model fit and 95% credible intervals, where “number of threats” refers to the number of
different threats that each species, whose populations are locally monitored, are exposed to
on a global scale. See Methods for details on deriving types of threats and Table S2 for
model outputs.

For an example of one of the few articles I know that has tested drivers of 'extinction
risk' vs. 'threat risk', Sodhi et al. (2008 Div Disturb 14:1; again, that the authors did
not cite). Somewhat interestingly, the latter in fact supports the authors' contention
that extinction risk is difficult to predict, but the authors here are in no way testing
'extinction risk'!

We agree that extinction risk and threat risk are different terms, but they both relate to
population declines and fluctuations. We did not set out to test extinction risk, but how local
population trends relate to species' threat of going globally extinct, and we have clarified this
in the revised text (lines 111-125):

Here, we asked how the trends and fluctuations of vertebrate populations vary with
biogeography, taxa, phylogenetic relationships and across species' rarity metrics and IUCN
conservation and threat categories.

...
We investigated whether the heterogeneity in population change globally is explained by
differences in species' rarity and IUCN conservation status.

...
In a *post-hoc* analysis, we categorized each population based on the global IUCN threat
categories to determine how the number and type of threats influences local-scale
population change.

In response to the reviewer's feedback, we have added a new set of analyses testing
population change across the types of threats species face. We found that there were no
distinct signatures of global species threats which were associated with predominantly
declining local trends of monitored populations (Figure 4e) and there were increasing,
decreasing and stable trends across all threat types. We thank the reviewer for providing
further useful references. We are already at the citation limits and thus have decided to
focus on large-scale studies that use similar metrics to our analysis. Though, if the editor
deems it appropriate, we would be happy to add additional references.

Revisions 1 and 7

Finally, the authors take a decidedly narrow view of extinction of life, when in fact there are plenty of examples from palaeontological work that establishes rather convincingly the very patterns these authors say do not exist. In fact, one of the best examples of how rarity and geographic size affect actual extinction risk is palaeo work across 1/2 billion years. Taking the modern snapshot to define generalities is therefore suspect because the paleo record is clear here:
doi:10.1098/rspb.2012.1902.

We thank the reviewer for placing our results in the larger palaeoecological context. In response to the reviewer's comment, we have included text about the palaeoecological context of the rarity-extinction relationships (see 92-95):

The fossil record indicates that on millennial time scales, rare species are more likely to decline and ultimately go extinct³⁰, but human actions have pushed Earth away from traditional geological trajectories³¹, and the relationships between rarity and population change across the planet have yet to be tested across the Anthropocene.

We think that comparing the modern-day relationships between population trends and rarity with those from the fossil record over geological time scales would be an interesting and important piece of research, but we feel it is beyond the scope of our present study.

Some more specific comments follow:

- L78-79: This is not entirely true. If we use Red List status as 'threat risk', then there are many missing references and taxa showing strong effects of geographic range — including: doi:10.1016/S0065-2881(09)56004-X (fish), doi:10.1371/journal.pone.0001636 (amphibians), doi:10.1111/j.1365-2745.2008.01408.x (legumes) — and these are just some examples.

We have clarified and revised the introduction and discussion to focus on population change, not extinction risk, and we have outlined how population change relates to extinction risk over longer time-scales than those studied here (population change across 1970-2014, see lines 104-106, 362-364, 384-387):

Testing population change across species' IUCN conservation status allows us to link contemporary changes in abundance with long-term probability of extinction³⁴.

...

While phylogenetic clustering might be lacking in contemporary trends, there is evidence that phylogenetic relatedness predicts extinction, a process occurring over much longer time scales^{6,7}.

...

We found that species with small populations were, nevertheless, more likely to fluctuate (Figure 3f), which may increase their probability of extinction, a process that could play out over longer time-scales than found for most population monitoring time-series^{22,23,54}.

Our goal was to test the relationships between rarity metrics (one of which being geographic range) and local population change, and not between geographic range and threat risk. We have rewritten the entire manuscript to echo this distinction.

Revision 1

- L93: Another important study that the authors are missing, but that I think supports the contention is for British birds, is where local climate is the dominant driver of expansion or contraction of range, with contributions of life-history pace and geographical context: doi:10.1098/rspb.2014.0744 (i.e., drivers of change are complicated, and locally dependent).

We thank the reviewer for providing a reference. We are already at the citation limits and thus have decided to cite studies reporting the relationships between rarity and population change, and not studies reporting changes in the rarity metrics themselves.

- L204-205: I contend that this result arises merely because of the lower expected measurement error associated with the better-monitored terrestrial species

We agree, and have added text to reflect that, but we also highlight that our state-space modelling framework already partitions estimated measurement error from population fluctuations. We have also added a section to the methods covering the heterogeneous methods of population monitoring within the Living Planet Database to better clarify the limitations of our study and global population monitoring in general (see lines 679-689):

Monitoring extent and survey techniques

The Living Planet Database combines population time-series where survey methods were consistent within time-series but varied among time-series. Thus, among populations, abundance was measured using different units and over varying spatial extents. There are no estimates of error around the raw population abundance values available and detection probability likely varies among species. We used state-space models to estimate trends and fluctuations to account for these limitations as this modelling framework is particularly appropriate for analyses of data collected using disparate methods^{37,78,79}. Because the precise coordinates of the polygons where the individual populations were monitored are not available, we were not able to test for the potential confounding effect of monitoring extent, but our sensitivity analysis indicated that survey units do not explain variation in the detected trends (Figure S13).

- L270-272: Some of these results could also be partially explained by the
observation that different regions are experiencing different phases of the extinction
process. For example, regions such as Europe have experienced massive declines
in species for many millennia, resulting in a depauperate fauna that has adapted to
the human-dominated landscape. Elsewhere in only recently disturbed areas, the
most sensitive species are only now starting to dwindle. Thus, you cannot compare
regions simply by species lists — you have to take time since disturbance into
account, or the conclusions are not valid (apples and oranges, as we say).

We thank the reviewer for bringing attention to the importance of differences in when sites
around the world have experienced major disturbances. This is a very important point and is
a research avenue we have explicitly explored in the context of forest cover change in
another manuscript. We have added text to discuss this (see lines 376-384):

*The power of rarity metrics to predict population trends could also be mediated by whether*
*species are naturally rare, or have become rare due to external drivers in recent years^{50,51}.*
*Naturally rare species might be more likely to persist over time, whereas species that have*
*more recently become rare might be more likely to decline in response to environmental*
*disturbance. Furthermore, the timing and magnitude of past and current disturbance events*
*influence population trends^{41,52} and there could be temporal lags in both positive and*
*negative abundance changes over time^{41,53}. However, disentangling the processes leading*
*to rarity over time remains challenging, and across the 2084 species we studied, there are*
*likely cases of both natural and human-driven population change.*

We have included a reference to another analysis of ours that was not published as a
preprint at the time of submitting this manuscript, where we used the same population time-
series to test the effects of time since disturbance (forest loss) on both historic (850-2015)
and contemporary (2000-2016) time-scales. We found that at sites experiencing the highest
all-time levels of peak forest loss (calculated between 850-2015) during the monitoring
period of population time-series, the abundance of species was twice more likely to decline
compared to when monitoring started before or after peak forest loss. Across all sites
represented in the publicly available Living Planet Database, however, there were both
increases and decreases in species' abundance after forest loss. These results were not
completed at the time of submission of the present study, but now we can cross reference
the forest cover change analysis in this study.

Reference:

Daskalova, G.N., Myers-Smith, I.H., Bjorkman, A.D., Blowes, S.A., Supp, S.R., Magurran,
858 A., Dornelas M. (2018) Forest loss as a catalyst of population and biodiversity change.
bioRxiv <https://doi.org/10.1101/473645>

*Revision 1*

- L288-289: Completely unsupported speculation

We have revised the text to clarify that we suggest that the local scale variation in the threats
to which species are exposed is a possible contributing factor to the heterogeneity in
population trends across the planet. We have edited the text so that no causal link is implied
and testing this possible explanation empirically will be the focus of future studies from our
team:

Coarsely represented biogeographic regions and global-scale species' IUCN status and
threat categories might not capture the drivers acting in the locations of the specific
populations we studied^{32,38-40}. Furthermore, the same driver can have opposing effects on
population abundance at different sites⁴¹. A lack of of biome-specific directional trends in
population change, despite a spatial clustering of human pressure around the world¹², can
also arise due to differences in species traits and vulnerability to environmental change
within biomes¹⁶⁻¹⁸.

...

As human activities continue to accelerate, the next key step is to determine how intrinsic
factors, such as rarity attributes and threats, interact with extrinsic global change drivers and
together influence the persistence of Earth's biota.

We have already found evidence that the effects of forest loss vary across the globe, with
both increases and declines in population abundance detected after peak forest loss across
sites (Daskalova *et al.*, preprint available on bioRxiv). We are planning to explore other
potential drivers of change in species' populations in our future studies.

Reference:

Daskalova, G.N., Myers-Smith, I.H., Bjorkman, A.D., Blowes, S.A., Supp, S.R., Magurran,
892 A., Dornelas M. (2018) Forest loss as a catalyst of population and biodiversity change.
bioRxiv <https://doi.org/10.1101/473645>

*Revision 1*

- L380: This is an arbitrary value with no justification. Some sensitivity analysis is
required at the very least.

We thank the reviewer for bringing attention to the importance of considering time-series
duration. We have added reference to a new study by Wauchope *et al.* that was not
available at the time of initial submission of this manuscript, addressing precisely the issue
raised by the reviewer (see lines 465 and 695). By explicitly testing how different durations
affect the detected trends, Wauchope *et al.* found that in 80-90% of cases when there was a
directional population trend, time-series with a 5-year duration or more will capture that
directional trend accurately. We have also included a statistical test and visualisations of the

effects of study duration on detected population trends across the time-series we studied
(see Figures S3 and S6, as well as model outputs for the “Duration” models in Tables S2
and S3). Both positive and negative vertebrate population trends had a smaller magnitude
for longer time-series.

In our revised manuscript, we have also included a new sensitivity analysis (see lines 727-
735). Following Fournier *et al.* 2019⁶³, we tested the time-series we analyzed for site-
selection bias. Removing the first five survey points reduces the bias stemming from starting
population surveys at points when individual density is high, whereas removing the last five
916 years reduces the bias of starting surveys when species are very rare. The distribution of
917 population trend values across time-series was not sensitive to the omission of the first five
(left-truncation) or the last five years (right-truncation) of population records (Figure S8).
Overall, our sensitivity analyses confirmed that our findings were robust to the potential
confounding effects of differences in monitoring duration, sampling method and site-
selection.

**Figure S8. The distribution of population trend values across time-series was not**
**sensitive to the omission of the first five (left-truncation) or the last five years (right-**
**truncation) of population records.** Following Fournier *et al.* 2019⁴, we tested the time-
series we analyzed for site-selection bias. Removing the first five survey points reduces the
bias stemming from starting population surveys at points when individual density is high,
whereas removing the last five years reduces the bias of starting surveys when species are

| very rare. There were slightly fewer trends centered on zero (no net change in abundance
| over time) when we left- and right-truncated the data, suggesting that longer time-series are
| more likely to detect no net changes in abundance (see Figure S6 for a visualization of
| population trends versus monitoring duration

References:

Wauchope, H. S., Johnston, A., Amano, T. & Sutherland, W. J. When can we trust population trends? A method for quantifying the effects of sampling interval and duration. *bioRxiv* (2019). doi:10.1101/498170

Fournier, A. M. V., White, E. R. & Heard, S. B. Site-selection bias can drive apparent population declines in long-term studies. doi:10.7287/peerj.preprints.27507v1

Revisions 1 and 9

Reviewer #3 (Remarks to the Author):

This paper explores patterns of population change in vertebrates, using data from the Living Planet Index Database (LPD). It tests the hypothesis that species rarity and conservation status predict population declines and fluctuations.

Exploring these questions is of obvious interest to applied nature conservation perspective, but they also address the relationship population dynamics and species status, which fundamentally underpins macroecology and dynamic biogeography. Therefore, I expect the results of this analysis, which are based on an impressively large dataset, to generate a high level of interest among a broad readership.

We thank the reviewer for highlighting that our analyses will likely generate a high level of interest and will attract a broad readership.

Overall I was impressed by the scope of the study and am sympathetic to its aims. However, I have a number of serious reservations about the conclusions, about potential biases in the analysis and about a lack of detail in the methods.

We thank the reviewer for their positive assessment of our study and its aims. We are further grateful for the reviewer bringing up important points regarding clarity in the methods – we have revised our manuscript to include more details on the methods, including statistical decisions, a section on the limitations of the database we used, further citations to back up our modelling framework and the equations for all models. We have included conceptual diagrams to visualise our methods (Figures S1 and S2). We have also conducted new sensitivity analyses testing for site-selection biases following Fournier *et al.* 2019, which showed no identifiable signs of such bias (see lines 727-735). We believe that through the structural and content revisions we have undertaken, as well as the additional new analyses, our manuscript has been greatly improved (further details below).

Reference:

Fournier, A. M. V., White, E. R. & Heard, S. B. Site-selection bias can drive apparent population declines in long-term studies. doi:10.7287/peerj.preprints.27507v1

Revisions 1, 2, 4, 5 and 9

Major comments

1. This is an important analysis utilising LPD data in a novel way. The abstract and title are concise and clear, but the manuscript itself doesn't provide the same kind of synthesis – it's a list of results. For me the big conclusion is not so much that rarity metrics are unimportant, but that variation within categories is greater than among. In addition, the study reported no net population declines (except for amphibians) which stands in contrast to the LPI report which found a 58% decline in species abundance. Seeing as both analyses use the same database, this should definitely be discussed.

We appreciate the reviewer highlighting the importance and novelty of our study and we appreciate the feedback that the results lack synthesis. In line with the reviewer's suggestions, we have restructured our manuscript and have majorly revised the text to add more interpretation of the findings in the discussion. In addition, we have added a paragraph specifically discussing our results and how they compare with the Living Planet Index results.

The Living Planet Index reports (e.g., McRae *et al.* 2012, McRae *et al.* 2016 and WWF 2018) also document that the numbers of declining and increasing species are similar across this dataset, but the calculation of an overall index in the reports involves differential weighting of population trends derived using geometric means. The calculation of the index used involves differential weighting of population trends derived using logged abundance data, geometric means and generalized additive models. The LPI is hierarchically averaged from populations to species, taxa and realm and is also weighted by the estimated and relative number of species within biomes, which influences the direction and magnitude of the index. Our analysis explores the heterogeneity in local trends and fluctuations of monitored species from the raw population abundance data, and thus, we did not use an index with weightings and we did not aggregate population trends to a species-level. Unpacking the weightings behind the LPI and how each weighting influences the net population change of the index is an analysis that we are interested in exploring, but we believe that this analysis would be beyond the scope of this current study.

References

McRae, L. *et al.* The Living Planet Index in: Living Planet Report 2012 (ed. Almond R). WWF, Gland, Switzerland. (2012).

McRae, L., Freeman, R. & Marconi, V. 'The Living Planet Index' in: Living Planet Report 2016: Risk and resilience in a new era (ed. Oerlemans N). WWF International, Gland, Switzerland. (2016).

WWF. Living Planet Report - 2018: Aiming Higher. Grooten, M. and Almond, R.E.A.(Eds). WWF, Gland, Switzerland. (2018).

Revision 3

2. Sceptics might argue that the selected populations don't capture the full range of threats experienced by species. You can only measure population trends on extant populations whereas habitat loss completely removes populations. I would hazard that most species are listed under Criterion A not because populations have been monitored, but because habitat is perceived to have been lost. For this reason, the lack of correlation between IUCN rating and population trend may not be surprising.

We have added new analyses where we test how the number and types of threats that species are exposed to globally relate to local population change. We found that on local scales, there are increasing, decreasing and stable populations across the full spectrum of the globally-determined species' conservation status and anthropogenic threats (IUCN Red List Index). Populations from species that were exposed to a larger number of threats on a global scale based on the species' IUCN threat categorizations did not decline more compared to those categorized with a smaller number of threats (Figure 4f). There were no distinct signatures of global species threats which were associated with predominantly declining local trends of monitored populations (Figure 4e) and there were increasing, decreasing and stable trends across all threat types.

We have also included new text discussing the habitat loss, a topic raised by the reviewer, where we have referenced a new study we did that was not public at the time of submission of this manuscript (Daskalova *et al.*, bioRxiv). We compared population trends across the amounts of forest loss and gain, and the types of habitat transitions that occurred in the specific locations where the population monitoring took place. We have added further discussion about the mismatch between local scale population trends and globally determined species IUCN conservation status. The lack of a correspondence between population change and IUCN extinction risk categories might not be surprising to all readers, but it does test a key assumption made by certain studies in the literature (e.g., Ceballos *et al.* 2017, Betts *et al.* 2017). Population trends are complex and cover a full spectrum from declines to increases. For example, there are declining populations of Least concern species, and increasing populations of Critically endangered species as we discuss in the main text (lines 245-250):

We found increasing, decreasing and stable populations across all Red List categories of conservation status (Figure 4a). For example, a population of the Least concern species red deer (*Cervus elaphus*) in Canada declined by 68% over seven years going from 606 to 194 individuals and a population of the Critically-endangered Hawksbill sea turtle (*Eretmochelys*

imbricate) from Barbados increased by 269% over seven years going from 89 to 328 individuals.

In our study, we highlight that rarity metrics and species' conservation status alone do not capture the full heterogeneity in local population change over time. We discuss how common species should not be overlooked in conservation prioritization decisions as they could be as likely to decrease in abundance over time as rare species.

Reference:

Daskalova, G.N., Myers-Smith, I.H., Bjorkman, A.D., Blowes, S.A., Supp, S.R., Magurran, A., Dornelas M. (2018) Forest loss as a catalyst of population and biodiversity change. bioRxiv <https://doi.org/10.1101/473645>

Ceballos, G., Ehrlich, P. R. & Dirzo, R. Biological annihilation via the ongoing sixth mass extinction signaled by vertebrate population losses and declines. *Proc. Natl. Acad. Sci.* 201704949 (2017). doi:10.1073/pnas.1704949114

Betts, M. G. *et al.* Global forest loss disproportionately erodes biodiversity in intact landscapes. *Nature* **547**, 441–444 (2017).

Revisions 1 and 7

3. Use of 'population size' is confusing and potentially wrong. I think here it means 'the number of animals that were measured by monitoring'. This is different from the ecological concept of a population, which is based on being a relatively close demographic unit. The relationship between these two numbers is likely to vary systematically with body size and mobility. For example, surveying a 1km² may capture an entire ecological population of a small, immobile species, but will only provide a subsample for larger species with greater dispersal distances.

The reviewer is raising a valid point and we agree. We have added a definition of how we use population size the first time we mention the term in both the main text and the methods (see lines 128-130 and 460-461). As the reviewer suggested, we used "the number of animals that were measured by monitoring" when we first mention population size, and we used "population size" from that point onwards because the full definition would be too lengthy to use in all cases across the manuscript:

We analyzed vertebrate population time-series from the Living Planet Database (133,092 records) covering the period between 1970 and 2014. These time-series represent repeated monitoring surveys of the number of individuals in a given area (species' abundance over time), hereafter called "populations". We focus on two aspects of population change – overall changes

in abundance over time (population trend) and abundance variability over time (population fluctuations).

Throughout the text, we have also highlighted that our findings only refer to the monitored population sizes encompassed by the Living Planet Database time-series, e.g.:

Across realms, **monitored** vertebrate populations experienced net population increases... (line 153)

Revision 1

4. The results are not interpretable without having read the methods. This is a structural issue that needs to be addressed. It is not clear what the slopes are that are reported in the results. Moreover, the Methods are lacking in key details. For example, the reader isn't told whether analyses were done at species or population level. This lack of detail makes it difficult to understand results, and hindering reproducibility. For example, what does 'scaling' abundance mean (Line 381-382)?

We thank the reviewer for this feedback that has helped us to improve the structure and clarity of our methods. We have included a paragraph outlining our broad methodological framework in the main text, added information in each caption about what the different variables show, and we have restructured the detailed methods section. We have added equations and references justifying our analytical framework and have clarified that the analyses were done at the population level, e.g. lines 132-136:

In the first stage of our analyses, we quantified trends and fluctuations for each population using state-space models that account for observation error and random fluctuations³⁶ (Figure S1). In the second stage, we modelled the population trend and fluctuation estimates from the first stage across latitude, realm, biome, taxa, rarity metrics, phylogenetic relatedness, species' conservation status and threat type using a Bayesian modelling framework (Figure S2).

And later on when we discuss our results in light of the Living Planet Index (lines 401-403):

Our analysis explores the heterogeneity in local trends and fluctuations of monitored species from the raw population abundance data, and thus, we did not use an index with weightings and we did not aggregate population trends to a species-level.

We visualised our workflows through two conceptual diagrams in the Supplementary Information (Figures S1 and S2).

We scaled the population data to be between 0 and 1 to analyse within-population relationships and to make sure that we were not conflating within-population relationships and between-

population relationships (see van de Pol and Wright 2009 Animal Behaviour). We consider this approach to be appropriate for our analyses because the Living Planet Database includes thousands of populations measured using many different methods and units.

Reference:

van de Pol, M. & Wright, J. A simple method for distinguishing within- versus between-subject effects using mixed models. *Anim. Behav.* **77**, 753–758 (2009).

Revisions 2 and 5

Minor comments

- Introduction: Try some other words in place of However, Yet, and Thus.

We have revised the introduction, as well as the text and flow of the whole manuscript to avoid to many of the same linking words and to prevent the text from sounding too repetitive.

- Line 43: Fundamental (not fundament)

Corrected.

- Line 62: “However, although...” is just bad English

Corrected.

- line 65: the definition of “population change” is not clear at this point. Readers could understand this as synonymous with trend, or as something broader. Trend I think is clear: a directional change in population size, but change is more vague and ambiguous. Conceivably “trend” could be interpreted as special case of “change”, along with “fluctuations” (which is introduced later): other types of change might include a change in the frequency of population cycles, or a directional trend in interannual variance in numbers¹ (independent from a trend in numbers). Similarly, they use both “characteristics” and “metrics” when referring to the trio of geographic range size, habitat specialism and population size.

We have revised our text to be consistent throughout, using rarity “metrics” when referring to geographic range, mean population size and habitat specificity, and we have added definitions for the two types of population change we study (see lines 130-132). We have included the following definitions of population change and rarity:

We focus on two aspects of population change – overall changes in abundance over time (population trend) and abundance variability over time (population fluctuations).

We defined rarity following a simplified version of the ‘seven forms of rarity’ model⁷², and thus consider rarity to be the state in which species exist when they have a small geographic range, low population size, or narrow habitat specificity.

Revision 1

- Lines 65-81: This is a critical paragraph that is potentially misleading to readers. The paragraph is framed in terms of population change, but all the examples are about traits correlated with extinction risk (i.e. a species-level metric, not a population metric). This is confusing. Also, whilst the Rabinovitz paper is a nice way to introduce the concepts, I’m not convinced that “rarity” is a useful theme around which to pin these analysis, for two reasons. First, species can be listed as at risk of extinction due to a rapid decline (criterion A) without meeting any of the seven definitions of rarity, so it’s not clear that one would expect any kind of “generalizable relationship” (line 78). Second, my interpretation is that “rarity” is an outdated concept that was killed-off by Rabinovitz. If you’re interested in the traits associated with change at the population level then there is already a literature^{1–4} that is more relevant than the citations on correlates of extinction risk.

We have revised this paragraph to reflect the focus of our study, which is on population trends, and not on quantifying the correlates of extinction risk, and we have removed the references that were not relevant. The introductory paragraph now reads:

Alterations to ecosystems currently unfolding all around the globe are modifying the abundances of the different species forming Earth’s biota. As global change continues to accelerate^{1,2}, the need for comprehensive assessments of the factors explaining the variety of ecological changes observed across taxa and biomes is also growing³. However, existing empirical studies of the predictors of population change mostly focus on either specific taxa⁴ or on population declines alone^{2,5}. A critical research challenge is to disentangle the sources of heterogeneity across the full spectrum of population change. Recent compilations of long-term population time-series, extensive occurrence, phylogenetic, habitat preference and conservation status data^{6–8} provide a unique opportunity to quantify population trends and fluctuations among the world’s well-monitored vertebrate species and test which species- and population-level attributes explain variation in population change. Population change is the underlying process leading to community reassembly⁹ and the resulting changes to biodiversity are vitally important for ecosystem functions and services¹⁰.

Additionally, we have improved the structure of the whole manuscript to more clearly focus on population change and have distinguished population declines from species extinctions (see lines 104-106, 362-364, 384-387):

Testing population change across species' IUCN conservation status allows us to link contemporary changes in abundance with long-term probability of extinction³⁴.

...

While phylogenetic clustering might be lacking in contemporary trends, there is evidence that phylogenetic relatedness predicts extinction, a process occurring over much longer time scales^{6,7}.

...

We found that species with small populations were, nevertheless, more likely to fluctuate (Figure 3f), which may increase their probability of extinction, a process that could play out over longer time-scales than found for most population monitoring time-series^{22,23,54}.

Revision 1

- Fig 1 legend does not stand alone

We thank the reviewer for bringing this to our attention. We have added further detailed information in the captions for all figures so that the figures can stand alone.

- Fig 2 & 3: Clarify whether these are credible intervals on the mean estimate, or across all species?

We have revised all of our figures to reflect the new structure of the manuscript and improve flow, and for each figure, we have specified what the points and their errors show.

- Line 51: Why is extinction risk defined here when it is being discussed before (eg Line 41)
- Line 86: Mentions previous studies but doesn't provide reference

We have fully revised our introduction and have made sure that definitions appear when the terms are first introduced, and all statements are backed up with references.

- Line 199: There is a literature on Taylor's power law that defines the expected relationship between population size and the magnitude of fluctuations. Please place this result (and the methods used to test it) within that literature.

We have linked our study with Taylor's power law (see lines 83-86):

As per population dynamics theory^{22,23} and Taylor's power law²⁴, species with small populations are more likely to undergo stochastic fluctuations that could lead to pronounced declines, local extinction and eventually global species extinction⁵.

- - Line 262: "Thus, although..." is grammatically okay but doesn't read well

Corrected.

- Line 376: indicate whether this is ALL the LPD data

We have clarified that we are working with the publicly available Living Planet Database. There are newer versions of the LPD that we do not have access to which are housed with the ZSL and WWF.

- Line 441: Mentions a software package but it is not clear that R is being used until line 543

We have moved the mention of the software we used at the start of the methods (R and its specific version).

- Line 446: Why is the model not presented? This text is not enough to reproduce.

We thank the reviewer for highlighting the need for further details in the methods. We have added details on our methods in the main text, model equations and further clarifications to the methods section. We have provided definitions and references for all metrics of population trends and fluctuations we use, as well as references supporting our analytical framework. For additional clarity, we have included two workflow diagrams in the Supplementary Information outlining the two stages of our analyses, the input data for each statistical test, the model and prior structure and outputs. With these revisions, we believe that our analyses will be entirely reproducible, particularly if readers also make use of our code and data repository.

All code and data for our analyses are available on GitHub for review. To access the private repository, go to <https://github.com>, log in using username *PopChangeRarity* and password *atimeofchange9* and then navigate to <https://github.com/gndaskalova/PopChangeRarity>. We will make all code and data publicly available at time of publication with links and DOIs at the time of publication.

Revisions 2 and 5

REFERENCES

1. Oliver, T. H., Roy, D. B., Brereton, T. & Thomas, J. A. Reduced variability in range-

edge butterfly populations over three decades of climate warming. *Glob. Chang. Biol.* 18, 1531–1539 (2012).

2. Mace, G. M., Collen, B., Fuller, R. A. & Boakes, E. H. Population and geographic range dynamics: Implications for conservation planning. *Philos. Trans. R. Soc. B Biol. Sci.* 365, 3743–3751 (2010).

3. Cowlshaw, G., Pettifor, R. A. & Isaac, N. J. B. High variability in patterns of population decline: the importance of local processes in species extinctions. *Proc. Biol. Sci.* 276, 63–9 (2009).

4. Isaac, N. J. B. & Cowlshaw, G. How species respond to multiple extinction threats. *Proc. Biol. Sci.* 271, 1135–41 (2004).

We thank the reviewer for providing further useful references. We have included the following reference from the reviewer's suggestions:

Isaac, N. J. B. & Cowlshaw, G. How species respond to multiple extinction threats. *Proc. R. Soc. Lond. B Biol. Sci.* **271**, 1135–1141 (2004).

We are already at the citation limits and cannot include all possible citations, but we have incorporated references that focus on large-scale comparisons using similar metrics to our analysis.

Reviewers' Comments:

Reviewer #2:

Remarks to the Author:

The authors have done an impressive revision, and many of the issues that prevented a reasonable assessment have now been made clearer. However, this added methodological detail has revealed a particular flaw that was not obvious when I reviewed the first version.

1. As a result, I strongly suspect that the major results are merely a statistical artefact rather than revealing biological meaning. The issue is this — despite claiming that "... at least five time points of monitoring data ... considered a long enough period to detect a directional population trend (if one exists) ..." is unfortunately completely wrong. The authors cite an unpublished and non-peer-reviewed paper as a justification, but there is sufficient evidence now to the contrary that this is entirely inadequate to determine trends. Indeed, all population data should ideally be expressed in terms of generation time rather than years or 'time points', because statistically speaking, the demographic-rate processes of any species are allometrically scaled. Thus, for the expression of the crude emergent property of these processes that is abundance to reflect real trends, sufficient monitoring of enough generations is essential.

In fact, density-feedback processes are only truly revealed when the number of generations monitored is in the order of e^4 (i.e., $\sim > 50$ generations) ([http://dx.doi.org/10.1890/0012-9658\(2006\)87%5B1445%3ASOEFDD%5D2.0.CO;2](http://dx.doi.org/10.1890/0012-9658(2006)87%5B1445%3ASOEFDD%5D2.0.CO;2)). Thus, most of the LPI data most likely reflect a mere phasic equilibration to a moving-target carrying capacity (K) rather than true trending (in any direction). This now explains the observation that most time series had no evidence for a trend — entirely an expected result from an emergent statistical property.

Now, this isn't necessary fatal, but you cannot simply designate an arbitrary (and entirely inadequate) threshold beyond which the results magically obtain biological meaning. Instead, the length of the time series itself could be explicitly incorporated into the trend models and the statistical dependency could at least in part be taken into account quantitatively.

I therefore simply do not believe the majority of the results presented.

2. The authors' dismissal of the very real problem that most IUCN determinations are not related to population decline per se (Criterion B and above), and that only Criterion A (population decline) is relevant, means that comparison to IUCN threat categories across all criteria is entirely inappropriate. Their justification for ignoring this important issue is unconvincing, and so the important conclusions drawn from the 'threat risk' analysis (cf. extinction risk) are also probably wrong. Unless the authors focus exclusively on the Criterion A species, this analysis cannot be justified. On that note too, you cannot simply conclude that threat risk and extinction risk are the same thing, as I originally highlighted in the first review.

3. The problem of the small vs. declining population paradigms being conflated remains. Small populations are often small for entirely different reasons for what drives populations into decline. In fact, most species are rare (the rarity of commonness, popularised by Kevin Gaston in many papers), so if they are naturally rare, or even if they have become rare, there is no expectation whatsoever that declines will be higher in such species. Thus, rarity (i.e., smallness) is not an adequate hypothesis to test for an effect on trending, even if the trend data are statistically robust (which in this case, mostly are not).

Reviewer #3:

Remarks to the Author:

The authors have done a thorough and good job of responding to the comments from all three reviewers. This revised manuscript is a great improvement and I would really like to see this published. There are several remaining issues to address: three in direct response to the rebuttal, and the remainder are minor additional issues.

The addition of source code on a Github repository is welcome, but I was unable to access it. I logged in with the instructions provided, but received a message about an authentication code. I never received such a code so I assume it's gone to the lead author's email address. Therefore I have been unable to look at the source code. I should like to have an opportunity to review the code, as has been promised, before recommending that the article is suitable for publication.

Early parts of the main text make predictions in the past tense (e.g. "we expected" on lines 72-3 and "we predicted" on line 80). These statements don't make sense, because it's still the introductory phase of the article. It's not until line 111 that you define the focus and aims of the investigation. So reframing the introduction of these predictions would make the article more internally consistent.

Re my comment about the contrast with the LPI headline finding: I'm pleased to note the new paragraph starting at line 392. I was slightly disappointed, however, that the discussion focussed only on the difference between this study and the LPI headline. Readers will want to know which is "real". The current discussion raises more questions than it answers. What sorts of factors might explain the differences? For example, could it be that in averaging away the infraspecific variation in trends creates a bias in the headline LPI? Could this be a function of the scaling (from 0-1)? Is the 'no-net change' result an artefact of shrinkage? Or is the headline LPI driven solely by amphibian declines? There are some questions here waiting to be posed (although not answered - that is beyond the scope of investigation).

Reviewer 2 made a point about including estimates of uncertainty on the abundance estimates. This is another example where the authors dismissed the point on the grounds of "analysis not possible", rather than engaging seriously with the criticism. I think the reviewer is right that such error could be important, but I also agree with authors that such data are not available and that it's unreasonable to expect them to find it. However, they really ought to acknowledge in the discussion that this uncertainty is important: it could lead to biases in large scale analyses and estimates thereof be made available wherever possible. I also note that in principle one could model this uncertainty without actually having data for it. In a state space model, it's relatively straightforward to state that:

$Y_{\text{obs}} \sim \text{dnorm}(Y_{\text{true}}, \tau_{\text{obs}})$

$\tau_{\text{obs}} \leftarrow 1 / \sigma_{\text{obs}}^2$

where Y_{obs} is the observed data, Y_{true} is the unobserved truth, and σ_{obs} represents measurement error and/or sampling error. Currently you assume that $\sigma_{\text{obs}} = 0$, but you can ask the model to estimate the value of σ_{obs} that best fits the data. I'm not seriously recommending the authors undertake such an analysis, but acknowledging the possibility would be worthwhile.

The van Strien et al (2016) paper would be worth citing here: the rebuttal gives the false impression that van Strien et al (2016) tried and failed to incorporate uncertainty in the species indices into their Dutch LPI: in fact they incorporated one important component of this uncertainty (the sampling error).

line 392: "our finding that all is not declines" - it took me several reads to understand this. Please rephrase, something like "Our finding that declines are not universal, or even predominant"

Reviewer #4:

Remarks to the Author:

GENERAL

I will confess to being overwhelmed by the volume of information in this paper, the responses to the reviewers, and the supplemental materials: all told, 141 pages of materials and many dozens of complex figures and arguments. The authors have taken on a very important topic, and clearly devoted substantial efforts to addressing it, and I have mixed feelings and limited time in which to adjudicate between the reviewers and the authors.

There are two remaining central issues to tackle in this paper: (1) how many increasing and decreasing time series would you expect given the underlying data if there was no trend at all in the actual populations? i.e. what is the null distribution against which the results are compared? (2) Are the results correct and valid?

For point (1) some calculation needs to be made about how many time series are significantly different from zero. Perhaps this is exactly what the authors have done when reporting that 15% are declining and 18% are increasing, but this is not stated explicitly in the manuscript. The second part of this is what is the power to detect trends in abundance, which is what reviewer 2 is honing in on.

For point (2) there are some odd anomalies in the estimates, with bimodal distributions of values, and many duplicate estimates of μ as outlined below. This needs to be investigated, and either corrected or explained in the text. My initial reaction is that there is some kind of failure in model convergence where the final answer ends up on a bound or equal to the starting values.

When these two issues can be sorted out, the paper should be published.

SPECIFIC COMMENTS

The phrase "all is not declines" is used in the title and throughout. This is grammatically incorrect and needs to be changed, perhaps to "Not all populations are declining" or "Nearly equal numbers of declining and increasing populations" or "Increasing populations are slightly more common than declining populations".

Figure 1d: there is a problematic trend in the estimates of μ , in that there are bands where multiple species have exactly the same value of increase (+0.2) or decrease (-0.2), indeed zooming in there are additional bands at exactly +0.025 and -0.025, and at +0.1, +0.15, etc. Since each population should be independent, this suggests one of the following issues: (1) the model has a starting point at fixed values, and failed to converge to the true answer, resulting in multiple data points with exactly the same result; (2) the same population appears multiple times in the dataset resulting in duplicate estimates; (3) some kind of numerical issue related to model fitting to integer values (hard to think of a scenario). Regardless, this needs to be thoroughly investigated to ensure that the results are not artefacts of the data or fitting process, especially since the bimodal distributions in freshwater and terrestrial realms are entirely driven by the duplicate estimates of μ at +0.025 and -0.025. Similar issues are visible in all the other figures and in Figure S8. Perhaps there is a file drawer effect towards not publishing data if there is no trend ($\mu = 0$), explaining the missing peak at $\mu=0$? The cause of the issue appears to lie in the index values (Fig. S13).

Figure 1f and other plots of σ^2 : the y-axis should be plotted on a log scale so that it is possible to compare variance across realms. An alternative is to plot the coefficient of variation, $CV = \sigma/\mu$, which is the proportion of error, and commonly used to compare values that have different means.

lines 463-465: Reviewer 2 is correct in being skeptical that five time points are sufficient to detect population trends in 80-90% of cases. Note, however, that ref 68 is now published in *Methods in Ecology and Evolution* (<https://besjournals.onlinelibrary.wiley.com/doi/abs/10.1111/2041-210X.13302>). The key is that there are two items being measured: the magnitude of the trend, and the significance of the trend. The significance depends on the number of data points being fitted, and the variability in the data points being fitted. For short time series, most trends will be non-significant and will also have magnitudes that vary widely—the classic cone shape in meta-analysis studies seen in Figure S6. Thus it is more important in this paper to highlight how many populations are increasing or declining significantly (at some level of alpha), than to focus on the vast majority of populations with no detected trend.

I disagree with reviewer 2 that 50 generations is relevant to this paper. The paper is not looking to estimate density-dependent effects, only trends over time. For some species, like rockfish (*Sebastes*) with generation times of 30 yr, it is obviously not reasonable to require 1500 yr of data before trends in abundance can be estimated. With overlapping generations, small sample SEs, and large changes in abundance, it can easily be possible to detect trends with 3 data points.

The point that reviewer 3 makes about uncertainty in the abundance estimates is I believe covered by the state-space models estimating both observation and process error.

Response to reviewers

Rare and common vertebrates span a wide spectrum of population trends
Gergana N. Daskalova, Isla H. Myers-Smith and John L. Godlee

Reviewer #2 (Remarks to the Author):

The authors have done an impressive revision, and many of the issues that prevented a reasonable assessment have now been made clearer.

We thank the reviewer for the positive assessment of our revisions.

However, this added methodological detail has revealed a particular flaw that was not obvious when I reviewed the first version.

Response to reviewer 2 - 1:

We provide new summary statistics and figures (Figures S6-S8, S13) in this response to reviews and in the supp. info. to further explain why our results are not a statistical artifact - detailed below. We respectfully disagree with the statement that there is a flaw in the methods as described. We carefully describe our methods below and the rationale behind why our methods are rigorous and are in keeping with other similar studies with time series data of species' wild populations. We have devoted a lot of time to investigating trend analyses and their sensitivities to study duration and number of monitoring time points with these data and other types of monitoring studies that directly address the issues raised by reviewer 2. We have revised our manuscript text to better communicate our perspectives on our chosen analyses. There is a need for further research with global vertebrate population dynamics, but a very detailed investigation of population dynamics and sampling within and beyond the Living Planet Database is beyond the scope of our current study, and we feel detracts from our overall aim to test how population trends within the Living Planet data vary across biomes, taxa and rarity traits. Please see our detailed further thoughts on this issue below.

1. As a result, I strongly suspect that the major results are merely a statistical artefact rather than revealing biological meaning. The issue is this — despite claiming that "... at least five time points of monitoring data ... considered a long enough period to detect a directional population trend (if one exists) ..." is unfortunately completely wrong.

Response to reviewer 2 - 2:

The point that we were attempting to make with this now revised statement (see quote below) is that any time series length cut off is arbitrary, but that there is justification for removing very short time series (fewer than five years) and time series with very few monitoring points (fewer than five time points). Any trend analysis will be subject to the sample size and duration of the data, but also many other biological factors such as the generation time of the organism, the rate of reproduction, the seasonality of the study system, etc. These constraints apply to many ecological datasets that include multiple taxa and not just population data. All of these types of constraints

cannot be fully captured in any current statistical approach. However, duration, number of time points, taxonomic, phylogenetic and cross-biome differences are factors that we have carefully addressed in our study and factors that are often not captured in similar studies to the same degree. We believe our study contributes an advance beyond the current state-of-the-art and also stimulates important questions for future research with these data and other large-scale monitoring data compilations.

The authors cite an unpublished and non-peer-reviewed paper as a justification, but there is sufficient evidence now to the contrary that this is entirely inadequate to determine trends.

Response to reviewer 2 - 3:

Our manuscript was in review for over nine months, and thus pre-prints that were cited have now become published. It is great to see a number of recent papers get published in this area of research, which strengthen the justification for the approaches we have taken in our study.

The full original quote from our manuscript was:

“In our analysis, we retained only populations that had at least five time points of monitoring data, as that is considered a long enough period to detect a directional population trend (if one exists) in 80-90% of cases⁶⁸.”

Here we were referring to the results of the Wauchope et al. 2019 study published in the journal *Methods in Ecology and Evolution*:

Wauchope, H.S., Amano, T., Sutherland, W.J. and Johnston, A., 2019. When can we trust population trends? A method for quantifying the effects of sampling interval and duration. *Methods in Ecology and Evolution*, 10(12), pp.2067-2078.

<https://besjournals.onlinelibrary.wiley.com/doi/abs/10.1111/2041-210X.13302>

In the quote, we refer to the results presented in the Wauchope et al. 2019 study and not to our own work. We have now revised the sentence to make it clear that the sentence refers to the Wauchope et al. 2019 study:

“In our analysis, we omitted populations which had less than five time points of monitoring data, as previous studies of similar population time series to the ones that we have analysed have found that shorter time series might not capture directional trends in abundance⁶⁸.” (lines 473-475)

We also cite:

Fournier, A.M., White, E.R. and Heard, S.B., 2019. Site-selection bias and apparent population declines in long-term studies. *Conservation Biology*, 33(6), pp.1370-1379.

<https://conbio.onlinelibrary.wiley.com/doi/full/10.1111/cobi.13371>

This recent methods study in Conservation Biology also investigates the detection of trends in population dynamics time series studies. In Fournier et al. 2019, the authors recommend testing the sensitivity of results to the start and end point population values for each time series. From our study:

“Following Fournier et al. 2019⁶³, we tested the time-series that we analyzed for site-selection bias. Removing the first five survey points reduces the bias stemming from starting population surveys at points when individual density is high, whereas removing the last five years reduces the bias of starting surveys when species are very rare. The distribution of population trend values across time-series was not sensitive to the omission of the first five (left-truncation) or the last five years (right-truncation) of population records (Figure S6).” (lines 793-798)

Another recent relevant paper that we now cite is Didham et al. 2020:

Didham, R.K., Basset, Y., Collins, C.M., Leather, S.R., Littlewood, N.A., Menz, M.H., Müller, J., Packer, L., Saunders, M.E., Schönrogge, K. and Stewart, A.J., 2020. Interpreting insect declines: seven challenges and a way forward. *Insect Conservation and Diversity*, 13(2), pp.103-114. <https://onlinelibrary.wiley.com/doi/full/10.1111/icad.12408>

Didham et al. 2020 identify seven key challenges in drawing robust inference about insect population declines, which are also relevant to the analysis of population trends in vertebrate populations including: establishment of the historical baseline, representativeness of site selection, robustness of time series trend estimation, mitigation of detection bias effects, and ability to account for potential artefacts of density dependence, phenological shifts and scale-dependence in extrapolation from sample abundance to population-level inference.

“Thus, the detected trends and the baseline across which trends are compared might be influenced by when monitoring takes place and at what temporal frequency. These challenges are present across not just the specific vertebrate population time series we analysed, but more broadly across population data, including invertebrates⁸⁸.” (lines 715-719)

The concepts discussed in Wauchope et al. 2019, Fournier et al. 2019 and Didham et al. 2020 are broadly generalisable to any time series analysis.

Statistically, the likelihood of detecting a trend is related to the number of time points analysed and as the number of time points increases, the probability of detecting a trend by chance decreases. Another indicator of whether a trend has been detected by chance is the error around the trend relative to the trend itself. By incorporating the error around the population abundance estimates in two different ways into the analyses themselves, we partially account for detecting trends by chance in our own work. First, we used the state-space framework, which is particularly suitable for analysing data with varying properties (e.g., in our case monitoring methods, duration, time points varied across time series). This modelling approach partitions error around the trends in measurement error and process noise. Second, we repeated our second-stage analyses (e.g., population trends across taxa, biomes, etc.) using weighted regressions, with the weight being

the observation error (τ) around the population trend estimate, and found similar results. So while we are not able to incorporate error at the initial raw data stage (i.e., we have only one value for say tree sparrow abundance in 1972, thus we can't calculate error around that, see Response to reviewer 3 - 7), we are able and we have incorporated error at the level of the population trend over time.

Indeed, all population data should ideally be expressed in terms of generation time rather than years or 'time points', because statistically speaking, the demographic-rate processes of any species are allometrically scaled.

Response to reviewer 2 - 4:

We agree that ideally population data should be expressed in terms of demographic-rate processes including generation time. However, the Living Planet Database does not have this level of granularity or precision. The ideal monitoring frequencies will be taxa specific and will relate to the organisms' biology and the temporal dynamics of the study system.

In the version of the Living Planet database we worked with, 17.9% of populations were sampled annually or in rare cases multiple times per year. The LPD currently represents vertebrate species that span a large variation in age, generation times and other demographic-rate processes. For example, from other work that we have conducted, when generation time data are available for species within the Living Planet Database, we find for birds (approximately 50.0% or 484/968 of the LPD bird species) the mean generation time is 5.0 years from a minimum of 3.4 years to a maximum of 14.3 years and for mammals (15.6% or 48/306 of the LPD mammal species) the mean generation time is 8.3 years from a minimum of 0.3 years to a maximum of 25 years (Daskalova et al. Science accepted). Thus, we believe that most vertebrate time series within the LPD capture multiple generations. These details have been added to the manuscript text on lines 464-471.

In our analysis of nearly 10,000 abundance time-series from over 2,000 vertebrate species, species- and habitat-specific parameters of demographic-rate processes cannot easily be estimated, let alone incorporated into analyses. Assuming variation amongst taxa or study systems without data to inform those assumptions could incorporate substantial bias into subsequent analyses.

We were very careful in our work to remove any potential bias into the analysis and to treat all species and data from different study systems consistently using rigorous statistical methods as advocated for in recent studies such as Wauchope et al. 2019, Fournier et al. 2019 and Didham et al. 2020.

We also analysed phylogenetic patterns in trends within these data finding no strong phylogenetic patterning in population change. The phylogenetic relationships may sometimes capture similarities in demographic-rate processes. Further work incorporating demographic-rate processes could be conducted in future, but are beyond the scope of our current study. We know from other work we have conducted that there is very small overlap between the species included

in the Living Planet Database and the species in demographic databases (<10% of LPD mammal species have publicly available demographic data), such as COMADRE (<https://compadredb.wordpress.com>). As more demographic data become publicly available, we will be better poised to synthesise population and demographic information in future studies.

In our study, we have investigated the influence of duration on trends detected. These results are reported in Figures S7-S8. Here we summarise our additional analyses on duration conducted in our original submitted manuscript and through the previous and current round of revisions in point form:

- In the Living Planet Dataset, the number of survey points within time-series positively correlates with time-series duration (Figure S8), with 17.9% of populations being sampled annually or more frequently.
- In an analysis of population trend versus duration, both positive and negative vertebrate population trends were smaller in magnitude for longer time-series of data; however, there were similar numbers of increasing and declining populations (Figure S7, now updated with distributions of population trends across different duration categories).
- Left and right-truncation as per Fournier et al. 2019 didn't influence the finding of similar numbers of increasing and declining populations (Figure S6).
- We have conducted a model of population change as a function of both taxa and study system with an interaction with duration (Table S2) and the credible intervals around the posterior mean for the interaction effects overlapped zero, indicating for the system model results are not sensitive to duration within the Living Planet Dataset, with the exception of reptile populations, where longer durations have more negative trends. These new results have been added to the supp. info. in the following figures and text:

Figure S7. Both positive and negative vertebrate population trends were smaller in magnitude for longer time-series of data. Monitoring duration results are for 9286 populations from 2084 species. Population trends (μ) were estimated for all populations monitored for more than five time points using state-space models (a, d) and linear models (b). Population fluctuations (c) are plotted on a log₁₀ y axis and represent the estimates for process noise (σ^2 , the process noise is the total variance around the population trend minus the variance attributed to observation error) derived from state-space models. Error bars on (a) and (b) show 95% confidence intervals. The sample sizes for the duration categories were as follows 5 - 10 years: 2084 time series; 10 - 25 years: 3358 time series; 25 - 44 years: 3844 time series. Plot (e) shows the raw population trend data behind 12 time series which had the same population trend values ($\mu = 0.20$). These time series are part of a “band” of time series which had very similar population trend estimates. Eighty, or approximately 1% of the time series we analysed form linear relationships over time with errors around the slopes of <0.001 , such that we suspect these data are modelled rather than measured population data. The presence of modelled data within the dataset may help explain the low variance bands of σ^2 values (c) and the pattern of two peaks in weak population increases and decreases for longer time series (d). Please see Methods sections “Time series with low variation” and “Clustering in the values of population trends and fluctuations” for further details.

“Monitoring duration was weakly positively related to vertebrate population trends, with slightly greater population increases found for longer duration studies (Figure S7, Table S2). There was a similar weakly positive effect of number of time points within time series (Table S2). Additionally, we tested if monitoring duration influenced the relationships between population trends across systems, and population trends across taxa. We found that duration did not influence those relationships, with the exception of reptiles, where declines were more frequent as monitoring duration increased (Table S2).” (lines 784-790)

Thus, for the expression of the crude emergent property of these processes that is abundance to reflect real trends, sufficient monitoring of enough generations is essential.

Response to reviewer 2 - 5:

Across the Living Planet Database, we analysed 9286 population time-series from 2084 species, with a mean duration of 23.9 years and a mean sampling frequency of 23.3 time points. It is our assessment that these data represent sufficient monitoring to speak to trends within the Living Planet Database.

We are careful in our manuscript not to imply that this analysis reflects anything more than the sample of data that we are working with from sites around the world. This is directly in contrast to many other studies using the Living Planet Database that imply that this sample is broadly representative of population change across the planet. However, we do believe that this unique and comprehensive dataset is the largest compiled population dataset available for vertebrate populations globally, and thus represents an important database to explore questions of what explains variation in population trends across specific biomes, taxa or types of species within the

dataset - the aim of our study. Here are some example quotes where we highlight that our findings refer only to the time series we analysed:

“Across the time series we analysed, 15% (1381 time series) of populations were declining, 18% (1656 time series) were increasing, and 67% (6249 time series) showed no net changes in abundance over time, in contrast to a null distribution derived from randomised data (Figure S6b).” (lines 146-149)

“Taken together, our analysis of nearly 10 000 vertebrate population time series using a state-space modelling approach demonstrated ubiquitous alterations in vertebrate abundance over time across all biomes on Earth.” (lines 297-299)

“We found that for the populations and species represented by current monitoring, rarity does not explain variation in population trends, but we note that the relationship between population change and rarity metrics could differ for highly endemic specialist species or species different to the ones included in the Living Planet Database⁸⁹.” (lines 682-685)

In fact, density-feedback processes are only truly revealed when the number of generations monitored is in the order of e^4 (i.e., $\sim > 50$ generations) ([http://dx.doi.org/10.1890/0012-9658\(2006\)87%5B1445%3ASOEFDD%5D2.0.CO;2](http://dx.doi.org/10.1890/0012-9658(2006)87%5B1445%3ASOEFDD%5D2.0.CO;2)). Thus, most of the LPI data most likely reflect a mere phasic equilibration to a moving-target carrying capacity (K) rather than true trending (in any direction). This now explains the observation that most time series had no evidence for a trend — entirely an expected result from an emergent statistical property.

Response to reviewer 2 - 6:

We agree that the Living Planet Data represent a snapshot in time and only a sample of species and locations. It was never our intention with these data to representatively capture all real-world vertebrate populations, but rather to ask what are the population trends within this global and comprehensive dataset. Based on our analyses of these data, we cannot make a statement as to whether the Living Planet Database data “reflect a mere phasic equilibration to a moving-target carrying capacity (K) rather than true trending (in any direction)” - any planet-wide “true trending” are rather claims that other publications may make using the LPD.

For our study, these data are merely a large dataset within which we can ask whether there are increasing or decreasing trends within the duration and time points captured and what might explain why certain species have or not have trends within the available data. One interpretation of our findings of no net population trends, despite other studies with the LPD indicating otherwise, is indeed that variable temporal dynamics are being detected rather than long-term directional change. We have revised the discussion to better communicate this point as well as added to the Data Limitations paragraphs in the Methods (Taxonomic and geographic gaps, Monitoring extent and survey techniques, Temporal gaps, Time series with low variation, Clustering in the values of population trends and fluctuations)..

“Taken together, our analysis of nearly 10 000 vertebrate population time series using a state-space modelling approach demonstrated ubiquitous alterations in vertebrate abundance over time across all biomes on Earth. We revealed that population change includes both increasing and decreasing populations and spans a wide spectrum of magnitudes, and while anthropogenic impacts have accelerated in recent decades, our results highlight that vertebrate species span a wide spectrum of population change.”

“The magnitude of population trends could be influenced by how long populations are monitored⁶¹, as well as whether monitoring began during a population peak or a population trough⁶². While overall, we did not find a strong effect of duration on the detected population trends in the Living Planet Database (Figures S7-8, Table S2), our findings demonstrated that for reptiles, time series with longer durations are more likely to capture declines (Table S2). We also found a bimodal pattern of weak population increases and decreases in time series with longer durations particularly for terrestrial bird species with the monitoring unit being an index (Figure S13). Seven key challenges have been identified when drawing robust inference about population trends over time: establishment of the historical baseline, representativeness of site selection, robustness of time series trend estimation, mitigation of detection bias effects, and ability to account for potential artefacts of density dependence, phenological shifts and scale-dependence in extrapolation from sample abundance to population-level inference⁶³. New methods to rigorously account for different sources of uncertainty in time series monitoring will allow the analyses of available population data to better inform global estimates of net trends across taxa.”
(lines 395-408)

Now, this isn't necessarily fatal, but you cannot simply designate an arbitrary (and entirely inadequate) threshold beyond which the results magically obtain biological meaning. Instead, the length of the time series itself could be explicitly incorporated into the trend models and the statistical dependency could at least in part be taken into account quantitatively.

Response to reviewer 2 - 7:

It was never our intention to suggest that an arbitrary threshold creates biological meaning, but rather that by removing data that do not meet our minimum duration and time point criteria we increase the chance of detecting trends rather than noise within the available time series data. In our revised manuscript, we include an analysis where we compare the population trends we detected to a null distribution derived from a randomisation of the data (see Figure 6b and also attached below). We found that there are more population increases and decreases as would be expected by change. The question of whether the detected statistically significant trends represent real-world population dynamics beyond the taxa and study locations in this dataset is a separate point and one that we already address within the study, but to which we have carefully emphasised in response to this series of comments.

a Left- and right-truncation

b Randomisation and null hypothesis

Figure S6. The distribution of population trend values across time-series was not sensitive to the omission of the first five (left-truncation) or the last five years (right-truncation) of population records and it differed from a null distribution derived from randomised data. Following Fournier et al. 20194, we tested the time-series that we analysed for site-selection bias. Removing the first five survey points reduces the bias stemming from starting population surveys at points when individual density is high, whereas removing the last five years reduces the bias of starting surveys when species are very rare. There were slightly fewer trends centred on zero (no net change in abundance over time) when we left- and right-truncated the data, suggesting that longer time-series are more likely to detect no net changes in abundance (see Figure S7 for a visualization of population trends versus monitoring duration. We also compared the distribution of estimated population trends against a null hypothesis (b). To derive a null distribution, we used a randomisation approach. Within each time series, we randomised the abundance data, keeping the overall range of the original data. The two peaks of μ are apparent in the overall distribution of time series data. These peaks are created by many weakly positive and negative population trends from longer time series that often are bird species from terrestrial systems. We hypothesise that there might be a publication bias against no net change studies, or a bias against including such studies in global databases.

I therefore simply do not believe the majority of the results presented.

Response to reviewer 2 - 8:

We would clarify that the results of this study refer to the actual data analysed. We too do not have confidence in extrapolating beyond these data. However, we do have confidence that we are detecting statistical trends that are both positive and negative within the available time series. In this study, we set out to analyse those dataset-specific trends and ask if there were covariates that explained why some species were increasing versus decreasing within the LPD duration. Our headline findings from the abstract are:

"We found that 15% of populations declined, 18% increased, and 67% showed no net changes over time. Against a backdrop of no biogeographic and phylogenetic patterning in population change, we uncovered a distinct taxonomic signal. Amphibians were the only taxa that experienced net declines in the analysed data, while birds, mammals and reptiles experienced

net increases. Population trends were poorly captured by species' rarity and global-scale threats."
(lines 32-36)

We hope that the reviewer can, given the scope of our study, see the value in our results with respect to this sample of vertebrate population time series, even if they do not have confidence that the Living Planet Database represents enough temporal or spatial or taxonomic replication to represent real-world change beyond the extent of the dataset.

Previous response from the last round of revisions addressing this point:

Here we describe our inclusion of the Wauchope et al. 2019 study in the previous round of revisions.

We thank the reviewer for bringing attention to the importance of considering time-series duration. We have added reference to a new study by Wauchope *et al.* that was not available at the time of initial submission of this manuscript, addressing precisely the issue raised by the reviewer (see lines 465 and 695). By explicitly testing how different durations affect the detected trends, Wauchope *et al.* found that in 80-90% of cases when there was a directional population trend, time-series with a 5-year duration or more will capture that directional trend accurately. We have also included a statistical test and visualisations of the effects of study duration on detected population trends across the time-series we studied (see Figures S3 and S6, as well as model outputs for the "Duration" models in Tables S2 and S3). Both positive and negative vertebrate population trends had a smaller magnitude for longer time-series.

Here we describe our inclusion of the Fournier et al. 2019 study in the previous round of revisions.

In our revised manuscript, we have also included a new sensitivity analysis (see lines 727-735). Following Fournier *et al.* 2019⁶³, we tested the time-series we analyzed for site-selection bias. Removing the first five survey points reduces the bias stemming from starting population surveys at points when individual density is high, whereas removing the last five years reduces the bias of starting surveys when species are very rare. The distribution of population trend values across time-series was not sensitive to the omission of the first five (left-truncation) or the last five years (right-truncation) of population records (Figure S6). Overall, our sensitivity analyses confirmed that our findings were robust to the potential confounding effects of differences in monitoring duration, sampling method and site-selection.

2. The authors' dismissal of the very real problem that most IUCN determinations are not related to population decline per se (Criterion B and above), and that only Criterion A (population decline) is relevant, means that comparison to IUCN threat categories across all criteria is entirely inappropriate. Their justification for ignoring this important issue is unconvincing, and so the important conclusions drawn from the 'threat risk' analysis (cf. extinction risk) are also probably wrong. Unless the authors focus exclusively on the Criterion A species, this analysis cannot be justified. On that note too, you cannot simply conclude that threat risk and extinction risk are the same thing, as I originally highlighted in the first review.

Response to reviewer 2 - 9:

We agree that the IUCN classification system of IUCN Red List Categories is complex, but we disagree with the characterisation that we are dismissing a very real problem. We believe if we clarify the terms used to describe the IUCN Red List Categories including “extinction” and “threat” and the question of whether it is circular to include categories that are derived in part from population declines to compare among species population trends this will help to partially satisfy this reviewer.

We were previously using the terminology that the IUCN uses including “extinction” and “threat” for clarity with other literature. From the IUCN website: “The IUCN Red List Categories and Criteria are intended to be an easily and widely understood system for classifying species at high risk of global extinction”. See the IUCN website and publications for further details on their terminology (<https://www.iucnredlist.org/>). For clarity, we will now use the following terms throughout the manuscript:

- 1) IUCN Red List Categories - the categories that the IUCN defines as “species' risk of global extinction”
- 2) threat types or number of threats - the types of threats and numbers of threats listed for each species as per their IUCN Red List profile
- 3) habitat specificity - the number habitats each species is characterised with in their IUCN Red List profile

We want to clarify that we are certainly not dismissing the fact that IUCN Red List Categories are partially based on population data. However the data used in IUCN determination are not necessarily the same population data as analysed here. We do not make any assumption that IUCN determinations are either directly related or not related to population declines in our manuscript or analyses, we simply test whether there are differences in the population change in the LPD among IUCN Red List Categories. Given that we do not find high correspondence between the IUCN Red List Categories and the population time series in the Living Planet Database, there must not be strong circularity across these two datasets.

We are including the IUCN Red List Categories as they have been used in many recent studies investigating population change and conservation (for example Di Marco et al. N. Comms. 2018) and thus we believe the results presented will be of interest to readers. If the editors believe that the reviewer concerns about the IUCN terminology and methods justify the removal of these analyses we can do so, but it is not clear to us that there is an issue here with respect to the findings that we present.

The reviewer's previous comment:

Another major concern is that 'extinction risk' is being inferred constantly throughout this manuscript from 'threat risk'. These are not the same thing. First, the IUCN Red List uses multiple criteria to set relative threat risk (not extinction risk), yet the authors do not distinguish among

Criteria when testing their relationships. There is really only one Criterion here that is relevant - Criterion A justifications (population decline), yet the authors use all Criteria to establish 'extinction risk'. This mixes different drivers and will necessarily dilute the capacity to identify relationships. It is therefore not surprising in the slightest that the authors did not find the relationships they purport to test hypothetically.

Previous response from the last round of revisions addressing this point:

Here in our previous revision comment, we responded to this point in detail describing how extinction risk is the terminology that the IUCN uses and not our chosen terminology.

We agree that threat and extinction are not the same conceptually; however, our goal was not to redefine terms already in use in the literature. In our study, we use the classification provided by the IUCN, which they title "extinction risk" categories. By incorporating the IUCN extinction risk categories, we sought to test how local scale population trends vary across the globally determined Red List species' status. We use the IUCN Red List categories as extinction risk metrics to be consistent with IUCN's terminology - see <https://www.iucnredlist.org> and specifically the definition "The IUCN Red List Categories and Criteria are intended to be an easily and widely understood system for classifying species at high risk of global extinction." The Red List classifications have been frequently used as metrics of extinction risk, for example the recent publication published in the journal Nature Communications by Di Marco *et al.* 2018 (<https://www.nature.com/articles/s41467-018-07049-5>).

Here in our previous revision comment, we outlined how the IUCN Red List categories are determined based on a variety of criteria - we did not make it entirely clear that we recognize that population trends are information that are indeed included in the IUCN assessments, which we now have stated more clearly in our current response and in the manuscript. The population data used in the IUCN assessments may or may not be the same population data including in the Living Planet Database and the fact that there are not clear relationships here is intriguing and may warrant further examination in future studies.

The reviewer is correct that the IUCN Red List categories are determined based on different criteria, or a combination of criteria. Most of our study species were classified based on multiple risk criteria, thus providing an extraction of only Criterion A was not possible. However, we are using these categories in the same ways as other studies and they do provide a test of whether species categorized by the IUCN as globally under threat are experiencing more or less population change, including declines, at local scales. We found increasing, decreasing and stable populations across all Red List categories of conservation status (Figure 4a). In a follow up analysis, we explore land-use and forest cover change specifically (Daskalova *et al.*, preprint available on bioRxiv). Additionally, we are in the process of exploring the cumulative and interactive effects of climate change, land-use change and other types of global change on population trends over time, but that is the focus of another manuscript.

References:

Figure 4. On local scales, there are increasing, decreasing and stable populations across the full spectrum of the globally-determined species' IUCN Red List Category and anthropogenic threat type from the species' IUCN Red List profiles. Numbers in the legend for plots a-d and in plots e-f show sample size for each metric. Plots a and c show the density distribution of population trends across Red List status, the raw values (points) and boxplots with the mean, first and third quartiles and boxplot whiskers that indicate the distance that covers 1.5 times the interquartile range. Plots b and d show the effect sizes and the 95% credible intervals of population trends (b) and fluctuations (d) across Red List status categories. The μ values of population trend (plots a, e-f) and the σ^2 values of population fluctuation (plots c) were derived from state-space model fits of changes in abundance over the monitoring duration for each population. For the relationships between type and number of threats and population fluctuations, see Figure S19. Plots b and d show the effect sizes and the 95% credible intervals for population trends (b) and fluctuations (d) across IUCN Red List Categories. Effect sizes (plots b and d) were standardized by dividing the effect size by the standard deviation of the corresponding input data. Error bars in plots b and d show 95% credible intervals. Plot e shows the distributions of population trends across different threats that the species face globally, with the central tendencies of all distributions overlapping with zero. Lines in plot f show model fit and 95% credible intervals, where “number of threats” refers to the number of different threats that each species, whose populations are locally monitored, are exposed to on a global scale. See Figure 1 caption for further details on effect sizes, Methods for details on deriving the number and types of threats and Table S2 for model outputs.

3. The problem of the small vs. declining population paradigms being conflated remains. Small populations are often small for entirely different reasons for what drives populations into decline. In fact, most species are rare (the rarity of commonness, popularised by Kevin Gaston in many papers), so if they are naturally rare, or even if they have become rare, there is no expectation whatsoever that declines will be higher in such species. Thus, rarity (i.e., smallness) is not an adequate hypothesis to test for an effect on trending, even if the trend data are statistically robust (which in this case, mostly are not).

Response to reviewer 2 - 10:

We have revisited each mention of rarity in the manuscript to make sure that the specific uses of the rarity metrics and the interpretation of the analyses in our study are clear. We do not disagree that small versus large populations are not the same as rare versus common species, in fact this point was the original motivation behind our study.

We cite the work of Kevin Gaston in our manuscript (Gaston and Fuller, TREE 2008) and already refer to population dynamics theory that lays out these ideas in detail. Small populations do have different dynamics to large populations and rare species can have populations that experience different dynamics than common species. From our perspective, we have already addressed this comment in detail in our previous round of revisions and are not sure how to further address this reviewer comment as we do not believe we are conflating the concepts of small versus declining populations.

In our study, we set out to test whether there are differences in population trends within the LPD with respect to different metrics of rarity including geographic range, mean population size (number of animals that were recorded by monitoring) and habitat specificity independently. We found that species-level metrics, such as the three rarity metrics and the IUCN Red List Categories, did not explain the heterogeneity in trends of monitored populations in the Living Planet Database, and both species that would be classified as rare and common species using these four different metrics experienced declines, increases and fluctuations in population abundance over time (Figures 3 and 4).

“As per population dynamics theory^{22,23} and Taylor’s power law²⁴, species with small populations are more likely to undergo stochastic fluctuations that could lead to pronounced declines, local extinction and eventually global species extinction⁵. Small populations are also more likely to decline due to inbreeding, but note that there are also instances of naturally small and stable populations^{25,26}. Allee effects, the relationship between individual fitness and population density, further increase the likelihood of declines due to lack of potential mates and low reproductive output once populations reach a critically low density^{27,28}.” (lines 74-81)

Previous response from the last round of revisions addressing this point:

Here we describe in detail the revisions previously made to address this point in the text.

We thank the reviewer for raising this important point, we have clarified the distinction between small and declining populations in our manuscript and have added text to highlight that population declines and species' extinction occur on very different time scales (lines 83-86, 104-106, 362-364, 384-387), with the revised text being:

As per population dynamics theory^{22,23} and Taylor's power law²⁴, species with small populations are more likely to undergo stochastic fluctuations that could lead to pronounced declines, local extinction and eventually global species extinction⁵. Small populations are also more likely to decline due to inbreeding^{25,26}. Allee effects, the relationship between individual fitness and population density, further increase the likelihood of declines due to lack of potential mates and low reproductive output once populations reach a critically low density^{27,28}.

...

Testing population change across species' IUCN conservation status allows us to link contemporary changes in abundance with long-term probability of extinction³⁴.

...

While phylogenetic clustering might be lacking in contemporary trends, there is evidence that phylogenetic relatedness predicts extinction, a process occurring over much longer time scales^{6,7}.

...

We found that species with small populations were, nevertheless, more likely to fluctuate (Figure 3f), which may increase their probability of extinction, a process that could play out over longer time-scales than found for most population monitoring time-series^{22,23,54}.

Reviewer #3 (Remarks to the Author):

The authors have done a thorough and good job of responding to the comments from all three reviewers. This revised manuscript is a great improvement and I would really like to see this published. There are several remaining issues to address: three in direct response to the rebuttal, and the remainder are minor additional issues.

We thank the reviewer for their additional feedback on our manuscript and for the positive assessment of our revisions.

The addition of source code on a Github repository is welcome, but I was unable to access it. I logged in with the instructions provided, but received a message about an authentication code. I never received such a code so I assume it's gone to the lead author' email address. Therefore I have been unable to look at the source code. I should like to have an opportunity to review the code, as has been promised, before recommending that the article is suitable for publication.

Response to reviewer 3 - 1:

We are very sorry that the reviewer could not access the code. This authentication issue was not forwarded to us as authors. Authentication issues do happen with GitHub repositories when GitHub determines the access to be from a new user. We can make the repository public at this

time to allow this reviewer to assess the code (<https://github.com/gndaskalova/PopChangeRarity>, DOI 10.5281/zenodo.3817207). Because there are other groups working on similar analyses, we have always been uncertain about when to make our code publically available. In other studies, we make our code public at the time of submission. It is our aim that our reviewers have access to our code during review and we apologise that the reviewers were not able to access the code in this instance.

Early parts of the main text make predictions in the past tense (e.g. “we expected” on lines 72-3 and “we predicted” on line 80). These statements don’t make sense, because it’s still the introductory phase of the article. It’s not until line 111 that you define the focus and aims of the investigation. So reframing the introduction of these predictions would make the article more internally consistent.

Response to reviewer 3 - 2:

We thank the reviewer for bringing this to our attention. We used the past tense because we developed our hypotheses prior to conducting our analyses. But given that these sentences are in the earlier part of the introduction, we agree that the sentences should be reframed. We have now rephrased and reorganised these statements so that we give the ecological background information first and our predictions second, for example in the following:

“Here, we asked how the trends and fluctuations of vertebrate populations vary with biogeography, taxa, phylogenetic relationships and across species’ rarity metrics and IUCN Red List Categories and threat types from the species’ IUCN Red List profiles. We tested the following predictions: 1) There will be biogeographic patterns in population trends and fluctuations across the planet’s realms and biomes, in line with particular regions of the world experiencing high rates of environmental change (e.g., tropical forests³⁷). 2) Populations of rare species will be more likely to decline and fluctuate than the populations of common species. 3) Populations of species with a higher number of threats are more likely to decline and fluctuate than the populations of least concern species and those exposed to a lower number of threats.” (lines 100-108)

Re my comment about the contrast with the LPI headline finding: I’m pleased to note the new paragraph starting at line 392. I was slightly disappointed, however, that the discussion focussed only on the difference between this study and the LPI headline. Readers will want to know which is “real”.

Response to reviewer 3 - 3:

As per our response to reviewer 1 (Response to reviewer 2 - 6), we believe we cannot make a determination of global vertebrate trends from the Living Planet Database and thus cannot determine what is “real”. We can however, clearly state that the net change of individual time series in the Living Planet Database do not indicate overall declines as have been found with the Living Planet Index. We believe that the strong assumptions and weighting used in the Living Planet Index calculations nor the analysis of the population data as we have done do not represent the real-world net vertebrate population dynamics across the planet, as these data do not represent a large enough sample of populations of different species from sites around the world.

However, it was not the aim of our study to estimate the net global vertebrate population trend. Our aim was to test why some populations are increasing versus decreasing within the available Living Planet Database with respect to different factors such as taxa, geographic variation, and rarity. We have revised this paragraph further to make our stance more clear and to better communicate the limitations of interpreting global net change from heterogeneous samples of population time series. We include the new paragraph below:

“Our finding that declines are not universal, or even predominant, for vertebrate populations monitored for longer than five years in the Living Planet Database contrasts with reports of an overall decline in the Living Planet Index⁵⁶, a weighted summary of population change across all abundance time series in the Living Planet Database. Consistent with our results, the Living Planet Reports^{56–58} also document that the numbers of declining and increasing species are similar across this database, but the Living Planet Reports document a larger magnitude of population declines relative to increases. The calculation of the Living Planet Index involves differential weighting of population trends derived using logged abundance data, geometric means and generalized additive models, which could explain the discrepancies between our study findings and those of the Living Planet Reports⁵⁹. The Living Planet Index is hierarchically averaged from populations to species, taxa and realm and is also weighted by the estimated and relative number of species within biomes, which influences the direction and magnitude of the Living Planet Index^{59,60}. In contrast, our analysis explores the heterogeneity in local trends and fluctuations of monitored species from the raw population abundance data, and thus, we did not use an index with weightings and we did not aggregate population trends to a species-level. Rather than summarising trends with an index, our goal was to explain variability in abundance over time across better monitored vertebrates around the world. We detected net population declines at local scales over time only in the amphibian taxa, in contrast with the overall negative trend of the aggregate weightings of the Living Planet Index⁵⁶. We caution that distilling the heterogeneity of local population change at sites around the world into a simple metric may hide diverging trends at local scales, where we found both increases and declines among species.” (lines 373-393)

The current discussion raises more questions than it answers. What sorts of factors might explain the differences? For example, could it be that in averaging away the infraspecific variation in trends creates a bias in the headline LPI? Could this be a function of the scaling (from 0-1)? Is the ‘no-net change’ result an artefact of shrinkage? Or is the headline LPI driven solely by amphibian declines? There are some questions here waiting to be posed (although not answered - that is beyond the scope of investigation).

Response to reviewer 3 - 4:

We agree that there are many questions raised by the Living Planet Index methods. Recreating the Living Planet Index is difficult as the methods are not fully public and transparent and this was never a research goal of ours. We believe based on our understanding how the Living Planet Index is calculated (see Tables on pages 19-20 in the Living Planet Report Technical Supplement https://d2ouvy59p0dg6k.cloudfront.net/downloads/lpi_technical_supplement_2016.pdf, with one of the tables included below) that there are strong weightings towards declining populations

created by their taxonomic and system weightings. We did not want to raise questions that we cannot answer within the context of this manuscript, but we believe other authors are currently addressing some of these questions through other manuscripts that are currently under review, so this work may be forthcoming shortly. We include here the table of the weightings from the Living Planet Index work (ref) from which our assumption that taxonomic weightings might primarily create the strong declines in the index:

a. Terrestrial realm weightings applied to data:

	Afrotropical	Nearctic	Neotropical	Palaearctic	Indo-Pacific
Birds	0.387	0.376	0.387	0.433	0.396
Mammals	0.197	0.249	0.127	0.249	0.172
Reptiles and amphibians	0.414	0.373	0.484	0.316	0.431

b. Freshwater realm weightings applied to data:

	Afrotropical	Nearctic	Neotropical	Palaearctic	Indo-Pacific
Fishes	0.590	0.565	0.584	0.592	0.493
Birds	0.192	0.203	0.107	0.211	0.176
Mammals	0.009	0.013	0.010	0.015	0.008
Reptiles and amphibians	0.207	0.217	0.298	0.179	0.321

c. Marine realm weightings applied to data:

	Arctic	Atlantic North Temperate	Atlantic Tropical and Sub-tropical	Pacific North Temperate	Tropical and Sub-tropical Indo-Pacific	South Temperate and Antarctic
Fishes	0.792	0.920	0.922	0.892	0.940	0.922
Birds	0.172	0.068	0.069	0.080	0.048	0.054
Mammals	0.035	0.009	0.006	0.025	0.004	0.022
Reptiles	0	0.001	0.001	0.001	0.005	0.001

Reviewer 2 made a point about including estimates of uncertainty on the abundance estimates. This is another example where the authors dismissed the point on the grounds of “analysis not possible”, rather than engaging seriously with the criticism.

Response to reviewer 3 - 5:

We apologise if the reviewer felt that we were not appropriately engaging with a criticism. In this case, I believe we are actually already incorporating uncertainty in abundance estimates in a variety of ways in the manuscript, but in this specific case there are no available data on this particular source of uncertainty that we can incorporate, if we are making the correct assumption that the reviewer is referring to uncertainty in the abundance estimates for each monitoring time point in each time series. Here is the updated relevant section of the manuscript:

“The Living Planet Database combines population time series where survey methods were consistent within time series but varied among time series. Thus, among populations, abundance was measured using different units and over varying spatial extents. There are no estimates of error around the raw population abundance values available and detection probability likely varies among species. Thus, it is challenging to make informed decisions about baseline uncertainty in abundance estimates without prior information. We used state-space models to estimate trends and fluctuations to account for these limitations as this modelling framework is particularly appropriate for analyses of data collected using disparate methods^{39,79,80}. Another approach to partially account for observer error that has been applied to the analysis of population trends is the use of occupancy models³⁴. Because the precise coordinates of the polygons where the individual populations were monitored are not available, we were not able to test for the potential confounding effect of monitoring extent, but our sensitivity analysis indicated that survey units do not explain variation in the detected trends (Figure S13).” (lines 391-703)

I think the reviewer is right that such error could be important, but I also agree with authors that such data are not available and that it's unreasonable to expect them to find it. However, they really ought to acknowledge in the discussion that this uncertainty is important: it could lead to biases in large scale analyses and estimates thereof be made available wherever possible. I also note that in principle one could model this uncertainty without actually having data for it. In a state-space model, it's relatively straightforward to state that:

$Y_{\text{obs}} \sim \text{dnorm}(Y_{\text{true}}, \tau_{\text{obs}})$

$\tau_{\text{obs}} \leftarrow 1 / \sigma_{\text{obs}}^2$

where Y_{obs} is the observed data, Y_{true} is the unobserved truth, and σ_{obs} represents measurement error and/or sampling error. Currently you assume that $\sigma_{\text{obs}} = 0$, but you can ask the model to estimate the value of σ_{obs} that best fits the data.

Response to reviewer 3 - 6:

We are very strong believers in incorporating all available uncertainty in our work. We choose a Bayesian framework in our research because of the flexibility in how uncertainty can be added into models. However, in any statistical analysis a limitation is always the available data. Our philosophy is that adding in non-data informed uncertainty can be problematic and introduce bias. So our general approach is to not include sources of uncertainty that we cannot quantify. We agree with the general point that the reviewer is making here and believe that we are making a subjective yet appropriate analytical decision to use only data-informed estimates of uncertainty in our analytical framework.

I'm not seriously recommending the authors undertake such an analysis, but acknowledging the possibility would be worthwhile.

Response to reviewer 3 - 7:

We agree that abundance estimate error is important as well. We have revised this part of the discussion above to state that better capturing the uncertainty in abundance estimates would improve analyses of population change and that these data could be better captured in monitoring

programmes and large-scale databases such as the Living Planet Database. See revised sentence:

“We advocate for more detailed metadata in future versions of the Living planet database to allow researchers to filter the database appropriately for individual analyses.” (lines 756-757)

The van Strien et al (2016) paper would be worth citing here: the rebuttal gives the false impression that van Strien et al (2016) tried and failed to incorporate uncertainty in the species indices into their Dutch LPI: in fact they incorporated one important component of this uncertainty (the sampling error).

Response to reviewer 3 - 8:

We have revised our discussion of the van Strein et al. (2016) study to incorporate that this study accounted for observation error, by using occupancy models. Such approaches cannot be implemented in our study however. We have added a new sentence to the discussion which reads:

“Another approach to partially account for observer error that has been applied to the analysis of population trends is the use of occupancy models³⁴.” (lines 698-700)

Previous response from the last round of revisions addressing this point:

Here we address the point that reviewer 3 makes about time-specific measurement error around the population estimates.

The reviewer’s point about the lack of time-specific measurement error around the population estimates is valid, and we have added text to our manuscript that discusses this limitation of the database (see lines 679-689, and the full text of the new content in the response to the reviewer’s previous point). Incorporating the measurement error for each population (derived from the state-space models) in our modelling framework showed very similar results to when the measurement error was not included in the models, thus increasing our confidence in the finding of no relationships between rarity and population trends.

Here we address the issue that large databases can lead to the detection of statistically significant, yet not biologically meaningful relationships.

With large datasets such as the Living Planet Database, we can find small effects that are statistically significant, and equally we can find a lot of variation that could obscure relationships. This means that both Type I and Type II errors could potentially be at play, over which as the reviewer pointed out we have no control, but what we can do is use appropriate modelling framework (state-space models) that partition variance around population estimates into observation error and population fluctuations.

Here we were trying to communicate that there are a lot of different potential sources of error that we cannot always adequately estimate from the available data.

By making the code and data available, we are facilitating further engagement with the analyses, allowing other potentially interested researchers to explore details of our study that are beyond the scope of the present analysis. The transparency in our workflows also means that as new data become available and for example, scientific advances improve the precision and accuracy of the monitoring of rare species, our analyses can be re-run, thus testing if the finding of no relationship between rarity and population change is also valid for species not currently included in the Living Planet Database.

Here we are trying to make the point that we feel that this issue can be further explored in future analyses.

For further details, please see our responses above addressing the state-space modeling approach and limitations of the data set.

A quote:

“Monitoring extent and survey techniques

The Living Planet Database combines population time-series where survey methods were consistent within time-series but varied among time-series. Thus, among populations, abundance was measured using different units and over varying spatial extents. There are no estimates of error around the raw population abundance values available and detection probability likely varies among species. We used state-space models to estimate trends and fluctuations to account for these limitations as this modelling framework is particularly appropriate for analyses of data collected using disparate methods^{37,78,79}. Because the precise coordinates of the polygons where the individual populations were monitored are not available, we were not able to test for the potential confounding effect of monitoring extent, but our sensitivity analysis indicated that survey units do not explain variation in the detected trends (Figure S13).”

line 392: “our finding that all is not declines” - it took me several reads to understand this. Please rephrase, something like “Our finding that declines are not universal, or even predominant”

We have implemented this change.

Reviewer #4 (Remarks to the Author):

GENERAL

I will confess to being overwhelmed by the volume of information in this paper, the responses to the reviewers, and the supplemental materials: all told, 141 pages of materials and many dozens of complex figures and arguments. The authors have taken on a very important topic, and clearly devoted substantial efforts to addressing it, and I have mixed feelings and limited time in which to adjudicate between the reviewers and the authors.

Response to reviewer 4 - 1:

We thank the reviewer for agreeing to assess our manuscript and we agree that there is a large volume of work here to assess. We appreciate the mixed feeling you express and hopefully our revisions and thoughts below help clarify the points that you have raised. This study has been conducted across a four-year period and new questions and new analytical approaches have been raised throughout that period. There are so many interesting analyses that can be conducted with these data and we hope that our current manuscript provides a comprehensive take that will stimulate future research.

There are two remaining central issues to tackle in this paper:

(1) how many increasing and decreasing time series would you expect given the underlying data if there was no trend at all in the actual populations? i.e. what is the null distribution against which the results are compared?

Response to reviewer 4 - 2:

We did not form a specific a priori hypothesis with respect to the number of increasing or decreasing trends. It is difficult to form a specific a priori expectation for how many trends should be statistically significantly different from zero given the differing time series properties such as duration and number of time points but also population estimation methods contained within the Living Planet Database.

To address this point we have simulated random data with the same structure as the Living Planet Database and run our population trend analyses to test the null expectation. We have added this analysis to the main text and supp. info. in the following sections:

“We tested whether the number of increasing and decreasing populations trends differed from a null expectation using a data randomisation approach (Figure S6b). We used linear models to estimate trends in the data and randomized data with identical structure to the Living Planet Database. We found that there were over 10 times more population declines and increases in the real data relative to the randomised data (2.29% of trends were declining and 2.30% were increasing in the randomised data, versus 28.9% and 32.5% of time series which had significant negative and positive slopes in the real data, respectively).” (lines 772-778)

For point (1) some calculation needs to be made about how many time series are significantly different from zero. Perhaps this is exactly what the authors have done when reporting that 15% are declining and 18% are increasing, but this is not stated explicitly in the manuscript.

Response to reviewer 4 - 3:

We apologise for not providing appropriate clarity in the manuscript. We did indeed report the number time series with significant trends that differ from zero. We have now clarified and enhanced the communication of these results to make this more explicit in the manuscript. These results are now presented in the following way:

“We found a broad spectrum of population trends across vertebrate populations within the Living Planet Database. Across the time series we analysed, 15% (1381 time series) of populations were declining, 18% (1656 time series) were increasing, and 67% (6249 time series) showed no net changes in abundance over time, in contrast to a null distribution derived from randomised data (Figure S6b). Trends were considered statistically different from no net change when the confidence intervals around the population trend estimates did not overlap zero. Our results were similar when we weighted population trends by the state-space model derived observation error (Figures 1-4 and Tables S2-3).” (lines 145-152)

The second part of this is what is the power to detect trends in abundance, which is what reviewer 2 is honing in on.

Response to reviewer 4 - 4:

There are many issues with this large and complex dataset to assess with respect to statistical power. And there is not to our minds one simple power analysis that can capture the complexity here. We have however, conducted a randomisation analysis to test the difference in population trends from a null expectation (Figure S6). See response above (Response to reviewer 2 - 7). Reviewer 2 raises concerns about signal to noise in trend fitting and whether our results are statistical artifacts (Response to reviewer 2 - 1 to 8), but also whether the Living Planet Data capture meaningful trends in vertebrate populations (Response to reviewer 2 - 6). We have provided some additional information about generation times within the subsets of the data (Response to reviewer 2 - 4).

(2) Are the results correct and valid?

Response to reviewer 4 - 5:

There are multiple elements to this question. Specific comments about the index data raised by reviewer 4 that we address below and the ultimately broader questions raised by all of the reviewers that we also share about the Living Planet Database and how well it represents real-world population trends - not one we set out to address in our study. There are also generalizable questions about any large dataset and whether they capture meaningful real-world patterns in nature. We think that a key element to address when these questions are raised is the scope of the study. We do not believe that the Living Planet Data are a fully representative sample of vertebrate populations worldwide, but we do believe we can learn a lot from analysing patterns within these data to test what might explain why some populations are increasing versus decreasing. Results are only valid within the scope of the study and in the case of our study our scope is limited to comparisons within the Living Planet Database. We hope we have now better communicated our scope in the manuscript revisions. See other responses to reviewer comments.

For point (2) there are some odd anomalies in the estimates, with bimodal distributions of values, and many duplicate estimates of μ as outlined below. This needs to be investigated, and either corrected or explained in the text. My initial reaction is that there is some kind of failure in model convergence where the final answer ends up on a bound or equal to the starting values.

Response to reviewer 4 - 6:

We thank the reviewer for raising this concern. We have carefully revisited the state-space model outputs to explain the similar μ values found for multiple time series. We did not have model convergence issues and believe these results reflect elements to the time series data in Living Planet Database. We have explored the data in detail and formed our hypotheses as to why these patterns are emerging. Please see further details below in the following responses.

When these two issues can be sorted out, the paper should be published.

SPECIFIC COMMENTS

The phrase “all is not declines” is used in the title and throughout. This is grammatically incorrect and needs to be changed, perhaps to “Not all populations are declining” or “Nearly equal numbers of declining and increasing populations” or “Increasing populations are slightly more common than declining populations”.

Response to reviewer 4 - 7:

We agree that all is not declines is awkward and should have been all are not declines to be grammatically correct - this was a reference to the expression “all is not loss”, but the reference does not work effectively. We have chosen a new title that we hope better communicates the main finding of our study: “Rare and common vertebrates span a wide spectrum of population trends”.

Figure 1d: there is a problematic trend in the estimates of μ , in that there are bands where multiple species have exactly the same value of increase (+0.2) or decrease (-0.2), indeed zooming in there are additional bands at exactly +0.025 and -0.025, and at +0.1, +0.15, etc. Since each population should be independent, this suggests one of the following issues:

Response to reviewer 4 - 8:

We have carefully revisited our analyses and the underlying data and address the issues raised by reviewer 4 below. We have included a plot from Figure 1 of the manuscript and a remake of that plot using simple linear model slopes to illustrate the “banding” pattern being discussed here. The “banding” pattern is found in both the linear and state-space models, but is more prominent in the state-space models. These plots are made with a jittering of points, so the strength of the banding is related to the number of estimates at or around a given value.

(1) the model has a starting point at fixed values, and failed to converge to the true answer, resulting in multiple data points with exactly the same result;

Response to reviewer 4 - 9:

We have carefully checked convergence in our state-space modelling. Our code actually stops at a population if the model does not converge and all warnings and errors will be generated as the code runs. When designing the analysis we occasionally encountered convergence issues, but resolved them. The final version of our analysis runs without errors and all models converge such that we do not think the banding issue can be created by a model convergence problem.

(2) the same population appears multiple times in the dataset resulting in duplicate estimates;

Response to reviewer 4 - 10:

We always had a filter to remove any duplicated time series based on the study ID in our code from the early versions of the analysis. It is possible that population data are replicated with different study ID numbers and metadata within the Living Planet Database, but we have found no evidence for this in our inspection of individual time series, so it would only likely occur very rarely if there were replicated time series with different study IDs. With almost 10,000 time series in the Living Planet Database, we cannot visually inspect and compare all time series. However, we do not think replication of the same time series data explains this “banding” issue.

(3) some kind of numerical issue related to model fitting to integer values (hard to think of a scenario).

Response to reviewer 4 - 11:

The Living Planet Database includes a wide variety of different types of population data with different units. Some of these data, particularly using indexes to report abundance over time do not have the same properties of count data. It is possible that these types of data could be

modelled in such a way with the state-space models (and to a lesser extent with simple linear models) such that there are very similar μ values reported for different time series. To understand the different time series that are found in the “banding” in the plots, we have provided some new plots below and in the supp. info. of the manuscript.

There were 13 populations which had a μ value of exactly 0.2. Those populations were from different species (mostly birds, but other taxa too), from different coordinates.

Figure S6e. Raw population trend data behind 12 time series which had the same population trend values ($\mu = 0.20$). These time series are part of a “band” of time series which had very similar population trend estimates. Eighty, or approximately 1% of the time series we analysed form linear relationships over time with errors around the slopes of <0.001 , such that we suspect these data are modelled rather than measured population data. The presence of modelled data within the dataset may help explain the low variance bands of σ^2 values (c) and the pattern of two peaks in weak population increases and decreases for longer time series (d). Please see Methods sections “Time series with low variation” and “Clustering in the values of population trends and fluctuations” for further details.

As you can see in this plot of those 12 species (one plot omitted for visualisation purposes so that the plots fit on 3 lines) there are three plots where the abundance data are surprisingly linear (1355, 14902 and 2577). There are also two plots where the abundance data are surprisingly logistic (10193 and 468).

There were seven populations with exactly $\mu = 0.025$ (see below). Some of these studies seem to have lower than expected variance perhaps, yet they share a similar population trend, thus the estimates of a μ of 0.025 for scaled abundance data seem reasonable to the eye..

Response to reviewers Figure 1. Seven population time series part of the Living Planet Database had an estimated population trend value of 0.025, which given the raw abundance data, we consider to be reasonable estimates. The time series come from a variety of species, studies and locations.

Regardless, this needs to be thoroughly investigated to ensure that the results are not artefacts of the data or fitting process, especially since the bimodal distributions in freshwater and terrestrial realms are entirely driven by the duplicate estimates of μ at +0.025 and -0.025.

Similar issues are visible in all the other figures and in Figure S8. Perhaps there is a file drawer effect towards not publishing data if there is no trend ($\mu = 0$), explaining the missing peak at $\mu=0$? The cause of the issue appears to lie in the index values (Fig. S13).

Response to reviewer 4 - 12:

We have looked into different time series that either display unexpectedly low variance for a linear model fit or have the same estimated μ values. We find that 80 time series (<1% of the 9286 time series) had very little variance (see Table S8 for full references for those studies). We find that these time series are often index data, often found for bird species and are often from the North American Breeding Bird Survey (Table S8). We now report this study information and the study references in a new table in the supp. info. (Table S8). However, not all time series found within the “banding” have these specific characteristics.

We found that the bimodality and the trough in μ values was most pronounced in longer time series (see response to reviewer). We hypothesise that there might be a publication bias against publishing no net change studies (the “file drawer effect” described by reviewer 4), which could

explain the trough in μ values of around zero in long-term studies. There may be a connection between some of the low variance time series and the “banding” effect, which we attribute to the potential existence of abundance data modelled by demographic information and thus lacking natural variance (e.g., Nolet & Baveco 1996 Biological Conservation).

We estimate that these issues are influencing less than 1% of the Living Planet Database time series, and thus will not alter our overall study findings. We advocate that future releases of the Living Planet Database include more detailed metadata information such that these types of issues can be better assessed and time series can be more easily filtered in the process of analysis of the data. We hope that future research will shed further light on this issue.

We have added the following paragraphs on this issue to the supp. info.:

“Time series with low variation

Eighty populations (<1% of the 9286 time series) had very little variance (see Table S7 for full references for those studies). The majority of those studies are for bird species and come from the North American breeding bird survey with a measurement unit of an index⁹⁰. We have also observed some time series that appear to show logistic relationships with little natural variance (e.g., time series 468, 10193, 17803, see Table S8 for full references). Inspecting the raw data showed that some populations have abundances which follow an almost perfect linear or logarithmic increase over time, as could be the case for modelled, versus raw field data. We provide the references for these studies and cannot definitely attribute the low variance to a particular cause across all studies. Some of these studies are reported in units that are an index which may not capture variation in the same way as other raw units of population data. Some of these time series may represent modelled population data based on demographic information rather than only direct observations of populations (e.g., time series 135591). We chose to not remove studies that may not be raw observation time series based on visual inspection of trends to avoid introducing bias against populations with naturally low variation into our analysis.” (lines 724-738)

“Clustering in the values of population trends and fluctuations

We found a clustering of population trend and fluctuations values in some parts of the population change spectrum. For example, we found two peaks – in small increases and in small decreases over time – which were most prevalent in terrestrial bird studies and species which were monitored using an index (Figure 2, Figure S13). Overall 11.4% of time series had trend values between 0.02 and 0.03 and 11.6% of time series had trend values between -0.03 and -0.02. There was also a similar, but smaller, clustering around trends of 0.25 and -0.25. All reported population trends are from models that converged successfully, and visual inspection indicated to us that the μ values are appropriate estimates for the individual time series (Figure S7e). We investigated the population time series where the value of the population trends over time were estimated to be the same value and found that they came from a variety of taxa, locations and survey methods (Figure S7e). We hypothesise that there might be a publication bias against publishing no net change studies, which could explain the trough in μ values of around zero in long-term studies. The clustering of values for some time series may sometimes be associated with the same time

series that also have low variance (Figure S7e, see discussion above). With the information available in the Living Planet Database metadata, we cannot fully explain the clustering in population trends. We advocate for more detailed metadata in future versions of the Living planet database to allow researchers to filter the database appropriately for individual analyses.” (lines 740-757)

Figure 1f and other plots of σ^2 : the y-axis should be plotted on a log scale so that it is possible to compare variance across realms. An alternative is to plot the coefficient of variation, $CV = \sigma/\mu$, which is the proportion of error, and commonly used to compare values that have different means.

Response to reviewer 4 - 13:

We have plotted the σ^2 on the log scale so that visual comparisons across realms are easier to assess in Figures 1, 2, 3, 4 and S7. We include an example revised figure below.

Figure 2. Population trends and fluctuations varied more among, rather than within, taxa, with amphibians being the only group showing pronounced declines over time. (full caption in manuscript)

lines 463-465: Reviewer 2 is correct in being skeptical that five time points are sufficient to detect population trends in 80-90% of cases. Note, however, that ref 68 is now published in *Methods in Ecology and Evolution* (<https://besjournals.onlinelibrary.wiley.com/doi/abs/10.1111/2041-210X.13302>). The key is that there are two items being measured: the magnitude of the trend, and the significance of the trend. The significance depends on the number of data points being fitted, and the variability in the data points being fitted. For short time series, most trends will be non-significant and will also have magnitudes that vary widely—the classic cone shape in meta-analysis studies seen in Figure S6. Thus it is more important in this paper to highlight how many populations are increasing or declining significantly (at some level of alpha), than to focus on the vast majority of populations with no detected trend.

Response to reviewer 4 - 14:

We entirely agree with this assessment of trends and variance around trends and statistical significance and that these are general properties of any time series data. These trend magnitude and significance relationships influence all sorts of interpretation of time series data. We have

now described the results of the Wauchope et al. 2019 study in greater detail and in response to the comments from reviewer 2. See Response to reviewer 2 - 3 above.

Often studies do only assess the trends in the absence of the variance around those trends and thus many detected trends will indeed be due to chance, for example in the remote sensing literature and trends in spectral vegetation indices. In our study however, we have also conducted second stage analyses in a Bayesian framework where we incorporated both the mean and the variance of the first stage models into the second stage analyses (see “weighted” estimate for effect sizes on graphs throughout the manuscript, with the weight in the models being the standard error around the population trend). This allows the variance around trends to be incorporated into the overall model estimates at the second stage.

Significance testing is a subjective framework - arbitrary cut offs are set to class trends as significant or not. Our own analytical preference is to retain all time series for our second stage analyses and to not analyse the “statistically significant” trends in isolation from other trends. We believe that the time series where no or weaker trends were detected also represent population dynamics relevant to our covariates of interest. However, we do not want to obscure information about the proportion of population trends that were statistically significant. We now report these results more clearly in the manuscript text and in the figure captions (please see main text for figure captions text).

“We found a broad spectrum of population trends across vertebrate populations within the Living Planet Database. Across the time series we analysed, 15% (1381 time series) of populations were declining, 18% (1656 time series) were increasing, and 67% (6249 time series) showed no net changes in abundance over time, in contrast to a null distribution derived from randomised data (Figure S6b). Trends were considered statistically different from no net change when the confidence intervals around the population trend estimates did not overlap zero. Our results were similar when we weighted population trends by the state-space model derived observation error (Figures 1-4 and Tables S2-3).” (lines 145-152)

I disagree with reviewer 2 that 50 generations is relevant to this paper. The paper is not looking to estimate density-dependent effects, only trends over time. For some species, like rockfish (Sebastes) with generation times of 30 yr, it is obviously not reasonable to require 1500 yr of data before trends in abundance can be estimated. With overlapping generations, small sample SEs, and large changes in abundance, it can easily be possible to detect trends with 3 data points.

Response to reviewer 4 - 15:

We agree with reviewer 4’s comments, yet we wished to provide more information in response to reviewer 2’s comments on generation times for a subset of the Living Planet Database species for which generation time data are available. We had already extracted generation time information for a related study (Daskalova et al. Science accepted). We have now included this information in the supp. info. for other interested readers. See Response to reviewer 2 - 4 above.

The point that reviewer 3 makes about uncertainty in the abundance estimates is I believe covered by the state-space models estimating both observation and process error.

Response to reviewer 4 - 16:

We believe that reviewer 3 is referring to time-point-level measurement error rather than the time-series-level estimates of observation and process error, though it is not entirely clear in the way the comments are stated. We do not have information on the uncertainty at the base level of the Living Planet Database. These data include different survey methods, population sizes and many other factors that would influence the uncertainty in the time-point-level estimates. We feel that in our analytical framework we should not estimate uncertainty without data to inform those estimates. See Response to reviewer 3 - 6 above.

Reviewers' Comments:

Reviewer #3:

Remarks to the Author:

I reviewed at least two previous versions of this manuscript. At the last occasion I recommended publication. I have been invited to review this new version as a referee between the views of Reviewer 2 (who was very skeptical) and Reviewer 4 (who was broadly sympathetic). I have therefore focussed my attention on the comments by R2 and R4, and the authors' responses to them.

My take is that R2s criticisms are mainly directed at the data, rather than the analysis. The authors have produced a very considerable quantity of supplementary analysis to explore the issues raised by R2 (and others). In fact, the authors have done everything possible to address these concerns, short of actually collect fresh data. Given that the same dataset is used by WWF to present an annual dire warning about biodiversity losses, it seems entirely reasonable to present this rigorous and thorough analysis which happens to show something different.

Reviewer 4's points are mostly requests for greater clarity. To each point, the authors have provided a clear response and made changes that will direct naive readers to the appropriate information. The only substantive issue concerned the potential artefact of a bimodal peak in population trends, which is referred to as "banding". The authors have responded with a new and very interesting analysis, which raises some questions about the quality of data going into the LPDD, whilst also demonstrating convincingly that it is not an artefact of their own modelling approach.

Under different circumstances I might make additional minor comments about what should be altered. However, this would be above and beyond the role that I have been asked to play. Moreover, the authors have patiently endured many rounds of peer review over more than 20 months. In the words of R4, my feeling is that all issues have been "sorted out", and that therefore "the paper should be published."

Nick Isaac

REVIEWERS' COMMENTS:

Reviewer #3 (Remarks to the Author):

I reviewed at least two previous versions of this manuscript. At the last occasion I recommended publication. I have been invited to review this new version as a referee between the views of Reviewer 2 (who was very skeptical) and Reviewer 4 (who was broadly sympathetic). I have therefore focussed my attention on the comments by R2 and R4, and the authors' responses to them.

My take is that R2s criticisms are mainly directed at the data, rather than the analysis. The authors have produced a very considerable quantity of supplementary analysis to explore the issues raised by R2 (and others). In fact, the authors have done everything possible to address these concerns, short of actually collect fresh data. Given that the same dataset is used by WWF to present an annual dire warning about biodiversity losses, it seems entirely reasonable to present this rigorous and thorough analysis which happens to show something different.

Reviewer 4's points are mostly requests for greater clarity. To each point, the authors have provided a clear response and made changes that will direct naive readers to the appropriate information. The only substantive issue concerned the potential artefact of a bimodal peak in population trends, which is referred to as "banding". The authors have responded with a new and very interesting analysis, which raises some questions about the quality of data going into the LPDD, whilst also demonstrating convincingly that it is not an artefact of their own modelling approach.

Under different circumstances I might make additional minor comments about what should be altered. However, this would be above and beyond the role that I have been asked to play. Moreover, the authors have patiently endured many rounds of peer review over more than 20 months. In the words of R4, my feeling is that all issues have been "sorted out", and that therefore "the paper should be published."

Nick Isaac

We appreciate the time and work that Reviewer 3 has dedicated to our manuscript which has significantly improved our work.